# Variance Reduction is an Antidote to Byzantine Workers: Better Rates, Weaker Assumptions and Communication Compression as a Cherry on the Top

**Eduard Gorbunov**[*]
MBZUAI, MIPT
Mila & UdeM, KAUST

**Samuel Horváth**[†]
MBZUAI, KAUST

**Peter Richtárik**
KAUST

**Gauthier Gidel**
Mila & UdeM
Canada CIFAR AI Chair

## Abstract

Byzantine-robustness has been gaining a lot of attention due to the growth of the interest in collaborative and federated learning. However, many fruitful directions, such as the usage of variance reduction for achieving robustness and communication compression for reducing communication costs, remain weakly explored in the field. This work addresses this gap and proposes Byz-VR-MARINA–a new Byzantine-tolerant method with variance reduction and compression. A key message of our paper is that variance reduction is key to fighting Byzantine workers more effectively. At the same time, communication compression is a bonus that makes the process more communication efficient. We derive theoretical convergence guarantees for Byz-VR-MARINA outperforming previous state-of-the-art for general non-convex and Polyak-Łojasiewicz loss functions. Unlike the concurrent Byzantine-robust methods with variance reduction and/or compression, our complexity results are tight and do not rely on restrictive assumptions such as boundedness of the gradients or limited compression. Moreover, we provide the first analysis of a Byzantine-tolerant method supporting non-uniform sampling of stochastic gradients. Numerical experiments corroborate our theoretical findings.

## 1 Introduction

Distributed optimization algorithms play a vital role in the training of the modern machine learning models. In particular, some tasks require training of deep neural networks having billions of parameters on large datasets (Brown et al., 2020; Kolesnikov et al., 2020). Such problems may take years of computations to be solved if executed on a single yet powerful machine (Li, 2020). To circumvent this issue, it is natural to use distributed optimization algorithms allowing to tremendously reduce the training time (Goyal et al., 2017; You et al., 2020). In the context of speeding up the training, distributed methods are usually applied in data centers (Mikami et al., 2018). More recently, similar ideas have been applied to train models using open collaborations (Kijsipongse et al., 2018; Diskin et al., 2021), where each participant (e.g., a small company/university or an individual) has very limited computing power but can donate it to jointly solve computationally-hard problems. Moreover, in Federated Learning (FL) applications (McMahan et al., 2017; Konečný et al., 2016; Kairouz et al., 2021), distributed algorithms are natural and the only possible choice since in such problems, the data is *privately* distributed across multiple devices.

In the optimization problems arising in collaborative and federated learning, there is a high risk that some participants deviate from the prescribed protocol either on purpose or not. In this paper, we

---

[*]Corresponding author: eduard.gorbunov@mbzuai.ac.ae. Part of the work was done when E. Gorbunov was a researcher at MIPT and Mila & UdeM and also a visiting researcher at KAUST, in the Optimization and Machine Learning Lab of P. Richtárik.

[†]Part of the work was done when S. Horváth was a Ph.D. student at KAUST.

call such participants as Byzantine workers[1] For example, such peers can maliciously send incorrect gradients to slow down or even destroy the training. Indeed, these attacks can break the convergence of naïve methods such as Parallel-SGD (Zinkevich et al., 2010). Therefore, it is crucial to use secure (a.k.a. Byzantine-robust/Byzantine-tolerant) distributed methods for solving such problems.

However, designing distributed methods with provable Byzantine-robustness is not an easy task. The non-triviality of this problem comes from the fact that the stochastic gradients of good/honest/regular workers are naturally different due to their stochasticity and possible data heterogeneity. At the same time, malicious workers can send the vectors looking like the stochastic gradients of good peers or create small but time-coupled shifts. Therefore, as it is shown in (Baruch et al., 2019; Xie et al., 2020; Karimireddy et al., 2021), Byzantine workers can circumvent popular defences based on applying robust aggregation rules (Blanchard et al., 2017; Yin et al., 2018; Damaskinos et al., 2019; Guerraoui et al., 2018; Pillutla et al., 2022) with Parallel-SGD. Moreover, in a broad class of problems with heterogeneous data, it is provably impossible to achieve any predefined accuracy of the solution (Karimireddy et al., 2022; El-Mhamdi et al., 2021).

Nevertheless, as it becomes evident from the further discussion, several works have provable Byzantine tolerance and rigorous theoretical analysis. In particular, Wu et al. (2020) propose a natural yet elegant solution to the problem of Byzantine-robustness based on the usage of variance-reduced methods (Gower et al., 2020) and design the first variance-reduced Byzantine-robust method called Byrd-SAGA, which combines the celebrated SAGA method (Defazio et al., 2014) with geometric median aggregation rule. As a result, reducing the stochastic noise of estimators used by good workers makes it easier to filter out Byzantine workers (especially in the case of homogeneous data). However, Wu et al. (2020) derive their results only for the strongly convex objectives, and the obtained convergence guarantees are significantly worse than the best-known convergence rates for SAGA, i.e., their results are not tight, even when there are no Byzantine workers and all peers have homogeneous data. It is crucial to bypass these limitations since the majority of the modern, practically interesting problems are non-convex. Furthermore, it is hard to develop the field without tight convergence guarantees. All in all, the above leads to the following question:

> **Q1:** *Is it possible to design variance-reduced methods with provable Byzantine-robustness and tight theoretical guarantees for general non-convex optimization problems?*

In addition to Byzantine-robustness, one has to take into account that naïve distributed algorithms suffer from the so-called *communication bottleneck*—a situation when communication is much more expensive than local computations on the devices. This issue is especially evident in the training of models with a vast number of parameters (e.g., millions or trillions) or when the number of workers is large (which is often the case in FL). One of the most popular approaches to reducing the communication bottleneck is to use *communication compression* (Seide et al., 2014; Konečný et al., 2016; Suresh et al., 2017), i.e., instead of transmitting dense vectors (stochastic gradients/Hessians/higher-order tensors) workers apply some compression/sparsification operator to these vectors and send the compressed results to the server. Distributed learning with compression is a relatively well-developed field, e.g., see (Vogels et al., 2019; Gorbunov et al., 2020b; Richtárik et al., 2021; Philippenko & Dieuleveut, 2021) and references therein for the recent advances.

Perhaps surprisingly, there are not many methods with compressed communication in the context of Byzantine-robust learning. In particular, we are only aware of the following works (Bernstein et al., 2018; Ghosh et al., 2020; 2021; Zhu & Ling, 2021),. Bernstein et al. (2018) propose signSGD to reduce communication cost and and study the majority vote to cope with the Byzantine workers under some additional assumptions about adversaries. However, it is known that signSGD is not guaranteed to converge (Karimireddy et al., 2019). Next, Ghosh et al. (2020; 2021) apply aggregation based on the selection of the norms of the update vectors. In this case, Byzantine workers can successfully hide in the noise applying SOTA attacks (Baruch et al., 2019). Zhu & Ling (2021) study Byzantine-robust versions of compressed SGD (BR-CSGD) and SAGA (BR-CSAGA) and also propose a combination of DIANA (Mishchenko et al., 2019; Horváth et al., 2019b) with BR-CSAGA called BROADCAST. However, the derived convergence results for these methods have several limitations. First of all, the analysis is given only for strongly convex problems. In addition, it

---

[1]This term is standard for distributed learning literature (Lamport et al., 1982; Su & Vaidya, 2016; Lyu et al., 2020). Using this term, we follow standard terminology and do not want to offend any group. It would be great if the community found and agreed on a more neutral term to denote such workers.

Table 1: Comparison of the state-of-the-art (in theory) Byzantine-tolerant distributed methods. Columns: "NC" = does the theory works for general smooth non-convex functions?; "PL" = does the theory works for functions satysfying PŁ-condition (As. 2.5)?; "Tight?" = does the theory recover tight best-known results for the version of the method with $\delta = 0$ (no Byzantines)?; "Compr.?" = does the method use communication compression?; "VR?" = is the method variance-reduced?; "No UBV?" = does the theory work without assuming uniformly bounded variance of the stochastic gradients, i.e., without the assumption that for all $x \in \mathbb{R}^d$ the good workers have an access to the unbiased estimators $g_i(x)$ of $\nabla f_i(x)$ such that $\mathbb{E}\|g_i(x) - \nabla f_i(x)\|^2 \leq \sigma^2$ for all $i \in \mathcal{G}$ and $\sigma \geq 0$?; "No BG?" = does the theory work without assuming uniformly bounded second moment of the stochastic gradients, i.e., without the assumption that for all $x \in \mathbb{R}^d$ the good workers have an access to the unbiased estimators $g_i(x)$ of $\nabla f_i(x)$ such that $\mathbb{E}\|g_i(x)\|^2 \leq D^2$ for all $i \in \mathcal{G}$ and $\sigma > 0$?; "Non-US?" = does the theory support non-uniform sampling of the stochastic gradients; "Het.?" = does the theory work under $\zeta^2$-heterogeneity assumption (As. 2.2)?

| Method | NC | PL | Tight? | Compr.? | VR? | No UBV? | No BG? | Non-US? | Het.? |
|--------|----|----|--------|---------|-----|---------|--------|---------|-------|
| BR-SGDm (Karimireddy et al., 2021; 2022) | ✓ | ✗ | ✓ | ✗ | ✗ | ✗ | ✓ | ✗ | ✓ |
| BTARD-SGD (Gorbunov et al., 2021a) | ✓ | ✗✓[(1)] | ✓ | ✗ | ✗ | ✗ | ✓ | ✗ | ✗ |
| Byrd-SAGA (Wu et al., 2020) | ✗ | ✗✓[(1)] | ✗ | ✗ | ✓ | ✓ | ✓ | ✗ | ✓ |
| BR-MVR (Karimireddy et al., 2021) | ✓ | ✗ | ✓ | ✗ | ✓ | ✗ | ✓ | ✗ | ✗ |
| BR-CSGD (Zhu & Ling, 2021) | ✗ | ✗✓[(1)] | ✗ | ✓ | ✗ | ✗ | ✗ | ✗ | ✓ |
| BR-CSAGA (Zhu & Ling, 2021) | ✗ | ✗✓[(1)] | ✗ | ✓ | ✓ | ✗ | ✗ | ✗ | ✓ |
| BROADCAST (Zhu & Ling, 2021) | ✗ | ✗✓[(1)] | ✗ | ✓ | ✓ | ✓ | ✓ | ✗ | ✓ |
| Byz-VR-MARINA [This work] | ✓ | ✓ | ✓ | ✓ | ✓ | ✓ | ✓ | ✓ | ✓ |

[(1)] Strong convexity of $f$ is assumed.

relies on restrictive assumptions. Namely, Zhu & Ling (2021) assume uniform boundedness of the second moment of the stochastic gradient in the analysis of BR-CSGD and BR-CSAGA. This assumption rarely holds in practice, and it also implies the boundedness of the gradients, which contradicts the strong convexity assumption. Next, although the bounded second-moment assumption is not used in the analysis of BROADCAST, Zhu & Ling (2021) derive the rates of BROADCAST under the assumption that the compression operator is very accurate, which implies that in theory workers apply almost no compression to the communicated messages (see remark (5) under Table 2). Finally, even if there are no Byzantine workers and no compression, similar to the guarantees for Byrd-SAGA, the rates obtained for BR-CSGD, BR-CSAGA, and BROADCAST are outperformed with a large margin by the known rates for SGD and SAGA. All of these limitations lead to the following question:

**Q2:** *Is it possible to design distributed methods with compression, provable Byzantine-robustness and tight theoretical guarantees without making strong assumptions?*

In this paper, we give confirmatory answers to **Q1** and **Q2** by proposing and rigorously analyzing a new Byzantine-tolerant variance-reduced method with compression called Byz-VR-MARINA. Detailed related work overview is deferred to Appendix A.

**Our Contributions.** Before we proceed, we need to specify the targeted problem. We consider a centralized distributed learning in the possible presence of malicious or so-called *Byzantine* peers. We assume that there are $n$ clients consisting of the two groups: $[n] = \mathcal{G} \sqcup \mathcal{B}$, where $\mathcal{G}$ denotes the set of good clients and $\mathcal{B}$ is the set of bad/malicious/Byzantine workers. The goal is to solve the following optimization problem

$$\min_{x \in \mathbb{R}^d} \left\{ f(x) = \frac{1}{G} \sum_{i \in \mathcal{G}} f_i(x) \right\}, \quad f_i(x) = \frac{1}{m} \sum_{j=1}^{m} f_{i,j}(x) \quad \forall i \in \mathcal{G}, \tag{1}$$

where $G = |\mathcal{G}|$ and functions $f_{i,j}(x)$ are assumed to be smooth, but not necessarily convex. Here each good client has its dataset of the size $m$, $f_{i,j}(x)$ is the loss of the model, parameterized by vector $x \in \mathbb{R}^d$, on the $j$-th sample from the dataset on the $i$-th client. Following the classical convention

**Table 2:** Comparison of the state-of-the-art complexity results for Byzantine-tolerant distributed methods. Columns: "Assumptions" = additional assumptions to smoothness of all $f_i(x)$, $i \in \mathcal{G}$ (although our results require more refined As. 2.3); "Complexity (NC)" and "Complexity (PŁ)" = number of communication rounds required to find such $x$ that $\mathbb{E}\|\nabla f(x)\|^2 \leq \varepsilon^2$ in the general non-convex case and such $x$ that $\mathbb{E}[f(x) - f(x^*)] \leq \varepsilon$ in PŁ case respectively. Dependencies on numerical constants (and logarithms in PŁ setting), smoothness constants, and initial suboptimality are omitted in the complexity bounds. Although BR-SGDm, BR-MVR, BTARD-SGD, Byrd-SAGA, BR-CSGD, BR-CSAGA, BROADCAST are analyzed for unit batchsize only ($b = 1$), one can easily generalize them to the case of $b > 1$ and we show these generalizations in the table. Notation: $\varepsilon$ = desired accuracy; $\delta$ = ratio of Byzantine workers; $c$ = parameter of the robust aggregator; $n$ = total number of workers; $b$ = batchsize; $\sigma^2$ = uniform bound on the variance of stochastic gradients; $D^2$ = uniform bound on the second moment of stochastic gradients; $C$ = the number of workers used by BTARD-SGD for the checks of computations after each step; $\mu$ = parameter from As. 2.5 (strong convexity parameter in the case of BTARD-SGD, Byrd-SAGA, BR-CSGD, BR-CSAGA, BROADCAST); $m$ = size of the local dataset on workers; $p = \min\{b/m, 1/(1+\omega)\}$ = probability of communication in Byz-VR-MARINA.

| Setup | Method | Assumptions | Complexity (NC) | Complexity (PŁ) |
|---|---|---|---|---|
| Hom. data, no compr. | BR-SGDm
(Karimireddy et al., 2021; 2022) | UBV | $\frac{1}{\varepsilon^2} + \frac{\sigma^2(c\delta+1/n)}{b\varepsilon^4}$ | ✗ |
| | BR-MVR
(Karimireddy et al., 2021) | UBV | $\frac{1}{\varepsilon^2} + \frac{\sigma\sqrt{c\delta+1/n}}{\sqrt{b}\varepsilon^3}$ | ✗ |
| | BTARD-SGD
(Gorbunov et al., 2021a) | UBV[1] | $\frac{1}{\varepsilon^2} + \frac{n^2\delta\sigma^2}{Cb\varepsilon^2} + \frac{\sigma^2}{nb\varepsilon^4}$ | $\frac{1}{\mu} + \frac{\sigma^2}{nb\mu\varepsilon} + \frac{n^2\delta\sigma}{C\sqrt{b\mu\varepsilon}}$[7] |
| | Byrd-SAGA[2]
(Wu et al., 2020) | Smooth $f_{i,j}$ | ✗ | $\frac{m^2}{b^2(1-2\delta)\mu^2}$[7] |
| | Byz-VR-MARINA
**Cor. E.1 & Cor. E.5** | As. 2.4 | $\frac{1+\sqrt{\frac{c\delta m^2}{b^3}+\frac{m}{b^2 n}}}{\varepsilon^2}$ | $\frac{1+\sqrt{\frac{c\delta m^2}{b^3}+\frac{m}{b^2 n}}}{\mu}+\frac{m}{b}$ |
| Het. data, no compr. | BR-SGDm[3]
(Karimireddy et al., 2022) | UBV | $\frac{1}{\varepsilon^2} + \frac{\sigma^2(c\delta+1/n)}{b\varepsilon^4}$ | ✗ |
| | Byrd-SAGA[2],[3]
(Wu et al., 2020) | Smooth $f_{i,j}$ | ✗ | $\frac{m^2}{b^2(1-2\delta)\mu^2}$[7] |
| | Byz-VR-MARINA[3],[4]
**Cor. E.2 & Cor. E.6** | As. 2.4 | $\frac{1+\sqrt{\frac{c\delta m^2}{b^2}(1+\frac{1}{b})+\frac{m}{b^2 n}}}{\varepsilon^2}$ | $\frac{1+\sqrt{\frac{c\delta m^2}{b^2}(1+\frac{1}{b})+\frac{m}{b^2 n}}}{\mu}+\frac{m}{b}$ |
| Het. data, compr. | BR-CSGD[2],[3]
(Zhu & Ling, 2021) | UBV, BG | ✗ | $\frac{1}{\mu^2}$[7] |
| | BR-CSAGA[2],[3]
(Zhu & Ling, 2021) | Smooth $f_{i,j}$
UBV, BG | ✗ | $\frac{m^2}{b^2\mu^2(1-2\delta)^2}$[7] |
| | BROADCAST[2],[3],[5]
(Zhu & Ling, 2021) | Smooth $f_{i,j}$ | ✗ | $\frac{m^2(1+\omega)^{3/2}}{b^2\mu^2(1-2\delta)}$[7] |
| | Byz-VR-MARINA[3],[6]
**Cor. E.3 & Cor. E.7** | As. 2.4 | $\frac{1+\sqrt{c\delta(1+\omega)(1+\frac{1}{b})}}{p\varepsilon^2}$ $+\frac{\sqrt{(1+\omega)(1+\frac{1}{b})}}{\sqrt{pn}\varepsilon^2}$ | $\frac{1+\sqrt{c\delta(1+\omega)(1+\frac{1}{b})}}{p\mu}$ $+\frac{\sqrt{(1+\omega)(1+\frac{1}{b})}}{\sqrt{pn}\mu}$ $+\frac{m}{b}+\omega$ |

[1] Gorbunov et al. (2021a) assume additionally that the tails of the noise distribution in stochastic gradients are sub-quadratic.

[2] Although the analyses by Wu et al. (2020); Zhu & Ling (2021) support inexact geometric median computation, for simplicity of presentation, we assume that geometric median is computed exactly.

[3] BR-SGDm: $\varepsilon^2 = \Omega(c\delta\zeta^2)$; Byrd-SAGA: $\varepsilon = \Omega(\zeta^2/(\mu^2(1-2\delta)^2))$; Byz-VR-MARINA: $\varepsilon^2 = \Omega(\max\{m/b, 1+\omega\}c\delta\zeta^2)$ for general non-convex case and $\varepsilon = \Omega(\max\{m/b, 1+\omega\}c\delta\zeta^2/\mu)$ for the case of PŁ functions (with $\omega = 0$, where there is no compression); BR-CSGD: $\varepsilon = \Omega((\sigma^2+\zeta^2+\omega D^2)/(\mu^2(1-2\delta)^2))$ (positive even when $\zeta^2 = 0$); BR-CSAGA: $\varepsilon = \Omega((\zeta^2+\omega D^2)/(\mu^2(1-2\delta)^2))$ (positive even when $\zeta^2 = 0$); BROADCAST: $\varepsilon = \Omega((1+\omega)\zeta^2/(\mu^2(1-2\delta)^2))$.

[4] The term $\frac{m\sqrt{c\delta}}{b\varepsilon^2}$ is proportional to much smaller Lipschitz constant than the term $\frac{m\sqrt{c\delta}}{b^{3/2}\varepsilon^2}$ does. A similar statement holds in PŁ case as well.

[5] For this result Zhu & Ling (2021) assume that $\omega \leq \frac{\mu^2(1-2\delta)^2}{56L^2(2-2\delta^2)}$, which is a very restrictive assumption even when $\delta = 0$. For example, even for well-conditioned problems with $\mu/L \sim 10^{-3}$ and $\delta = 0$ (no Byzantine workers), this bound implies that $\omega$ should be not larger than $10^{-7}$. Such a value of $\omega$ corresponds to almost non-compressed communications.

[6] The term $\frac{1+\sqrt{c\delta(1+\omega)}}{p\varepsilon^2} + \frac{\sqrt{1+\omega}}{\sqrt{pn}\varepsilon^2}$ is proportional to much smaller Lipschitz constant than the term $\frac{1+\sqrt{c\delta(1+\omega)}}{\sqrt{b}p\varepsilon^2} + \frac{\sqrt{1+\omega}}{\sqrt{pnb}\varepsilon^2}$ does. A similar statement holds in PŁ case as well.

[7] The rate is derived under the strong convexity assumption. Strong convexity implies PŁ-condition but not vice versa: there exist non-convex PŁfunctions (Karimi et al., 2016).

(Lyu et al., 2020), we make no assumptions on the malicious workers $\mathcal{B}$, i.e., Byzantine workers are allowed to be *omniscient*. **Our main contributions are summarized below.**

⬦ **New method:** Byz-VR-MARINA. We propose a new Byzantine-robust variance-reduced method with compression called Byz-VR-MARINA (Alg. 1). In particular, we make VR-MARINA (Gorbunov et al., 2021b), which is a variance-reduced method with compression, applicable to the context of Byzantine-tolerant distributed learning via using the recent tool of robust agnostic aggregation of Karimireddy et al. (2022). As Tbl. 1 shows, Byz-VR-MARINA and our analysis of the method leads to several important improvements upon the previously best-known methods.

⬦ **New SOTA results.** Under quite general assumptions listed in Section 2, we prove theoretical convergence results for Byz-VR-MARINA in the cases of smooth non-convex (Thm. 2.1) and Polyak-

Łojasiewicz (Thm. 2.2) functions. As Tbl. 2 shows, our complexity bounds in the non-convex case are always better than previously known ones when the target accuracy $\varepsilon$ is small enough. In the PŁ case, our results improve upon previously known guarantees when the problem has bad conditioning or when $\varepsilon$ is small enough. Moreover, we provide the first theoretical convergence guarantees for Byzantine-tolerant methods with compression in the non-convex case for arbitrary adversaries.

$\diamond$ **Byzantine-tolerant variance-reduced method with tight rates.** Our results are tight, i.e., when there are no Byzantine workers, our rates recover the rates of VR-MARINA, and when additionally no compression is applied, we recover the optimal rates of Geom-SARAH (Horváth et al., 2022)/PAGE (Li et al., 2021). In contrast, this is not the case for previously known variance-reduced Byzantine-robust methods such as Byrd-SAGA, BR-CSAGA, and BROADCAST that in the homogeneous data scenario have worse rates than single-machine SAGA.

$\diamond$ **Support of the compression without strong assumptions.** As we point out in Tbl. 2, the analysis of BR-CSGD and BR-CSAGA relies on the bounded second-moment assumption, which contradicts strong convexity, and the rates for BROADCAST are derived under the assumption that the compression operator almost coincides with the identity operator, meaning that in practice workers essentially do not use any compression. In contrast, our analysis does not have such substantial limitations.

$\diamond$ **Enabling non-uniform sampling.** In contrast to the existing works on Byzantine-robustness, our analysis supports non-uniform sampling of stochastic gradients. Considering the dependencies on smoothness constants, one can quickly notice our rates' even more significant superiority compared to the previous SOTA results.

## 2 Byz-VR-MARINA: BYZANTINE-TOLERANT VARIANCE REDUCTION WITH COMMUNICATION COMPRESSION

We start by introducing necessary definitions and assumptions.

**Robust aggregation.** One of the main building blocks of our method relies on the notion of $(\delta, c)$-*Robust Aggregator* introduced in (Karimireddy et al., 2021; 2022).

**Definition 2.1** ($(\delta, c)$-Robust Aggregator). *Assume that $\{x_1, x_2, \ldots, x_n\}$ is such that there exists a subset $\mathcal{G} \subseteq [n]$ of size $|\mathcal{G}| = G \geq (1 - \delta)n$ for $\delta < 0.5$ and there exists $\sigma \geq 0$ such that $\frac{1}{G(G-1)} \sum_{i,l \in \mathcal{G}} \mathbb{E}[\|x_i - x_l\|^2] \leq \sigma^2$ where the expectation is taken w.r.t. the randomness of $\{x_i\}_{i \in \mathcal{G}}$. We say that the quantity $\widehat{x}$ is $(\delta, c)$-Robust Aggregator ($(\delta, c)$-RAgg) and write $\widehat{x} = \mathtt{RAgg}(x_1, \ldots, x_n)$ for some $c > 0$, if the following inequality holds:*

$$\mathbb{E}\left[\|\widehat{x} - \overline{x}\|^2\right] \leq c\delta\sigma^2, \tag{2}$$

*where $\overline{x} = \frac{1}{|\mathcal{G}|} \sum_{i \in \mathcal{G}} x_i$. If additionally $\widehat{x}$ is computed without the knowledge of $\sigma^2$, we say that $\widehat{x}$ is $(\delta, c)$-Agnostic Robust Aggregator ($(\delta, c)$-ARAgg) and write $\widehat{x} = \mathtt{ARAgg}(x_1, \ldots, x_n)$.*

In fact, Karimireddy et al. (2021; 2022) propose slightly different definition, where they assume that $\mathbb{E}\|x_i - x_l\|^2 \leq \sigma^2$ for all fixed good workers $i, l \in \mathcal{G}$, which is marginally stronger than what we assume. Karimireddy et al. (2021) prove tightness of their definition, i.e., up to the constant $c$ one cannot improve bound (2), and prove that popular "middle-seekers" such as Krum (Blanchard et al., 2017), Robust Federated Averaging (RFA) (Pillutla et al., 2022), and Coordinate-wise Median (CM) (Chen et al., 2017) do not satisfy their definition. However, there is a trick called *bucketing* (Karimireddy et al., 2022) that provably robustifies Krum/RFA/CM. Nevertheless, the difference between our definition and the original one from (Karimireddy et al., 2021; 2022) is very subtle and it turns out that Krum/RFA/CM with bucketing fit Definition 2.1 as well (see Appendix D).

**Compression.** We consider unbiased compression operators, i.e., *quantizations*.

**Definition 2.2** (Unbiased compression (Horváth et al., 2019b)). *Stochastic mapping $\mathcal{Q} : \mathbb{R}^d \to \mathbb{R}^d$ is called unbiased compressor/compression operator if there exists $\omega \geq 0$ such that for any $x \in \mathbb{R}^d$*

$$\mathbb{E}\left[\mathcal{Q}(x)\right] = x, \quad \mathbb{E}\left[\|\mathcal{Q}(x) - x\|^2\right] \leq \omega\|x\|^2. \tag{3}$$

*For the given unbiased compressor $\mathcal{Q}(x)$, one can define the expected density as $\zeta_{\mathcal{Q}} = \sup_{x \in \mathbb{R}^d} \mathbb{E}\left[\|\mathcal{Q}(x)\|_0\right]$, where $\|y\|_0$ is the number of non-zero components of $y \in \mathbb{R}^d$.*

The above definition covers many popular compression operators such as RandK sparsification (Stich et al., 2018), random dithering (Goodall, 1951; Roberts, 1962), and natural compression (Horváth et al., 2019a) (see also the summary of various compression operators in (Beznosikov et al., 2020)). There exist also other classes of compression operators such as $\delta$-contractive compressors (Stich et al., 2018) and absolute compressors (Tang et al., 2019; Sahu et al., 2021). However, these types of compressors are out of the scope of this work.

**Assumptions.** The first assumption is quite standard in the literature on non-convex optimization.

**Assumption 2.1.** *We assume that function $f : \mathbb{R}^d \to \mathbb{R}$ is $L$-smooth, i.e., for all $x, y \in \mathbb{R}^d$ we have $\|\nabla f(x) - \nabla f(y)\| \leq L\|x - y\|$. Moreover, we assume that $f$ is uniformly lower bounded by $f_* \in \mathbb{R}$, i.e., $f_* = \inf_{x \in \mathbb{R}^d} f(x)$.*

Next, we need to restrict the data heterogeneity of regular workers. Indeed, in arbitrarily heterogeneous scenario, it is impossible to distinguish regular workers and Byzantine workers. Therefore, we use a quite standard assumption about the heterogeneity of the local loss functions.

**Assumption 2.2** ($\zeta^2$-heterogeneity). *We assume that good clients have $\zeta^2$-heterogeneous local loss functions for some $\zeta \geq 0$, i.e.,*

$$\frac{1}{G} \sum_{i \in \mathcal{G}} \|\nabla f_i(x) - \nabla f(x)\|^2 \leq \zeta^2 \quad \forall x \in \mathbb{R}^d. \tag{4}$$

We emphasize here that the homogeneous data case ($\zeta = 0$) is realistic in collaborative learning. This typically means that the workers have an access to the entire data. For example, this can be implemented using so-called *dataset streaming* when the data is received just in time in chunks (Diskin et al., 2021; Kijsipongse et al., 2018) (this can also be implemented without using the server via special protocols similar to BitTorrent).

The following assumption is a refinement of a standard assumption that $f_i$ is $L_i$-smooth for all $i \in \mathcal{G}$.

**Assumption 2.3** (Global Hessian variance assumption (Szlendak et al., 2021)). *We assume that there exists $L_\pm \geq 0$ such that for all $x, y \in \mathbb{R}^d$*

$$\frac{1}{G} \sum_{i \in \mathcal{G}} \|\nabla f_i(x) - \nabla f_i(y)\|^2 - \|\nabla f(x) - \nabla f(y)\|^2 \leq L_\pm^2 \|x - y\|^2. \tag{5}$$

If $f_i$ is $L_i$-smooth for all $i \in \mathcal{G}$, then the above assumption is always valid for some $L_\pm \geq 0$ such that $L_{\text{avg}}^2 - L^2 \leq L_\pm^2 \leq L_{\text{avg}}^2$, where $L_{\text{avg}}^2 = \frac{1}{G} \sum_{i \in \mathcal{G}} L_i^2$ (Szlendak et al., 2021). Moreover, Szlendak et al. (2021) show that there exist problems with heterogeneous functions on workers such that (5) holds with $L_\pm = 0$, while $L_{\text{avg}} > 0$.

We propose a generalization of the above assumption for samplings of stochastic gradients.

**Assumption 2.4** (Local Hessian variance assumption). *We assume that there exists $\mathcal{L}_\pm \geq 0$ such that for all $x, y \in \mathbb{R}^d$*

$$\frac{1}{G} \sum_{i \in \mathcal{G}} \mathbb{E} \|\widehat{\Delta}_i(x, y) - \Delta_i(x, y)\|^2 \leq \frac{\mathcal{L}_\pm^2}{b} \|x - y\|^2, \tag{6}$$

*where $\Delta_i(x, y) = \nabla f_i(x) - \nabla f_i(y)$ and $\widehat{\Delta}_i(x, y)$ is an unbiased mini-batched estimator of $\Delta_i(x, y)$ with batch size $b$.*

We notice that the above assumption covers a wide range of samplings of mini-batched stochastic gradient differences, e.g., standard uniform sampling or importance sampling. We provide the examples in Appendix E.1. We notice that all previous works on Byzantine-robustness focus on the standard uniform sampling only. However, uniform sampling can give $m$ times worse constant $\mathcal{L}_\pm^2$ than importance sampling. This difference significantly affect the complexity bounds.

**New Method: Byz-VR-MARINA.** Now we are ready to present our new method—Byzantine-tolerant Variance-Reduced MARINA (Byz-VR-MARINA). Our algorithm is based on the recently proposed variance-reduced method with compression (VR-MARINA) from (Gorbunov et al., 2021b). At each iteration of Byz-VR-MARINA, good workers update their parameters $x^{k+1} = x^k - \gamma g^k$ using estimator $g^k$ received from the parameter-server (line 7). Next (line 8), with (typically small) probability

$p$ each good worker $i \in \mathcal{G}$ computes its full gradient, and with (typically large) probability $1 - p$ this worker computes compressed mini-batched stochastic gradient difference $\mathcal{Q}(\widehat{\Delta}_i(x^{k+1}, x^k))$, where $\widehat{\Delta}_i(x^{k+1}, x^k)$ satisfies Assumption 2.4. After that, the server gathers the results of computations from the workers and applies $(\delta, c)$-ARAgg to compute the next estimator $g^{k+1}$ (line 10).

Let us elaborate on several important parts of the proposed algorithm. First, we point out that with large probability $1 - p$ good workers need to send just compressed vectors $\mathcal{Q}(\widehat{\Delta}_i(x^{k+1}, x^k))$, $i \in \mathcal{G}$. Indeed, since the server knows when workers compute full gradients and when they compute compressed stochastic gradients, it needs just to add $g^k$ to all received vectors to perform robust aggregation from line 10. Moreover, since the server knows the type of compression operator that good workers apply, it can typically easily filter out those Byzantine workers who try to slow down the training via sending dense vectors instead of compressed ones (e.g., if the compression operator is RandK sparsification, then Byzantine workers cannot send more than $K$ components; otherwise they will be easily detected and can be banned). Next, the right choice of probability $p$ allows equalizing the communication cost of all steps when good workers send dense gradients and compressed gradient differences. The same is true for oracle complexity: if $p \leq {}^b\!/\!_m$, then the computational cost of full-batch computations is not bigger than that of stochastic gradients.

---

**Algorithm 1** Byz-VR-MARINA: Byzantine-tolerant VR-MARINA

1: **Input:** starting point $x^0$, stepsize $\gamma$, minibatch size $b$, probability $p \in (0, 1]$, number of iterations $K$, $(\delta, c)$-ARAgg
2: Initialize $g^0 = \nabla f(x^0)$
3: **for** $k = 0, 1, \ldots, K - 1$ **do**
4:     Get a sample from Bernoulli distribution with parameter $p$: $c_k \sim \mathrm{Be}(p)$
5:     Broadcast $g^k$, $c_k$ to all workers
6:     **for** $i \in \mathcal{G}$ in parallel **do**
7:         $x^{k+1} = x^k - \gamma g^k$
8:         Set $g_i^{k+1} = \begin{cases} \nabla f_i(x^{k+1}), & \text{if } c_k = 1, \\ g^k + \mathcal{Q}\left(\widehat{\Delta}_i(x^{k+1}, x^k)\right), & \text{otherwise,} \end{cases}$ where minibatched estimator

        $\widehat{\Delta}_i(x^{k+1}, x^k)$ of $\nabla f_i(x^{k+1}) - \nabla f_i(x^k)$; $\mathcal{Q}(\cdot)$ for $i \in \mathcal{G}$ are computed independently
9:     **end for**
10:     $g^{k+1} = \mathrm{ARAgg}(g_1^{k+1}, \ldots, g_n^{k+1})$
11: **end for**
12: **Return:** $\hat{x}^K$ chosen uniformly at random from $\{x^k\}_{k=0}^{K-1}$

---

**Challenges in designing variance-reduced algorithm with tight rates and provable Byzantine-robustness.** In the introduction, we explain why variance reduction is a natural way to handle Byzantine attacks (see the discussion before **Q1**). At first glance, it seems that one can take any variance-reduced method and combine it with some robust aggregation rule to get the result. However, this is not as straightforward as it may appear. As one can see from Table 2, combination of SAGA with geometric median estimator (Byrd-SAGA) gives the rate $\widetilde{\mathcal{O}}(\frac{m^2}{b^2(1-2\delta)\mu^2})$ (smoothness constant and logarithmic factors are omitted) in the smooth strongly convex case — this rate is in fact $\mathcal{O}(\frac{m^2}{b^2\mu^2})$ times worse than the rate of SAGA even when $\delta = 0$. Therefore, it becomes clear that the full potential of variance reduction in Byzantine-robust learning is not revealed via Byrd-SAGA.

The key reason for that is the sensitivity of SAGA (and SAGA-based methods) to the unbiasedness of the stochastic estimator in the analysis. Since Byrd-SAGA uses the geometric median for the aggregation, which is necessarily biased, it is natural that it has a much worse convergence rate than SAGA even in the $\delta = 0$ case. Moreover, one can't solve such an issue by simply changing one robust estimator for another since all known robust estimators are generally biased.

To circumvent this issue, we consider Geom-SARAH/PAGE-based estimator (Horváth & Richtárik, 2019; Li et al., 2021) and study how it interacts with the robust aggregation. In particular, we observe that the averaged pair-wise variance for the stochastic gradients of good workers could be upper bounded by a constant multiplied by $\mathbb{E}\|x^{k+1} - x^k\|^2$ plus some additional terms appearing due to heterogeneity (see Lemma E.2). Then, we notice that the robust aggregation only leads to the additional term proportional to $\mathbb{E}\|x^{k+1} - x^k\|^2$ (plus additional terms due to heterogeneity). We

show that this term can be directly controlled using another term proportional to $-\mathbb{E}\|x^{k+1} - x^k\|^2$, which appears in the original analysis of PAGE/VR-MARINA.

These facts imply that although the difference between Byz-VR-MARINA and VR-MARINA is only in the choice of the aggregation rule, it is not straightforward beforehand that such a combination should be considered and that it will lead to better rates. Moreover, as we show next, we obtain vast improvements upon the previously best-known theoretical results for Byzantine-tolerant learning.

**General Non-Convex Functions.** Our main convergence result for general non-convex functions follows. All proofs are deferred to Appendix E.

**Theorem 2.1.** *Let Assumptions 2.1, 2.2, 2.3, 2.4 hold. Assume that $0 < \gamma \leq \frac{1}{L+\sqrt{A}}$, where*
$A = \frac{6(1-p)}{p}\left(\frac{4c\delta}{p} + \frac{1}{2G}\right)\left(\omega L^2 + \frac{(1+\omega)\mathcal{L}_{\pm}^2}{b}\right) + \frac{6(1-p)}{p}\left(\frac{4c\delta(1+\omega)}{p} + \frac{\omega}{2G}\right)L_{\pm}^2$. *Then for all $K \geq 0$ the point $\widehat{x}^K$ chosen uniformly at random from the iterates $x^0, x^1, \ldots, x^K$ produced by* Byz-VR-MARINA *satisfies*

$$\mathbb{E}\left[\|\nabla f(\widehat{x}^K)\|^2\right] \leq \frac{2\Phi_0}{\gamma(K+1)} + \frac{24c\delta\zeta^2}{p}, \tag{7}$$

*where $\Phi_0 = f(x^0) - f_* + \frac{\gamma}{p}\|g^0 - \nabla f(x^0)\|^2$ and $\mathbb{E}[\cdot]$ denotes the full expectation.[2]*

We highlight here several important properties of the derived result. First of all, this is the first theoretical result for the convergence of Byzantine-tolerant methods with compression in the non-convex case with arbitrary adversaries. Next, when $\zeta > 0$ the theorem above does not guarantee that $\mathbb{E}[\|\nabla f(\widehat{x}^K)\|^2]$ can be made arbitrarily small. However, this is not a drawback of our analysis but rather an inevitable limitation of all algorithms in heterogeneous case. This is due to Karimireddy et al. (2022) who proved a lower bound showing that in the presence of Byzantine workers, all algorithms satisfy $\mathbb{E}[\|\nabla f(\widehat{x}^K)\|^2] = \Omega(\delta\zeta^2)$, i.e., the constant term from (7) is tight up to the factor of $1/p$. However, when $\zeta = 0$, Byz-VR-MARINA can achieve any predefined accuracy of the solution, if $\delta$ is such that ARAgg is $(\delta, c)$-robust (see Theorem D.1). Finally, as Table 2 shows[3], Byz-VR-MARINA achieves $\mathbb{E}[\|\nabla f(\widehat{x}^K)\|^2] \leq \varepsilon^2$ faster than all previously known Byzantine-tolerant methods for small enough $\varepsilon$. Moreover, unlike virtually all other results in the non-convex case, Theorem 2.1 does not rely on the uniformly bounded variance assumption, which is known to be very restrictive (Nguyen et al., 2018). For further discussion we refer to Appendix E.5.

**Functions Satisfying Polyak-Łojasiewicz (PŁ) Condition.** We extend our theory to the functions satisfying *Polyak-Łojasiewicz condition* (Polyak, 1963; Łojasiewicz, 1963). This assumption generalizes regular strong convexity and holds for several non-convex problems (Karimi et al., 2016). Moreover, a very similar assumption appears in over-parameterized deep learning (Liu et al., 2022).

**Assumption 2.5** (PŁ condition). *We assume that function $f$ satisfies Polyak-Łojasiewicz (PŁ) condition with parameter $\mu$, i.e., for all $x \in \mathbb{R}^d$ there exists $x^* \in \arg\min_{x\in\mathbb{R}^d} f(x)$ such that*

$$\|\nabla f(x)\|^2 \geq 2\mu\left(f(x) - f(x^*)\right). \tag{8}$$

Under this and previously introduced assumptions, we derive the following result.

**Theorem 2.2.** *Let Assumptions 2.1, 2.2, 2.3, 2.4, 2.5 hold. Assume that $0 < \gamma \leq \min\left\{\frac{1}{L+\sqrt{2A}}, \frac{p}{4\mu}\right\}$, where $A = \frac{6(1-p)}{p}\left(\frac{4c\delta}{p} + \frac{1}{2G}\right)\left(\omega L^2 + \frac{(1+\omega)\mathcal{L}_{\pm}^2}{b}\right) + \frac{6(1-p)}{p}\left(\frac{4c\delta(1+\omega)}{p} + \frac{\omega}{2G}\right)L_{\pm}^2$. Then for all $K \geq 0$ the iterates produced by* Byz-VR-MARINA *satisfy*

$$\mathbb{E}\left[f(x^K) - f(x^*)\right] \leq (1 - \gamma\mu)^K \Phi_0 + \frac{24c\delta\zeta^2}{\mu}, \tag{9}$$

*where $\Phi_0 = f(x^0) - f_* + \frac{2\gamma}{p}\|g^0 - \nabla f(x^0)\|^2$.*

---

[2]In all the results and the proofs, $\mathbb{E}[\cdot]$ denotes the full expectation if the opposite is not specified.

[3]To have a fair comparison, we take $p = \min\{b/m, 1/(1+\omega)\}$ since in this case, at each iteration each worker sends $\mathcal{O}(\zeta_Q)$ components, when $\omega + 1 = \Theta(d/\zeta_Q)$ (which is the case for RandK sparsification and $\ell_2$-quantization, see (Beznosikov et al., 2020)), and makes $\mathcal{O}(b)$ oracle calls in expectation (computations of $\nabla f_{i,j}(x)$). With such choice of $p$, the total expected (communication and oracle) cost of steps with full gradients computations/uncompressed communications coincides with the total cost of the rest of iterations.

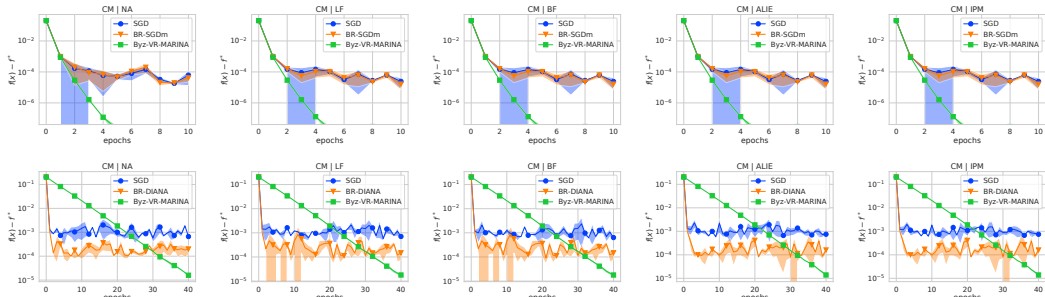

Figure 1: The optimality gap $f(x^k) - f(x^*)$ of 3 aggregation rules (AVG, CM, RFA) under 5 attacks (NA, LF, BF, ALIE, IPM) on a9a dataset, where each worker has access full dataset with 4 good and 1 Byzantine worker. In the first row, we do not use any compression, in the second row each method uses Rand$K$ sparsification with $K = 0.1d$.

Similarly to the general non-convex case, in the PŁ-setting Byz-VR-MARINA is able to achieve $\mathbb{E}[f(x^K) - f(x^*)] = \mathcal{O}(c\delta\zeta^2/\mu)$ accuracy, which matches (up to the factor of $1/p$) the lower bound from Karimireddy et al. (2022) derived for $\mu$-*strongly convex* objectives. Next, when $\zeta = 0$, Byz-VR-MARINA converges linearly asymptotically to the exact solution. Moreover, as Table 2 shows, our convergence result in the PŁ-setting outperforms the known rates in more restrictive strongly-convex setting. In particular, when $\varepsilon$ is small enough, Byz-VR-MARINA has better complexity than BTARD-SGD. When the conditioning of the problem is bad (i.e., $L/\mu \gg 1$) our rate dominates results of BR-CSGD, BR-CSAGA, and BROADCAST. Furthermore, both BR-CSGD and BR-CSAGA rely on the uniformly bounded second moment assumption (contradicting the strong convexity), and the rate of the BROADCAST algorithm is based on the assumption that $\omega = \mathcal{O}(\mu^2/L^2)$ implying that $\mathcal{Q}(x) \approx x$ (no compression) even for well-conditioned problems.

## 3 NUMERICAL EXPERIMENTS

In this section, we demonstrate the practical performance of the proposed method. The main goal of our experimental evaluation is to showcase the benefits of employing SOTA variance reduction to remedy the presence of Byzantine workers. For the task, we consider the standard logistic regression model with $\ell_2$-regularization $f_{i,j}(x) = -y_{i,j}\log(h(x, a_{i,j})) - (1-y_{i,j})\log(1-h(x, a_{i,j})) + \lambda\|x\|^2$, where $y_{i,j} \in \{0,1\}$ is the label, $a_{i,j} \in \mathbb{R}^d$ represents the features vector, $\lambda$ is the regularization parameter and $h(x, a) = 1/(1+e^{-a^\top x})$. One can show that this objective is smooth, and for $\lambda > 0$, it is also strongly convex, therefore, it satisfies PŁcondition. We consider *a9a* LIBSVM dataset (Chang & Lin, 2011) and set $\lambda = 0.01$. In the experiments, we focus on an important feature of Byz-VR-MARINA: it guarantees linear convergence for homogeneous datasets across clients even in the presence of Byzantine workers, as shown in Theorem 2.2. To demonstrate this experimentally, we consider the setup with four good workers and one Byzantine worker, *each worker can access the entire dataset*, and the server uses coordinate-wise median with bucketing as the aggregator (see the details in Appendix D). We consider five different attacks: ● **No Attack (NA):** clean training; ● **Label Flipping (LF):** labels are flipped, i.e., $y_{i,j} \rightarrow 1 - y_{i,j}$; ● **Bit Flipping (BF):** a Byzantine worker sends an update with flipped sign; ● **A Little is enough (ALIE)** (Baruch et al., 2019): the Byzantine workers estimate the mean $\mu_{\mathcal{G}}$ and standard deviation $\sigma_{\mathcal{G}}$ of the good updates, and send $\mu_{\mathcal{G}} - z\sigma_{\mathcal{G}}$ to the server where $z$ is a small constant controlling the strength of the attack; ● **Inner Product Manipulation (IPM)** (Xie et al., 2020): the attackers send $-\frac{\epsilon}{G}\sum_{i\in\mathcal{G}}\nabla f_i(x)$ where $\epsilon$ controls the strength of the attack. For bucketing, we use $s = 2$, i.e., partitioning the updates into the groups of two, as recommended by Karimireddy et al. (2022). We compare our Byz-VR-MARINA with the baselines without compression (SGD, BR-SGDm (Karimireddy et al., 2021)) and the baselines with random sparsification (compressed SGD and DIANA (BR-DIANA). We do not compare against Byrd-SAGA (and BR-CSAGA, BROADCAST from Zhu & Ling (2021)), which consumes large memory that scales linearly with the number of local data points and is not well suited for memory-efficient batched gradient computation (e.g., used in PyTorch). Our implementation is based on PyTorch (Paszke et al., 2019). Figure 1 showcases that, indeed, we observe linear convergence of our method while no baseline achieves this fast rate. In the first row, we display methods with no compression, and in the second row, each algorithm uses random sparsification. We defer further details and additional experiments with heterogeneous data to Appendix B.

## ACKNOWLEDGEMENTS

We thank anonymous reviewers for useful suggestions regarding additional experiments and discussion of the derived results. The work of E. Gorbunov was partially supported by a grant for research centers in the field of artificial intelligence, provided by the Analytical Center for the Government of the Russian Federation in accordance with the subsidy agreement (agreement identifier 000000D730321P5Q0002) and the agreement with the Moscow Institute of Physics and Technology dated November 1, 2021 No. 70-2021-00138. The work of P. Richtárik was partially supported by the KAUST Baseline Research Fund Scheme and by the SDAIA-KAUST Center of Excellence in Data Science and Artificial Intelligence.

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

CONTENTS

# A  DETAILED RELATED WORK

**Byzantine-robustness.** Classical approaches to Byzantine-tolerant optimization are based on applying special aggregation rules to Parallel-SGD (Blanchard et al., 2017; Chen et al., 2017; Yin et al., 2018; Damaskinos et al., 2019; Guerraoui et al., 2018; Pillutla et al., 2022). It turns out that such defences are vulnerable to the special type of attacks (Baruch et al., 2019; Xie et al., 2020). Moreover, Karimireddy et al. (2021) propose a reasonable formalism for describing robust aggregation rules (see Def. 2.1) and show that almost all previously known defences are not robust according to this formalism. In addition, they propose and analyze new Byzantine-tolerant methods based on the usage of Polyak's momentum (Polyak, 1964) (BR-SDGm) and momentum variance reduction (Cutkosky & Orabona, 2019) (BR-MVR). This approach is extended to the case of heterogeneous data and aggregators agnostic to the noise level by Karimireddy et al. (2022), and He et al. (2022) propose an extension to the decentralized optimization over fixed networks. Gorbunov et al. (2021a) propose an alternative approach based on the usage of AllReduce (Patarasuk & Yuan, 2009) with additional verifications of correctness and show that their algorithm has complexity not worse than Parallel-SGD when the target accuracy is small enough. Wu et al. (2020) are the first who applied variance reduction mechanism to tolerate Byzantine attacks (see the discussion above **Q1**). We also refer reader to (Chen et al., 2018; Rajput et al., 2019; Rodríguez-Barroso et al., 2020; Xu & Lyu, 2020; Alistarh et al., 2018; Allen-Zhu et al., 2021; Regatti et al., 2020; Yang & Bajwa, 2019a;b; Gupta et al., 2021; Gupta & Vaidya, 2021; Peng et al., 2021) for other advances in Byzantine-robustness (see the detailed summaries in (Lyu et al., 2020; Gorbunov et al., 2021a)). We further progress the field by obtaining new theoretical SOTA convergence results in our work.

**Compressed communications.** Methods with compression are relatively well studied in the literature. The first theoretical results were derived in (Alistarh et al., 2017; Wen et al., 2017; Stich et al., 2018; Mishchenko et al., 2019). During the last several years the field has been significantly developed. In particular, compressed methods are analyzed in the conjuction with variance reduction (Horváth et al., 2019b; Gorbunov et al., 2020b; Danilova & Gorbunov, 2022), acceleration (Li et al., 2020; Li & Richtárik, 2021; Qian et al., 2021b), decentralized communications (Koloskova et al., 2019; Kovalev et al., 2021), local steps (Basu et al., 2019; Haddadpour et al., 2021), adaptive compression (Faghri et al., 2020), second-order methods (Islamov et al., 2021; Safaryan et al., 2021), and min-max optimization (Beznosikov et al., 2021; 2022). However, to our knowledge, only one work studies communication compression in the context of Byzantine-robustness (Zhu & Ling, 2021) (see the discussion above **Q2**). Our work makes a further step towards closing this significant gap in the literature.

**Variance reduction** is a powerful tool allowing to speed up the convergence of stochastic methods (especially when one needs to achieve a good approximation of the solution). The first variance-reduced methods were proposed by Schmidt et al. (2017); Johnson & Zhang (2013); Defazio et al. (2014). Optimal variance-reduced methods for (strongly) convex problems are proposed in (Lan & Zhou, 2018; Allen-Zhu, 2017; Lan et al., 2019) and for non-convex optimization in (Nguyen et al., 2017; Fang et al., 2018; Li et al., 2021). Despite the noticeable attention to these kinds of methods (Gower et al., 2020), only a few papers study Byzantine-robustness in conjunction with variance reduction (Wu et al., 2020; Zhu & Ling, 2021; Karimireddy et al., 2021). Moreover, as we mentioned before, the results from Wu et al. (2020); Zhu & Ling (2021) are not better than the known ones for non-parallel variance-reduced methods, and Karimireddy et al. (2021) rely on the uniformly bounded variance assumption, which is hard to achieve in practice. In our work, we circumvent these limitations.

**Non-uniform sampling.** Originally proposed for randomized coordinate methods (Nesterov, 2012; Richtárik & Takáč, 2016; Qu & Richtárik, 2016), non-uniform sampling is extended in multiple ways to stochastic optimization, e.g., see (Horváth & Richtárik, 2019; Gower et al., 2019; Qian et al., 2019; Gorbunov et al., 2020b;a; Qian et al., 2021a). Typically, non-uniform sampling of stochastic gradients allows better dependence on smoothness constants in the theoretical results. Inspired by these advances, we propose the first Byzantine-robust optimization method supporting non-uniform sampling of stochastic gradients.

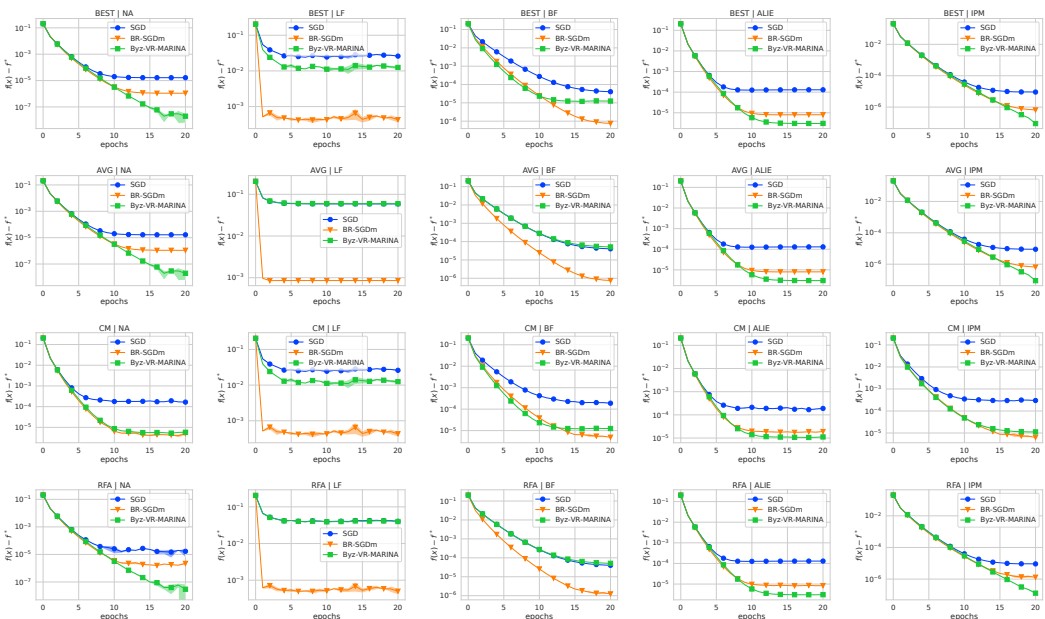

Figure 2: The optimality gap $f(x^k) - f(x^*)$ of 3 aggregation rules (AVG, CM, RFA) under 5 attacks (NA, LF, BF, ALIE, IPM) on a9a dataset with uniform split over 15 workers with 5 Byzantine workers. The top row displays the best performance in hindsight for a given attack.

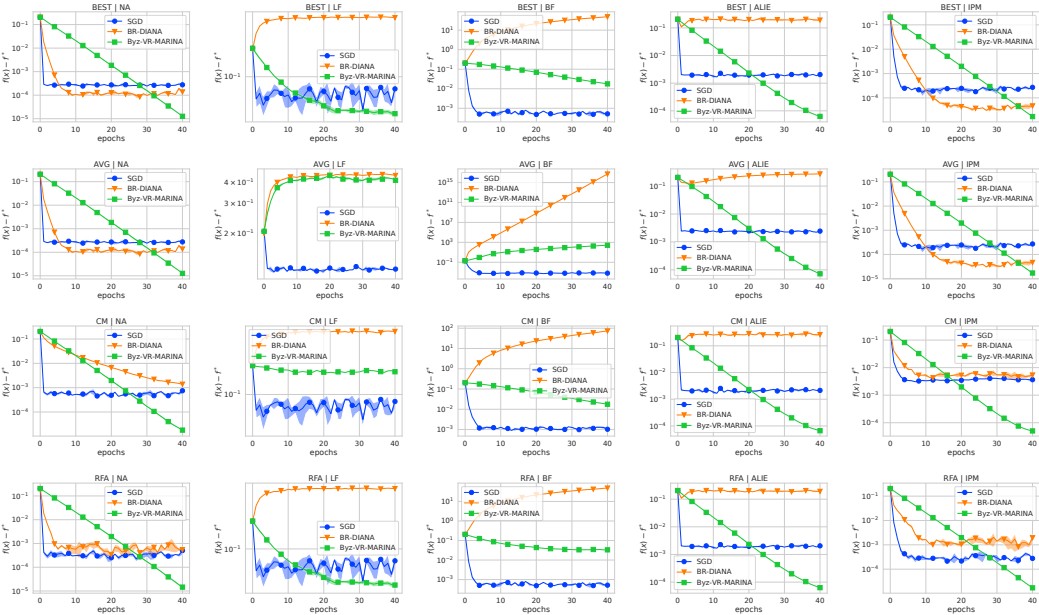

Figure 3: The optimality gap $f(x^k) - f(x^*)$ of 3 aggregation rules (AVG, CM, RFA) under 5 attacks (NA, LF, BF, ALIE, IPM) on a9a dataset with uniform split over 15 workers with 5 Byzantine workers. The top row displays the best performance in hindsight for a given attack. Each method uses Rand$K$ sparsification with $K = 0.1d$.

## B  EXTRA EXPERIMENTS AND EXPERIMENTAL DETAILS

### B.1  GENERAL SETUP

Our running environment has the following setup:

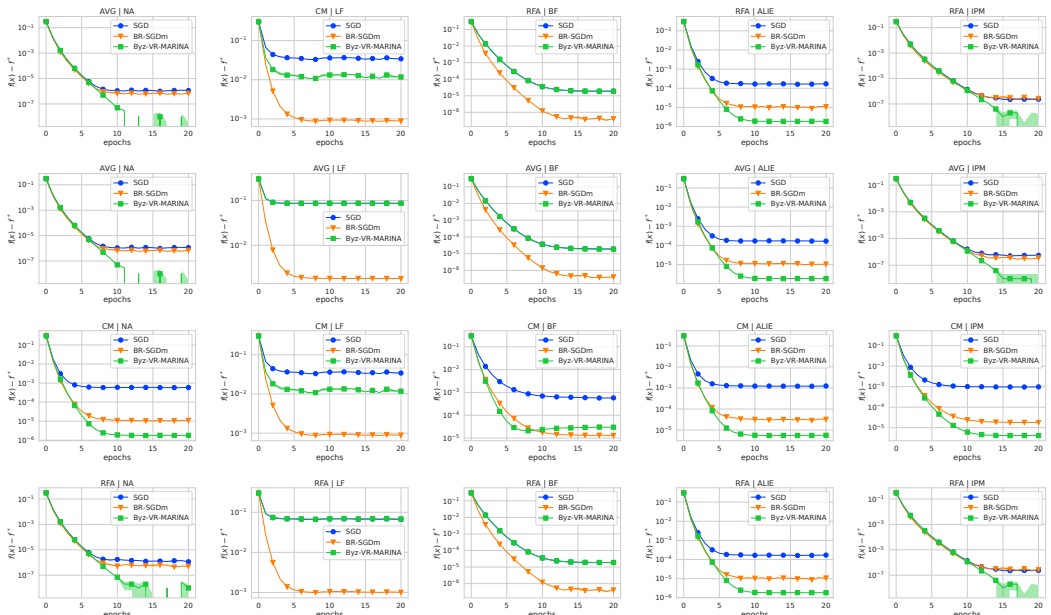

Figure 4: The optimality gap $f(x^k) - f(x^*)$ of 3 aggregation rules (AVG, CM, RFA) under 5 attacks (NA, LF, BF, ALIE, IPM) on w8a dataset with uniform split over 15 workers with 5 Byzantine workers. The top row displays the best performance in hindsight for a given attack. No compression is applied.

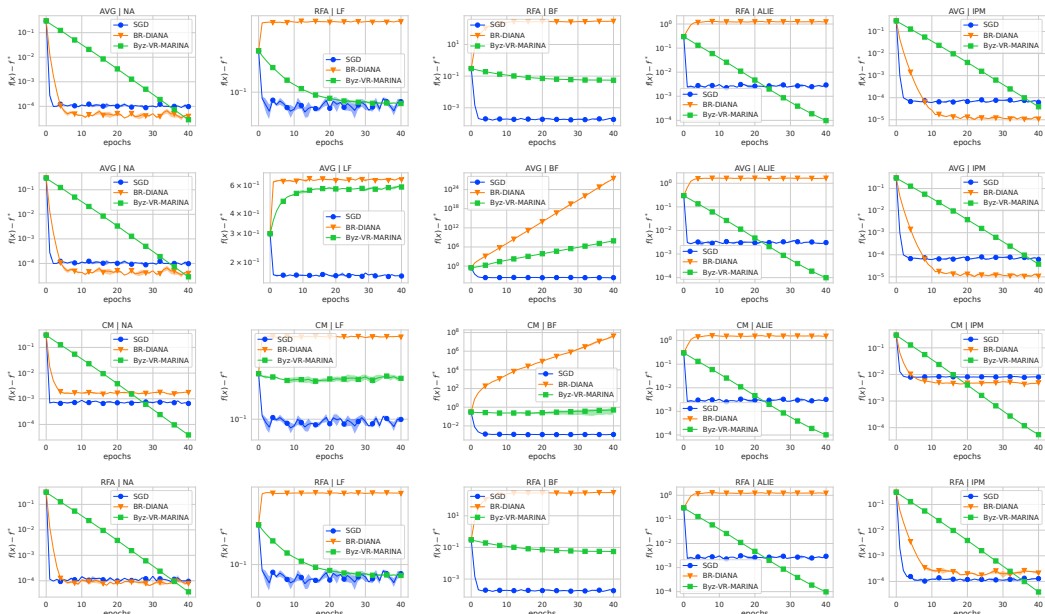

Figure 5: The optimality gap $f(x^k) - f(x^*)$ of 3 aggregation rules (AVG, CM, RFA) under 5 attacks (NA, LF, BF, ALIE, IPM) on w8a dataset with uniform split over 15 workers with 5 Byzantine workers. The top row displays the best performance in hindsight for a given attack. Each method uses Rand$K$ sparsification with $K = 0.1d$.

- 24 CPUs: Intel(R) Xeon(R) Gold 6146 CPU @ 3.20GHz ,
- GPU: NVIDIA TITAN Xp with CUDA version 11.3,
- PyTorch version: 1.11.0.

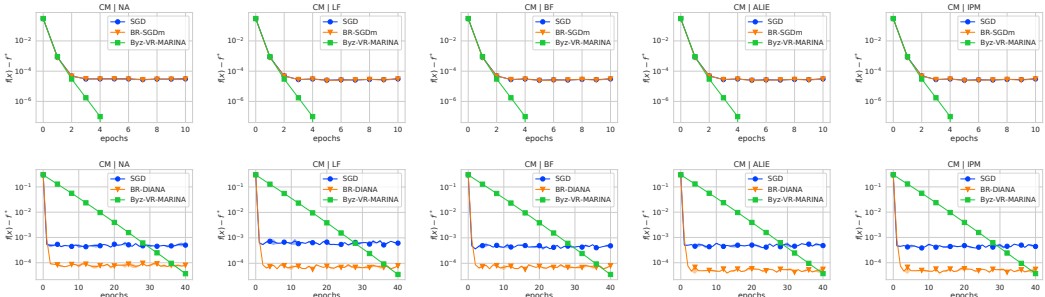

Figure 6: The optimality gap $f(x^k) - f(x^*)$ of 3 aggregation rules (AVG, CM, RFA) under 5 attacks (NA, LF, BF, ALIE, IPM) on w8a dataset, where each worker has access full dataset with 4 good and 1 Byzantine worker. In the first row, we do not use any compression, in the second row each method uses Rand$K$ sparsification with $K = 0.1d$.

## B.2 EXPERIMENTAL SETUP

For each experiment, we tune the step size using the following set of candidates $\{0.5, 0.05, 0.005\}$. The step size is fixed. We do not use learning rate warmup or decay. We use batches of size 32 for all methods. Each experiment is run with three varying random seeds, and we report the mean optimality gap with one standard error. The optimal value is obtained by running gradient descent (GD) on the complete dataset for 1000 epochs. Our implementation of attacks and robust aggregation schemes is based on the public implementation from (Karimireddy et al., 2022) available at https://github.com/epfml/byzantine-robust-noniid-optimizer. Our codes are available via an anonymized repository at https://github.com/SamuelHorvath/VR_Byzantine. We select the same set of hyperparameters as (Karimireddy et al., 2022), i.e.,

- RFA: the number of steps of smoothed Weisfield algorithm $T = 8$; see Section D for details,
- ALIE: a small constant that controls the strength of the attack $z$ is chosen according to (Baruch et al., 2019),
- IPM: a small constant that controls the strength of the attack $\epsilon = 0.1$.

## B.3 EXTRA EXPERIMENTS

### B.3.1 HETEROGENEOUS DATA

In this case, we randomly shuffle dataset and we sequentially distribute it among 15 good workers, where each worker has approximately the same amount of data and there is no overlap. We include five Byzantine workers who have access to an entire dataset and the exact updates computed at each client. For the aggregation, we consider three rules: *standard averaging* (AVG), *coordinate-wise median* (CM) with *bucketing*, and *robust federated averaging* (RFA) with *bucketing* (see the details in Appendix D).

**Discussion.** In Figure 2, we can see that momentum (BR-SGDm) the variance reduction (Byz-VR-MARINA) techniques consistently outperform the SGD baseline while none of them dominates for all the attacks. Byz-VR-MARINA is particularly useful in the clean data regime and against the ALIE and IPM attacks, and the BR-SGDm algorithm provides the best performance for label and bit flipping attacks. It would be interesting to automatically select the best technique, e.g., momentum or VR-MARINA, that provides the best defense against any given attack. We leave this for future work.

### B.3.2 COMPRESSION

In this section, we consider the same setup as for the previous experiment with a difference that we employ communication compression. We choose random unbiased sparsification for with sparsity level $10\%$. We compare our Byz-VR-MARINA algorithm to compressed SGD and DIANA (BR-DIANA).

**Discussion.** In Figure 3, we can see that Byz-VR-MARINA consistently outperforms both baselines except for the bit flipping attack. However, even in this case, it seems that Byz-VR-MARINA only

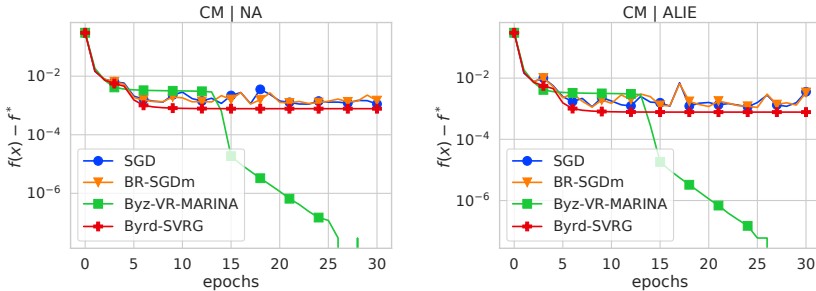

Figure 7: The optimality gap $f(x^k) - f(x^*)$ under 2 attacks (NA, ALIE) on a9a dataset, where each worker has access full dataset with 4 good and 1 Byzantine worker. No compression is applied.

needs more epochs to provide the better solution while SGD cannot further improve regardless of the number of epochs.

### B.3.3 EXTRA DATASET: W8A

In Figures 4–6, we perform the same experiments, but for the different LIBSVM dataset:*w8a*. We note that the obtained results are consistent with our observations for the *a9a* dataset.

### B.4 COMPARISON WITH Byrd-SVRG

As we note in the main part of the paper, Byrd-SAGA is not well suited for PyTorch due to the large memory consumption of SAGA-based methods. Nevertheless, one can use SVRG-estimator (Johnson & Zhang, 2013) as a proxy of SAGA-estimator due to similarities between SAGA and SVRG. We call the resulting method as Byrd-SVRG and compare its performance with Byz-VR-MARINA on the logistic regression task with non-convex regularization: an instance of (1) with $f_{i,j}(x) = -y_{i,j} \log(h(x, a_{i,j})) - (1 - y_{i,j}) \log(1 - h(x, a_{i,j})) + \lambda \sum_{i=1}^{d} \frac{x_i^2}{1+x_i^2}$. The results are presented in Figure 7. One can see that Byz-VR-MARINA converges to the exact solution asymptotically, while other methods are able to converge only to some neighborhood of the solution.

### B.5 EFFECT OF COMPRESSION

In this experiment, we illustrate the effect of compression in Byz-VR-MARINA on its communication efficiency. We compare the performance of Byz-VR-MARINA with and without compression in terms of the number of communicated bits between workers and server. The results are shown in Figure 8. One can see that communication compression does speed up the training (in terms of the number of transmitted bits).

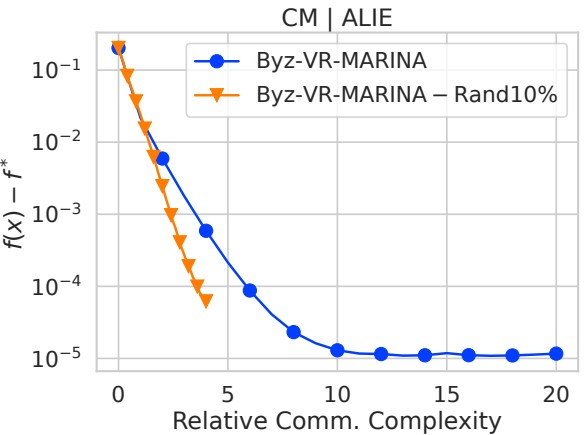

Figure 8: The optimality gap $f(x^k) - f(x^*)$ under ALIE attacks on a9a dataset, where each worker has access full dataset with 4 good and 1 Byzantine worker. The figure illustrates the effect of compression in Byz-VR-MARINA. Rand$K$ sparsification with $K = 0.1d$ is used as a compression operator. "Relative comm. compression" = number of transmitted bits divided by the number of transmitted bits per iteration for full-precision algorithm and then divided by the size of the dataset (when no compression is applied this quantity equals the number of epochs).

## C  USEFUL FACTS

For all $a, b \in \mathbb{R}^d$ and $\alpha > 0, p \in (0, 1]$ the following relations hold:

$$2\langle a, b \rangle = \|a\|^2 + \|b\|^2 - \|a - b\|^2, \tag{10}$$

$$\|a + b\|^2 \leq (1 + \alpha)\|a\|^2 + (1 + \alpha^{-1})\|b\|^2, \tag{11}$$

$$-\|a - b\|^2 \leq -\frac{1}{1 + \alpha}\|a\|^2 + \frac{1}{\alpha}\|b\|^2, \tag{12}$$

$$(1 - p)\left(1 + \frac{p}{2}\right) \leq 1 - \frac{p}{2}. \tag{13}$$

**Lemma C.1** (Lemma 5 from Richtárik et al. (2021)). *Let $a, b > 0$. If $0 \leq \gamma \leq \frac{1}{\sqrt{a} + b}$, then $a\gamma^2 + b\gamma \leq 1$. The bound is tight up to the factor of $2$ since $\frac{1}{\sqrt{a} + b} \leq \min\left\{\frac{1}{\sqrt{a}}, \frac{1}{b}\right\} \leq \frac{2}{\sqrt{a} + b}$.*

## D    FURTHER DETAILS ON ROBUST AGGREGATION

In Section 2, we consider robust aggregation rules satisfying Definition 2.1. As we notice, this definition slightly differs from the original one introduced by Karimireddy et al. (2022). In particular, we assume

$$\frac{1}{G(G-1)} \sum_{i,l \in \mathcal{G}} \mathbb{E}\left[\|x_i - x_l\|^2\right] \leq \sigma^2, \tag{14}$$

while Karimireddy et al. (2022) uses $\mathbb{E}[\|x_i - x_l\|^2] \leq \sigma^2$ for any *fixed* $i, l \in \mathcal{G}$. As we show next, this difference is very subtle and condtion (14) also allows to achieve robustness.

We consider robust aggregation via *bucketing* proposed by Karimireddy et al. (2022) (see Algorithm 2). This algorithm can robustify some non-robust aggregation rules Aggr. In particular,

---

**Algorithm 2** Bucketing: Robust Aggregation using bucketing (Karimireddy et al., 2022)

---

1: **Input:** $\{x_1, \ldots, x_n\}$, $s \in \mathbb{N}$ – bucket size, Aggr – aggregation rule
2: Sample random permutation $\pi = (\pi(1), \ldots, \pi(n))$ of $[n]$
3: Compute $y_i = \frac{1}{s} \sum_{k=s(i-1)+1}^{\min\{si,n\}} x_{\pi(k)}$ for $i = 1, \ldots, \lceil n/s \rceil$
4: **Return:** $\widehat{x} = \text{Aggr}(y_1, \ldots, y_{\lceil n/s \rceil})$

---

Karimireddy et al. (2022) show that Algorithm 2 makes Krum (Blanchard et al., 2017), Robust Federated Averaging (RFA) (Pillutla et al., 2022) (also known as geometric median), and Coordinate-wise Median (CM) (Chen et al., 2017) robust, in view of definition from Karimireddy et al. (2022).

Our main goal in this section is to show that Krum ∘ Bucketing, RFA ∘ Bucketing, and CM ∘ Bucketing satisfy Definition 2.1. Before we prove this fact, we need to introduce Krum, RFA, CM.

**Krum.**    Let $S_i \subseteq \{x_1, \ldots, x_n\}$ be the subset of $n - |\mathcal{B}| - 2$ closest vectors to $x_i$. Then, Krum-estimator is defined as

$$\text{Krum}(x_1, \ldots, x_n) \stackrel{\text{def}}{=} \underset{x_i \in \{x_1, \ldots, x_n\}}{\arg\min} \sum_{j \in S_i} \|x_j - x_i\|^2. \tag{15}$$

Krum requires computing all pair-wise distances of vectors from $\{x_1, \ldots, x_n\}$ resulting in $\mathcal{O}(n^2)$ computation cost for the server. Therefore, Krum is computationally expensive, when number of workers $n$ is large.

**Robust Federated Averaging.**    RFA-estimator finds a geometric median:

$$\text{RFA}(x_1, \ldots, x_n) \stackrel{\text{def}}{=} \underset{x \in \mathbb{R}^d}{\arg\min} \sum_{i=1}^{n} \|x - x_i\|. \tag{16}$$

The above problem has no closed form solution. However, one can compute approximate RFA using several steps of smoothed Weiszfeld algorithm having $\mathcal{O}(n)$ computation cost of each iteration (Weiszfeld, 1937; Pillutla et al., 2022).

**Coordinate-wise Median.**    CM-estimator computes a median of each component separately. That is, for $t$-th coordinate it is defined as

$$[\text{CM}(x_1, \ldots, x_n)]_t \stackrel{\text{def}}{=} \text{Median}([x_1]_t, \ldots, [x_n]_t) = \underset{u \in \mathbb{R}}{\arg\min} \sum_{i=1}^{n} |u - [x_i]_t|, \tag{17}$$

where $[x]_t$ is $t$-th coordinate of vector $x \in \mathbb{R}^d$. CM has $\mathcal{O}(n)$ computation cost (Chen et al., 2017; Yin et al., 2018).

**Robustness via bucketing.** The following lemma is the key to show robustness of `Krum ∘ Bucketing`, `RFA ∘ Bucketing`, and `CM ∘ Bucketing` in terms of Definition 2.1.

**Lemma D.1** (Modification of Lemma 1 from (Karimireddy et al., 2022))**.** *Assume that $\{x_1, x_2, \ldots, x_n\}$ is such that there exists a subset $\mathcal{G} \subseteq [n]$, $|\mathcal{G}| = G \geq (1 - \delta)n$ and $\sigma \geq 0$ such that (14) holds. Let vectors $\{y_1, \ldots, y_N\}$, $N = \,^{n}\!/\!s\,^{4}$ be generated by Algorithm 2 and $\widetilde{\mathcal{G}} = \{i \in [N] \mid B_i \subseteq \mathcal{G}\}$, where $y_i = \frac{1}{|B_i|} \sum_{j \in B_i} x_j$ and $B_i$ denotes the $i$-th bucket, i.e., $B_i = \{\pi((i-1) \cdot s + 1), \ldots, \pi(\min\{i \cdot s, n\})\}$. Then, $|\widetilde{\mathcal{G}}| = \widetilde{G} \geq (1 - \delta s)N$ and for any fixed $i, l \in \widetilde{\mathcal{G}}$ we have*

$$\mathbb{E}[y_i] = \mathbb{E}[\overline{x}] \quad and \quad \mathbb{E}\left[\|y_i - y_l\|^2\right] \leq \frac{\sigma^2}{s}, \tag{18}$$

*where $\overline{x} = \frac{1}{G} \sum_{i \in \mathcal{G}} x_i$.*

*Proof.* The proof is *almost* identical to the proof of Lemma 1 from (Karimireddy et al., 2022). Nevertheless, for the sake of mathematical rigor, we provide a complete proof. Since each Byzantine peer is contained in no more than 1 bucket and $|\mathcal{B}| \leq \delta n$ (here $\mathcal{B} = [n] \setminus \mathcal{G}$), we have that number of "bad" buckets is not greater than $\delta n \leq \delta s N$, i.e., $\widetilde{G} \geq (1 - \delta s)N$. Next, for any fixed $i \in \widetilde{\mathcal{G}}$ we have

$$\mathbb{E}_\pi[y_i \mid i \in \widetilde{\mathcal{G}}] = \frac{1}{|B_i|} \sum_{j \in B_i} \mathbb{E}_\pi[x_j \mid j \in \mathcal{G}] = \frac{1}{G} \sum_{j \in \mathcal{G}} x_j = \overline{x},$$

where $\mathbb{E}_\pi[\cdot]$ denotes the expectation w.r.t. the randomness comming from the permutation. Taking the full expectation, we obtain the first part of (18). To derive the second part we introduce the notation for workers from $B_i$ and $B_l$: $B_i = \{k_{i,1}, k_{i,2}, \ldots, k_{i,s}\}$ and $B_l = \{k_{l,1}, k_{l,2}, \ldots, k_{l,s}\}$. Then, for any fixed $i, l \in \widetilde{\mathcal{G}}$ the (ordered) pairs $\{(x_{k_{i,t}}, x_{k_{l,t}})\}_{t=1}^s$ are identically distributed random variables as well as for any fixed $t = 1, \ldots, s$ vectors $x_{k_{i,t}}, x_{k_{l,t}}$ are identically distributed. Therefore, we have

$$
\begin{aligned}
\mathbb{E}\left[\|y_i - y_l\|^2\right] &= \mathbb{E}\left[\left\|\frac{1}{s}\sum_{t=1}^s (x_{k_{i,t}} - x_{k_{l,t}})\right\|^2\right] \\
&= \frac{1}{s^2}\sum_{t=1}^s \mathbb{E}\left[\|x_{k_{i,t}} - x_{k_{l,t}}\|^2\right] + \frac{2}{s^2}\sum_{1 \leq t_1 < t_2 \leq s} \mathbb{E}\left[\langle x_{k_{i,t_1}} - x_{k_{l,t_1}}, x_{k_{i,t_2}} - x_{k_{l,t_2}}\rangle\right] \\
&= \frac{1}{s}\mathbb{E}\left[\|x_{k_{i,1}} - x_{k_{l,1}}\|^2\right] + \frac{s-1}{s}\mathbb{E}\left[\langle x_{k_{i,1}} - x_{k_{l,1}}, x_{k_{i,2}} - x_{k_{l,2}}\rangle\right] \\
&= \frac{1}{sG(G-1)} \sum_{\substack{i_1, i_2 \in \mathcal{G} \\ i_1 \neq i_2}} \mathbb{E}\left[\|x_{i_1} - x_{i_2}\|^2\right] \\
&\quad + \underbrace{\frac{s-1}{sG(G-1)(G-2)(G-3)} \sum_{\substack{i_1, i_2, i_3, i_4 \in \mathcal{G} \\ i_1 \neq i_2, i_3, i_4 \\ i_2 \neq i_3, i_4 \\ i_3 \neq i_4}} \mathbb{E}\left[\langle x_{i_1} - x_{i_2}, x_{i_3} - x_{i_4}\rangle\right]}_{S} \\
&= \frac{1}{sG(G-1)} \sum_{i_1, i_2 \in \mathcal{G}} \mathbb{E}\left[\|x_{i_1} - x_{i_2}\|^2\right] + S \overset{(14)}{\leq} \frac{\sigma^2}{s} + S.
\end{aligned}
$$

Finally, we notice that $S = 0$: for any fixed $i_1, i_2, i_3, i_4$ the sum contains one term proportional to $\mathbb{E}\left[\langle x_{i_1} - x_{i_2}, x_{i_3} - x_{i_4}\rangle\right]$ and one term proportional to $\mathbb{E}\left[\langle x_{i_2} - x_{i_1}, x_{i_3} - x_{i_4}\rangle\right]$ that cancel out. This concludes the proof. $\qquad\square$

Using the above lemma, we get the following result.

---

[4]For simplicity, we assume that $n$ is divisible by $s$.

**Theorem D.1** (Modification of Theorem I from (Karimireddy et al., 2022)). *Let $\{x_1, x_2, \ldots, x_n\}$ satisfy the conditions of Lemma D.1 for some $\delta \leq \delta_{\max}$. If Algorithm 2 is run with $s = \lfloor \delta_{\max}/\delta \rfloor$, then*

- $\texttt{Krum} \circ \texttt{Bucketing}$ *satisfies Definition 2.1 with $c = \mathcal{O}(1)$ and $\delta_{\max} < 1/4$,*

- $\texttt{RFA} \circ \texttt{Bucketing}$ *satisfies Definition 2.1 with $c = \mathcal{O}(1)$ and $\delta_{\max} < 1/2$,*

- $\texttt{CM} \circ \texttt{Bucketing}$ *satisfies Definition 2.1 with $c = \mathcal{O}(d)$ and $\delta_{\max} < 1/2$.*

*Proof.* The proof is identical to the proof of Theorem I from (Karimireddy et al., 2022), since Karimireddy et al. (2022) rely only on the general properties of $\texttt{Krum}$/$\texttt{RFA}$/$\texttt{CM}$ and (18) to get the result. $\qquad\square$

# E MISSING PROOFS AND DETAILS FROM SECTION 2

## E.1 EXAMPLES OF SAMPLINGS

Below we provide two examples of situations when Assumption 2.4 holds. In both cases, we assume that $f_{i,j}$ is $L_{i,j}$-smooth for all $i \in \mathcal{G}, j \in [m]$.

**Example E.1** (Uniform sampling with replacement). *Consider $\widehat{\Delta}_i(x,y) = \frac{1}{b} \sum_{j \in I_{i,k}} \Delta_{i,j}(x,y)$, where $\Delta_{i,j}(x,y) = \nabla f_{i,j}(x) - \nabla f_{i,j}(y)$ and $I_{i,k}$ is the set of $b$ i.i.d. samples from the uniform distribution on $[m]$. Then, Assumption 2.4 holds with $\mathcal{L}_\pm^2 = L_{\pm,US}^2$, where $L_{\pm,US}^2 \leq \frac{1}{G} \sum_{i \in \mathcal{G}} L_{i,\pm,US}^2$ and $L_{i,\pm,US}^2$ is such that*

$$\frac{1}{m} \sum_{j=1}^m \|\Delta_{i,j}(x,y) - \Delta_i(x,y)\|^2 \leq L_{i,\pm,US}^2 \|x-y\|^2.$$

*Lemma 2 from Szlendak et al. (2021) implies that $L_{i,US}^2 - L_i^2 \leq L_{i,\pm,US}^2 \leq L_{i,US}^2$, where $L_{i,US}^2$ is such that $\frac{1}{m} \sum_{j=1}^m \|\Delta_{i,j}(x,y)\|^2 \leq L_{i,US}^2 \|x-y\|^2$. We point out that in the worst case $L_{i,US}^2 = \frac{1}{m} \sum_{j=1}^m L_{i,j}^2$.*

**Example E.2** (Importance sampling with replacement). *Consider $\widehat{\Delta}_i^k = \frac{1}{b} \sum_{j \in I_{i,k}} \frac{\overline{L}_i}{L_{i,j}} \Delta_{i,j}(x,y)$, where $\Delta_{i,j}(x,y) = \nabla f_{i,j}(x) - \nabla f_{i,j}(y)$, $\overline{L}_i = \frac{1}{m} \sum_{j=1}^m L_{i,j}$, and $I_{i,k}$ is the set of $b$ i.i.d. samples from the distribution $\mathcal{D}_{i,IS}$ on $[m]$ such that for $j \sim \mathcal{D}_{i,IS}$ we have $\mathbb{P}\{j = t\} = \frac{L_{i,t}}{m\overline{L}_i}$. Then, Assumption 2.4 holds with $\mathcal{L}_\pm^2 = L_{\pm,IS}^2$ such that*

$$\frac{1}{mG} \sum_{i \in \mathcal{G}} \sum_{j=1}^m \frac{\overline{L}_i}{L_{i,j}} \|\Delta_{i,j}(x,y)\|^2 - \frac{1}{mG} \sum_{i \in \mathcal{G}} \|\Delta_i(x,y)\|^2 \leq L_{\pm,IS}^2 \|x-y\|^2.$$

*Lemma 2 from Szlendak et al. (2021) implies that $\frac{1}{G} \sum_{i \in \mathcal{G}} (\overline{L}_i^2 - L_i^2) \leq L_{\pm,IS}^2 \leq \frac{1}{G} \sum_{i \in \mathcal{G}} \overline{L}_i^2$. We point out that $\overline{L}_i^2 \leq L_{i,US}^2$ and in the worst case $m\overline{L}_i^2 = L_{i,US}^2$. Therefore, typically $L_{\pm,IS}^2 < L_{\pm,US}^2$.*

Next, we show that Assumption 2.4 holds whenever $f_{i,j}$ is $L_{i,j}$-smooth for all $i \in \mathcal{G}, j \in [m]$ and $\widehat{\Delta}_i(x,y)$ can be written as

$$\widehat{\Delta}_i(x,y) = \frac{1}{m} \sum_{j=1}^m \xi_{i,j} \left( \nabla f_{i,j}(x) - \nabla f_{i,j}(y) \right),$$

for some random variables $\xi_{i,j}$ such that $\mathbb{E}[\xi_{i,j}] = 1$, $i \in \mathcal{G}, j \in [m]$ and $\max_{i \in \mathcal{G}, j \in [m]} \mathbb{E}[\xi_{i,j}^2] = E^2 < \infty$[5]. Indeed, in this case, we have

$$
\begin{aligned}
\frac{1}{G} \sum_{i \in \mathcal{G}} \mathbb{E} \left\| \widehat{\Delta}_i(x,y) - \Delta_i(x,y) \right\|^2 &\leq \frac{1}{G} \sum_{i \in \mathcal{G}} \mathbb{E} \left\| \widehat{\Delta}_i(x,y) \right\|^2 \\
&= \frac{1}{G} \sum_{i \in \mathcal{G}} \mathbb{E} \left\| \frac{1}{m} \sum_{j=1}^m \xi_{i,j} \left( \nabla f_{i,j}(x) - \nabla f_{i,j}(y) \right) \right\|^2 \\
&\leq \frac{1}{Gm} \sum_{i \in \mathcal{G}} \sum_{j=1}^m \|\nabla f_{i,j}(x) - \nabla f_{i,j}(y)\|^2 \mathbb{E}\xi_{i,j}^2 \\
&\leq \frac{E^2}{GM} \sum_{i \in \mathcal{G}} \sum_{j=1}^m L_i^2 \|x-y\|^2 \\
&\leq E^2 \max_{i \in \mathcal{G}, j \in [m]} L_{i,j}^2 \|x-y\|^2,
\end{aligned}
$$

meaning that Assumption 2.4 holds with $\mathcal{L}_\pm \leq \sqrt{b} E \max_{i \in \mathcal{G}, j \in [m]} L_{i,j}$. However, as we show in Examples E.1 and E.2 constant $\mathcal{L}_\pm$ can be much smaller than the derived upper bound.

---

[5]We note that this assumption on the form of $\widehat{\Delta}_i(x,y)$ is very mild and holds for standard sampling strategies including uniform and importance samplings. We refer to (Gower et al., 2019) for more examples.

### E.2 KEY LEMMAS

Our theory works for a slightly more general setting than the one we discussed in the main part of the paper. In particular, instead of Assumption 2.2 we consider a more general assumption on the heterogeneity.

**Assumption E.1** (($B, \zeta^2$)-heterogeneity). *We assume that good clients have* ($B, \zeta^2$)-*heterogeneous local loss functions for some* $B \geq 0, \zeta \geq 0$, *i.e.,*

$$\frac{1}{G} \sum_{i \in \mathcal{G}} \|\nabla f_i(x) - \nabla f(x)\|^2 \leq B\|\nabla f(x)\|^2 + \zeta^2 \quad \forall x \in \mathbb{R}^d. \tag{19}$$

When $B = 0$ the above assumption recovers Assumption 2.2. Since we allow $B$ to be positive, it may reduce the value of $\zeta$ in some applications (for example, for over-parameterized models). As we show further, the best possible optimization error that Byz-VR-MARINA can achieve in this case is proportional to $\zeta^2$. We refer to Karimireddy et al. (2022) for the study of typical values of parameter $B$ for some over-parameterized models.

In proofs, we need the following lemma, wich is often used for analyzing SGD-like methods in the non-convex case.

**Lemma E.1** (Lemma 2 from Li et al. (2021)). *Assume that function $f$ is $L$-smooth and $x^{k+1} = x^k - \gamma g^k$. Then*

$$f(x^{k+1}) \leq f(x^k) - \frac{\gamma}{2}\|\nabla f(x^k)\|^2 - \left(\frac{1}{2\gamma} - \frac{L}{2}\right)\|x^{k+1} - x^k\|^2 + \frac{\gamma}{2}\|g^k - \nabla f(x^k)\|^2. \tag{20}$$

To estimate the "quality" of robust aggregation at iteration $k + 1$ we derive an upper bound for the averaged pairwise variance of estimators obtained by good peers (see also Definition 2.1).

**Lemma E.2** (Bound on the variance). *Let Assumptions 2.1, E.1, 2.3, 2.4 hold. Then for all $k \geq 0$ the iterates produced by Byz-VR-MARINA satisfy*

$$\frac{1}{G(G-1)} \sum_{i,l \in \mathcal{G}} \mathbb{E}\left[\|g_i^{k+1} - g_l^{k+1}\|^2\right] \leq A'\mathbb{E}\left[\|x^{k+1} - x^k\|^2\right] + 8Bp\mathbb{E}\|\nabla f(x^k)\|^2 + 4p\zeta^2, \tag{21}$$

*where $A' = \left(8BpL^2 + 4(1-p)\left(\omega L^2 + (1+\omega)L_{\pm}^2 + \frac{(1+\omega)\mathcal{L}_{\pm}^2}{b}\right)\right).$*

*Proof.* For the compactness, we introduce new notation: $\Delta^k = \nabla f(x^{k+1}) - \nabla f(x^k)$. Let $\mathbb{E}_{c_k}[\cdot]$ denote the expectation w.r.t. $c_k$. Then, by definition of $g_i^{k+1}$ for $i \in \mathcal{G}$ we have

$$\frac{1}{G(G-1)} \sum_{i,l \in \mathcal{G}} \mathbb{E}_{c_k}\left[\|g_i^{k+1} - g_l^{k+1}\|^2\right] = \underbrace{\frac{p}{G(G-1)} \sum_{i,l \in \mathcal{G}} \|\nabla f_i(x^{k+1}) - \nabla f_l(x^{k+1})\|^2}_{T_1}$$

$$+ \underbrace{\frac{1-p}{G(G-1)} \sum_{i,l \in \mathcal{G}} \|\mathcal{Q}(\widehat{\Delta}_i^k) - \mathcal{Q}(\widehat{\Delta}_l^k)\|^2}_{T_2}.$$

Taking the full expectation and using the tower property $\mathbb{E}[\mathbb{E}_{c_k}[\cdot]] = \mathbb{E}[\cdot]$, we derive

$$\frac{1}{G(G-1)} \sum_{i,l \in \mathcal{G}} \mathbb{E}\left[\|g_i^{k+1} - g_l^{k+1}\|^2\right] = \mathbb{E}[T_1] + \mathbb{E}[T_2]. \tag{22}$$

Term $\mathbb{E}[T_1]$ can be bounded via Assumption 19:

$$
\begin{aligned}
\mathbb{E}[T_1] \quad &= \quad \frac{p}{G(G-1)} \sum_{\substack{i,l \in \mathcal{G} \\ i \neq l}} \mathbb{E}\left[\|\nabla f_i(x^{k+1}) - \nabla f_l(x^{k+1})\|^2\right] \\
&\stackrel{(11)}{\leq} \quad \frac{p}{G(G-1)} \sum_{\substack{i,l \in \mathcal{G} \\ i \neq l}} \mathbb{E}\left[2\|\nabla f_i(x^{k+1}) - \nabla f(x^{k+1})\|^2 + 2\|\nabla f_l(x^{k+1}) - \nabla f(x^{k+1})\|^2\right] \\
&= \quad \frac{4p}{G} \sum_{i \in \mathcal{G}} \mathbb{E}\left[\|\nabla f_i(x^{k+1}) - \nabla f(x^{k+1})\|^2\right] \\
&\stackrel{(19)}{\leq} \quad 4Bp\mathbb{E}\left[\|\nabla f(x^{k+1})\|^2\right] + 4p\zeta^2 \\
&\stackrel{(11)}{\leq} \quad 8Bp\mathbb{E}\left[\|\nabla f(x^k)\|^2\right] + 8Bp\mathbb{E}\left[\|\nabla f(x^{k+1}) - \nabla f(x^k)\|^2\right] + 4p\zeta^2 \\
&\stackrel{As.\ 2.1}{\leq} \quad 8Bp\mathbb{E}\left[\|\nabla f(x^k)\|^2\right] + 8BpL^2\mathbb{E}\left[\|x^{k+1} - x^k\|^2\right] + 4p\zeta^2.
\end{aligned}
$$

To estimate $\mathbb{E}[T_2]$ we first derive an upper bound for $\mathbb{E}_k[T_2]$, where $\mathbb{E}_k[\cdot]$ denotes expectation w.r.t. the all randomness (compression and stochasticity of the the gradients) coming from the step $k+1$ of the algorithm:

$$
\begin{aligned}
\mathbb{E}_k[T_2] \quad &= \quad \frac{1-p}{G(G-1)} \sum_{\substack{i,l \in \mathcal{G} \\ i \neq l}} \mathbb{E}_k\left[\|\mathcal{Q}(\widehat{\Delta}_i^k) - \mathcal{Q}(\widehat{\Delta}_l^k)\|^2\right] \\
&= \quad \frac{1-p}{G(G-1)} \sum_{\substack{i,l \in \mathcal{G} \\ i \neq l}} \mathbb{E}_k\left[\|\mathcal{Q}(\widehat{\Delta}_i^k) - \Delta_i^k - (\mathcal{Q}(\widehat{\Delta}_l^k) - \Delta_l^k)\|^2\right] \\
&\qquad + \frac{1-p}{G(G-1)} \sum_{\substack{i,l \in \mathcal{G} \\ i \neq l}} \|\Delta_i^k - \Delta_l^k\|^2 \\
&\stackrel{(11)}{\leq} \quad \frac{1-p}{G(G-1)} \sum_{\substack{i,l \in \mathcal{G} \\ i \neq l}} \mathbb{E}_k\left[2\|\mathcal{Q}(\widehat{\Delta}_i^k) - \Delta_i^k\|^2 + 2\|\mathcal{Q}(\widehat{\Delta}_l^k) - \Delta_l^k\|^2\right] \\
&\qquad + \frac{1-p}{G(G-1)} \sum_{\substack{i,l \in \mathcal{G} \\ i \neq l}} \left(2\|\Delta_i^k - \Delta^k\|^2 + 2\|\Delta_l^k - \Delta^k\|^2\right) \\
&= \quad \frac{1-p}{G(G-1)} \sum_{\substack{i,l \in \mathcal{G} \\ i \neq l}} \mathbb{E}_k\left[\mathbb{E}_{\mathcal{Q}_k}\left[2\|\mathcal{Q}(\widehat{\Delta}_i^k) - \Delta_i^k\|^2 + 2\|\mathcal{Q}(\widehat{\Delta}_l^k) - \Delta_l^k\|^2\right]\right] \\
&\qquad + \frac{1-p}{G(G-1)} \sum_{\substack{i,l \in \mathcal{G} \\ i \neq l}} \left(2\|\Delta_i^k - \Delta^k\|^2 + 2\|\Delta_l^k - \Delta^k\|^2\right) \\
&= \quad \frac{4(1-p)}{G} \sum_{i \in \mathcal{G}} \mathbb{E}_k\left[\mathbb{E}_{\mathcal{Q}_k}\left[\|\mathcal{Q}(\widehat{\Delta}_i^k)\|^2\right] - \|\Delta_i^k\|^2\right] + \frac{4(1-p)}{G} \sum_{i \in \mathcal{G}} \left(\|\Delta_i^k\|^2 - \|\Delta^k\|^2\right) \\
&\stackrel{(3)}{\leq} \quad \frac{4(1-p)}{G} \sum_{i \in \mathcal{G}} \mathbb{E}_k\left[(1+\omega)\|\widehat{\Delta}_i^k\|^2 - \|\Delta_i^k\|^2\right] + \frac{4(1-p)}{G} \sum_{i \in \mathcal{G}} \left(\|\Delta_i^k\|^2 - \|\Delta^k\|^2\right) \\
&= \quad \frac{4(1-p)(1+\omega)}{G} \sum_{i \in \mathcal{G}} \mathbb{E}_k\left[\|\widehat{\Delta}_i^k - \Delta_i^k\|^2\right] + \frac{4(1-p)(1+\omega)}{G} \sum_{i \in \mathcal{G}} \left(\|\Delta_i^k - \Delta^k\|^2\right) \\
&\qquad + 4(1-p)\omega\|\Delta^k\|^2,
\end{aligned}
$$

where $\mathbb{E}_{Q_k}[\cdot]$ denotes the expectation w.r.t. the randomness coming from the compression at step $k+1$. Applying Assumptions 2.4, 2.3, and 2.1 and taking the full expectation, we get

$$
\mathbb{E}[T_2] \quad \leq \quad 4(1-p)\left(\omega L^2 + (1+\omega)\left(L_\pm^2 + \frac{\mathcal{L}_\pm^2}{b}\right)\right)\mathbb{E}\left[\|x^{k+1} - x^k\|^2\right].
$$

Plugging the upper bounds for $\mathbb{E}[T_1]$ and $\mathbb{E}[T_2]$ in (22), we obtain the result. $\qquad\square$

Using the above lemma, we derive the following technical result, which we rely on in the proofs of the main results.

**Lemma E.3** (Bound on the distortion). *Let Assumptions 2.1, E.1, 2.3, 2.4 hold. Then for all $k \geq 0$ the iterates produced by* Byz-VR-MARINA *satisfy*

$$
\mathbb{E}\left[\|g^{k+1} - \nabla f(x^{k+1})\|^2\right] \leq \left(1 - \frac{p}{2}\right)\mathbb{E}\left[\|g^k - \nabla f(x^k)\|^2\right] + 24Bc\delta\mathbb{E}\|\nabla f(x^k)\|^2 + 12c\delta\zeta^2
$$
$$
+\frac{Ap}{4}\mathbb{E}\left[\|x^{k+1} - x^k\|^2\right], \tag{23}
$$

*where* $A = \frac{48BL^2c\delta}{p} + \frac{6(1-p)}{p}\left(\frac{4c\delta}{p} + \frac{1}{2G}\right)\left(\omega L^2 + \frac{(1+\omega)\mathcal{L}_{\pm}^2}{b}\right) + \frac{6(1-p)}{p}\left(\frac{4c\delta(1+\omega)}{p} + \frac{\omega}{2G}\right)L_{\pm}^2.$

*Proof.* For convenience, we intoduce the following notation:

$$
\overline{g}^{k+1} = \frac{1}{G}\sum_{i\in\mathcal{G}} g_i^{k+1} = \begin{cases} \nabla f(x^{k+1}), & \text{if } c_k = 1, \\ g^k + \frac{1}{G}\sum_{i\in\mathcal{G}}\mathcal{Q}(\widehat{\Delta}_i^k), & \text{otherwise.} \end{cases} \tag{24}
$$

Using the introduced notation, we derive

$$
\mathbb{E}\left[\|g^{k+1} - \nabla f(x^{k+1})\|^2\right] \overset{(11)}{\leq} \left(1 + \frac{p}{2}\right)\mathbb{E}\left[\|\overline{g}^{k+1} - \nabla f(x^{k+1})\|^2\right]
$$
$$
+ \left(1 + \frac{2}{p}\right)\mathbb{E}\left[\|g^{k+1} - \overline{g}^{k+1}\|^2\right]. \tag{25}
$$

Next, we need to upper-bound the terms from the right-hand side of (25). Let $\mathbb{E}_{c_k}[\cdot]$ denote the expectation w.r.t. $c_k$. Then, in view of (24), we have

$$
\mathbb{E}_{c_k}\left[\|g^{k+1} - \nabla f(x^{k+1})\|^2\right] = (1-p)\left\|g^k + \frac{1}{G}\sum_{i\in\mathcal{G}}\mathcal{Q}(\widehat{\Delta}_i^k) - \nabla f(x^{k+1})\right\|^2.
$$

Taking expectation $\mathbb{E}_k[\cdot]$ w.r.t. the all randomness (compression and stochasticity of the the gradients) coming from the step $k+1$ of the algorithm and applying the variance decomposition and independence of mini-batch and compression computations on different workers, we get

$$
\mathbb{E}_k\left[\|\overline{g}^{k+1} - \nabla f(x^{k+1})\|^2\right] = (1-p)\mathbb{E}_k\left[\left\|g^k + \frac{1}{G}\sum_{i\in\mathcal{G}}\mathcal{Q}(\widehat{\Delta}_i^k) - \nabla f(x^{k+1})\right\|^2\right]
$$
$$
= (1-p)\|g^k - \nabla f(x^k)\|^2
$$
$$
+(1-p)\mathbb{E}_k\left[\left\|\frac{1}{G}\sum_{i\in\mathcal{G}}(\mathcal{Q}(\widehat{\Delta}_i^k) - \Delta_i^k)\right\|^2\right]
$$
$$
= (1-p)\|g^k - \nabla f(x^k)\|^2 + \frac{1-p}{G^2}\sum_{i\in\mathcal{G}}\mathbb{E}_k\left[\left\|\mathcal{Q}(\widehat{\Delta}_i^k) - \Delta_i^k\right\|^2\right].
$$

Let $\mathbb{E}_{\mathcal{Q}_k}[\cdot]$ denote the expectation w.r.t. the randomness coming from the compression at step $k+1$. The definition of the unbiased compression operator (Definition 2.2) implies

$$
\begin{aligned}
\mathbb{E}_k\left[\|\overline{g}^{k+1} - \nabla f(x^{k+1})\|^2\right] &= (1-p)\|g^k - \nabla f(x^k)\|^2 + \frac{1-p}{G^2}\sum_{i\in\mathcal{G}}\mathbb{E}_k\left[\|\mathcal{Q}(\widehat{\Delta}_i^k)\|^2\right]\\
&\quad - \frac{1-p}{G^2}\sum_{i\in\mathcal{G}}\|\Delta_i^k\|^2\\
&= (1-p)\|g^k - \nabla f(x^k)\|^2 + \frac{1-p}{G^2}\sum_{i\in\mathcal{G}}\mathbb{E}_k\left[\mathbb{E}_{\mathcal{Q}_k}\left[\|\mathcal{Q}(\widehat{\Delta}_i^k)\|^2\right]\right]\\
&\quad - \frac{1-p}{G^2}\sum_{i\in\mathcal{G}}\|\Delta_i^k\|^2\\
&\overset{(3)}{\leq} (1-p)\|g^k - \nabla f(x^k)\|^2 + \frac{(1-p)(1+\omega)}{G^2}\sum_{i\in\mathcal{G}}\mathbb{E}_k\left[\|\widehat{\Delta}_i^k\|^2\right]\\
&\quad - \frac{1-p}{G^2}\sum_{i\in\mathcal{G}}\|\Delta_i^k\|^2\\
&= (1-p)\|g^k - \nabla f(x^k)\|^2\\
&\quad + \frac{(1-p)(1+\omega)}{G^2}\sum_{i\in\mathcal{G}}\mathbb{E}_k\left[\|\widehat{\Delta}_i^k - \Delta_i^k\|^2\right]\\
&\quad + \frac{(1-p)\omega}{G^2}\sum_{i\in\mathcal{G}}\|\Delta_i^k\|^2\\
&= (1-p)\|g^k - \nabla f(x^k)\|^2\\
&\quad + \frac{(1-p)(1+\omega)}{G^2}\sum_{i\in\mathcal{G}}\mathbb{E}_k\left[\|\widehat{\Delta}_i^k - \Delta_i^k\|^2\right]\\
&\quad + \frac{(1-p)\omega}{G^2}\sum_{i\in\mathcal{G}}\|\Delta_i^k - \Delta^k\|^2 + \frac{(1-p)\omega}{G}\|\Delta^k\|^2.
\end{aligned}
$$

Using Assumptions 2.4, 2.3, and 2.1 and taking the full expectation, we arrive at

$$
\begin{aligned}
\mathbb{E}\left[\|\overline{g}^{k+1} - \nabla f(x^{k+1})\|^2\right] &\leq (1-p)\mathbb{E}\left[\|g^k - \nabla f(x^k)\|^2\right] \quad\quad (26)\\
&\quad + \frac{1-p}{G}\left(\omega L^2 + \omega L_{\pm}^2 + \frac{(1+\omega)\mathcal{L}_{\pm}^2}{b}\right)\mathbb{E}\left[\|x^{k+1} - x^k\|^2\right].
\end{aligned}
$$

That is, we obtained an upper bound for the first term in the right-hand side of (25). To bound the second term, we use the definition of $(\delta, c)$-ARAgg (Definition 2.1) and Lemma E.2:

$$
\begin{aligned}
\mathbb{E}\left[\|g^{k+1} - \overline{g}^{k+1}\|^2\right] &= \mathbb{E}\left[\mathbb{E}_k\left[\|g^{k+1} - \overline{g}^{k+1}\|^2\right]\right]\\
&\overset{(2)}{\leq} \mathbb{E}\left[\frac{c\delta}{G(G-1)}\sum_{i,l\in\mathcal{G}}\mathbb{E}_k\left[\|g_i^{k+1} - g_l^{k+1}\|^2\right]\right]\\
&= \frac{c\delta}{G(G-1)}\sum_{i,l\in\mathcal{G}}\mathbb{E}\left[\|g_i^{k+1} - g_l^{k+1}\|^2\right]\\
&\overset{(21)}{\leq} 8Bpc\delta + 4p\zeta^2 c\delta + A'c\delta\mathbb{E}\left[\|x^{k+1} - x^k\|^2\right], \quad\quad (27)
\end{aligned}
$$

where $A' = \left(8BpL^2 + 4(1-p)\left(\omega L^2 + (1+\omega)L_\pm^2 + \frac{(1+\omega)\mathcal{L}_\pm^2}{b}\right)\right)$. Plugging (26) and (27) in (25) and using $p \le 1$, we obtain

$$
\begin{aligned}
\mathbb{E}\left[\|g^{k+1} - \nabla f(x^{k+1})\|^2\right] &\le (1-p)\left(1 + \frac{p}{2}\right)\mathbb{E}\left[\|g^k - \nabla f(x^k)\|^2\right] \\
&\quad + \frac{3(1-p)}{2G}\left(\omega L^2 + \omega L_\pm^2 + \frac{(1+\omega)\mathcal{L}_\pm^2}{b}\right)\mathbb{E}\left[\|x^{k+1} - x^k\|^2\right] \\
&\quad + \frac{3}{p}\left(8Bpc\delta + 4p\zeta^2 c\delta + A'c\delta\mathbb{E}\left[\|x^{k+1} - x^k\|^2\right]\right) \\
&\stackrel{(13)}{\le} \left(1 - \frac{p}{2}\right)\mathbb{E}\left[\|g^k - \nabla f(x^k)\|^2\right] + 24Bc\delta + 12\zeta^2 c\delta \\
&\quad + \frac{Ap}{2}\mathbb{E}\left[\|x^{k+1} - x^k\|^2\right],
\end{aligned}
$$

where

$$
\begin{aligned}
A &= \frac{2}{p}\left(\frac{3(1-p)}{2G}\left(\omega L^2 + \omega L_\pm^2 + \frac{(1+\omega)\mathcal{L}_\pm^2}{b}\right) + \frac{3}{p}A'c\delta\right) \\
&= \frac{2}{p}\left(24BL^2 c\delta + 3(1-p)\left(\frac{4c\delta}{p} + \frac{1}{2G}\right)\left(\omega L^2 + \frac{(1+\omega)\mathcal{L}_\pm^2}{b}\right)\right) \\
&\quad + \frac{2}{p}\cdot 3(1-p)\left(\frac{4c\delta(1+\omega)}{p} + \frac{\omega}{2G}\right)L_\pm^2 \\
&= \frac{48BL^2 c\delta}{p} + \frac{6(1-p)}{p}\left(\frac{4c\delta}{p} + \frac{1}{2G}\right)\left(\omega L^2 + \frac{(1+\omega)\mathcal{L}_\pm^2}{b}\right) \\
&\quad + \frac{6(1-p)}{p}\left(\frac{4c\delta(1+\omega)}{p} + \frac{\omega}{2G}\right)L_\pm^2.
\end{aligned}
$$

This concludes the proof. $\qquad\square$

### E.3  GENERAL NON-CONVEX FUNCTIONS

**Theorem E.1** (Generalized version of Theorem 2.1). *Let Assumptions 2.1, E.1, 2.3, 2.4 hold. Assume that*

$$
0 < \gamma \le \frac{1}{L + \sqrt{A}}, \quad \delta < \frac{p}{48cB}, \tag{28}
$$

*where $A = \frac{48BL^2 c\delta}{p} + \frac{6(1-p)}{p}\left(\frac{4c\delta}{p} + \frac{1}{2G}\right)\left(\omega L^2 + \frac{(1+\omega)\mathcal{L}_\pm^2}{b}\right) + \frac{6(1-p)}{p}\left(\frac{4c\delta(1+\omega)}{p} + \frac{\omega}{2G}\right)L_\pm^2$. Then for all $K \ge 0$ the iterates produced by* Byz-VR-MARINA *satisfy*

$$
\mathbb{E}\left[\|\nabla f(\widehat{x}^K)\|^2\right] \le \frac{2\Phi_0}{\gamma\left(1 - \frac{48Bc\delta}{p}\right)(K+1)} + \frac{24c\delta\zeta^2}{p - 48Bc\delta}, \tag{29}
$$

*where $\widehat{x}^K$ is choosen uniformly at random from $x^0, x^1, \ldots, x^K$, and $\Phi_0 = f(x^0) - f_* + \frac{\gamma}{p}\|g^0 - \nabla f(x^0)\|^2$. The result of Theorem 2.1 is a special case of the statement above with $B = 0$, since for $B = 0$ we have $A = \frac{48BL^2 c\delta}{p} + \frac{6(1-p)}{p}\left(\frac{4c\delta}{p} + \frac{1}{2G}\right)\left(\omega L^2 + \frac{(1+\omega)\mathcal{L}_\pm^2}{b}\right) + \frac{6(1-p)}{p}\left(\frac{4c\delta(1+\omega)}{p} + \frac{\omega}{2G}\right)L_\pm^2 = \frac{6(1-p)}{p}\left(\frac{4c\delta}{p} + \frac{1}{2G}\right)\left(\omega L^2 + \frac{(1+\omega)\mathcal{L}_\pm^2}{b}\right) + \frac{6(1-p)}{p}\left(\frac{4c\delta(1+\omega)}{p} + \frac{\omega}{2G}\right)L_\pm^2$, $\left(1 - \frac{48Bc\delta}{p}\right) = 1$, $p - 48Bc\delta = p$, and the second condition from (28) always holds.*

*Proof.* For all $k \geq 0$ we introduce $\Phi_k = f(x^k) - f_* + \frac{\gamma}{p}\|g^k - \nabla f(x^k)\|^2$. Using the results of Lemmas E.1 and E.3, we derive

$$
\begin{aligned}
\mathbb{E}[\Phi_{k+1}] \overset{(20),(23)}{\leq} \quad & \mathbb{E}\left[f(x^k) - f_* - \left(\frac{1}{2\gamma} - \frac{L}{2}\right)\|x^{k+1} - x^k\|^2 + \frac{\gamma}{2}\|g^k - \nabla f(x^k)\|^2\right] \\
& -\frac{\gamma}{2}\mathbb{E}\left[\|\nabla f(x^k)\|^2\right] + \frac{\gamma}{p}\left(1 - \frac{p}{2}\right)\mathbb{E}\left[\|g^k - \nabla f(x^k)\|^2\right] \\
& +\frac{24Bc\delta\gamma}{p}\mathbb{E}\left[\|\nabla f(x^k)\|^2\right] + \frac{12c\delta\zeta^2\gamma}{p} + \frac{\gamma A}{2}\mathbb{E}\left[\|x^{k+1} - x^k\|^2\right] \\
= \quad & \mathbb{E}[\Phi_k] - \frac{\gamma}{2}\left(1 - \frac{48Bc\delta}{p}\right)\mathbb{E}\left[\|\nabla f(x^k)\|^2\right] + \frac{12c\delta\zeta^2\gamma}{p} \\
& -\frac{1}{2\gamma}\left(1 - L\gamma - A\gamma^2\right)\mathbb{E}\left[\|x^{k+1} - x^k\|^2\right] \\
\leq \quad & \mathbb{E}[\Phi_k] - \frac{\gamma}{2}\left(1 - \frac{48Bc\delta}{p}\right)\mathbb{E}\left[\|\nabla f(x^k)\|^2\right] + \frac{12c\delta\zeta^2\gamma}{p},
\end{aligned}
$$

where in the last step we use Lemma C.1 and our choice of $\gamma$ from (28). Next, in view of (28), we have $\frac{\gamma}{2}\left(1 - \frac{48Bc\delta}{p}\right) > 0$. Therefore, summing up the above inequality for $k = 0, 1, \ldots, K$ and rearranging the terms, we get

$$
\begin{aligned}
\frac{1}{K+1}\sum_{k=0}^{K}\mathbb{E}\left[\|\nabla f(x^k)\|^2\right] \quad \leq \quad & \frac{2}{\gamma\left(1 - \frac{48Bc\delta}{p}\right)(K+1)}\sum_{k=0}^{K}\left(\mathbb{E}[\Phi_k] - \mathbb{E}[\Phi_{k+1}]\right) \\
& +\frac{24c\delta\zeta^2}{p - 48Bc\delta} \\
= \quad & \frac{2\left(\mathbb{E}[\Phi_0] - \mathbb{E}[\Phi_{K+1}]\right)}{\gamma\left(1 - \frac{48Bc\delta}{p}\right)(K+1)} + \frac{24c\delta\zeta^2}{p - 48Bc\delta} \\
\overset{\Phi_{K+1}\geq 0}{\leq} \quad & \frac{2\mathbb{E}[\Phi_0]}{\gamma\left(1 - \frac{48Bc\delta}{p}\right)(K+1)} + \frac{24c\delta\zeta^2}{p - 48Bc\delta}.
\end{aligned}
$$

It remains to notice, that the lef-hand side equals $\mathbb{E}[\|\nabla f(\widehat{x}^K)\|^2]$, where $\widehat{x}^K$ is choosen uniformly at random from $x^0, x^1, \ldots, x^K$. $\qquad\square$

**On the differences between Byz-VR-MARINA and momentum-based methods.** Karimireddy et al. (2021) use momentum and momentum-based variance reduction in order to prevent the algorithm from being permutation-invariant, since *in the setup considered by Karimireddy et al. (2021)* all permuation-invariant algorithms cannot converge to any predefined accuracy even in the homogeneous case. In our paper, we provide a different perspective on this problem: it turns out that the method can be variance-reduced, Byzantine-robust, and permutation-invariant at the same time.

Let us first refine what we mean by permutation-invariance since our definition slightly differs from the one used by Karimireddy et al. (2021). That is, consider the homogeneous setup and assume that there are no Byzantine workers. We say that the algorithm is permutation-invariant if one can arbitrarily permute the results of stochastic gradients computations (not necessarily one stochastic gradient computation) between workers at any aggregation step without changing the output of the method. Then, Byz-VR-MARINA is permutation-invariant, since the output in line 10 depends only on $g^k$ and **the set** $\{\Delta_i^k\}_{i\in[n]}$ (note that $\mathcal{Q}(x) = x$, since we assume $\omega = 0$), not on their order. Our results do not contradict the ones from Karimireddy et al. (2021), since Karimireddy et al. (2021) assume that the variance of the stochastic gradient is bounded, while we apply variance reduction implying that the variance goes to zero and Byzantine workers cannot successfuly "hide in the noise" via time-coupled attacks anymore.

Before we move on to the corollaries, we ellaborate on the derived upper bound. In particular, it is important to estimate $\mathbb{E}[\Phi_0]$. By definition, $\Phi_0 = f(x^0) - f_* + \frac{\gamma}{p}\|g^0 - \nabla f(x^0)\|^2$, i.e., $\Phi_0$ depends

on the choice of $g^0$. For example, one can ask good workers to compute $h_i = \nabla f_i(x^0)$, $i \in \mathcal{G}$ and send it to the server. Then, the server can set $g^0$ as $g^0 = \texttt{ARAgg}(h_1, \ldots, h_n)$. This gives us

$$
\begin{aligned}
\mathbb{E}[\Phi_0] \quad &= \quad f(x^0) - f_* + \frac{\gamma}{p}\mathbb{E}\left[\|g^0 - \nabla f(x^0)\|^2\right] \\
&\overset{(2)}{\leq} \quad f(x^0) - f_* + \frac{\gamma c\delta}{pG(G-1)}\sum_{\substack{i,l\in\mathcal{G}\\i\neq l}}\|\nabla f_i(x^0) - \nabla f_l(x^0)\|^2 \\
&\overset{(11)}{\leq} \quad f(x^0) - f_* + \frac{2\gamma c\delta}{pG(G-1)}\sum_{\substack{i,l\in\mathcal{G}\\i\neq l}}\left(\|\nabla f_i(x^0) - \nabla f(x^0)\|^2 + \|\nabla f_l(x^0) - \nabla f(x^0)\|^2\right) \\
&= \quad f(x^0) - f_* + \frac{4\gamma c\delta}{pG}\sum_{i\in\mathcal{G}}\|\nabla f_i(x^0) - \nabla f(x^0)\|^2 \\
&\overset{(19)}{\leq} \quad f(x^0) - f_* + \frac{4\gamma c\delta B}{p}\|\nabla f(x^0)\|^2 + \frac{4\gamma c\delta\zeta^2}{p}.
\end{aligned}
$$

Function $f$ is $L$-smooth that implies $\|\nabla f(x^0)\|^2 \leq 2L\left(f(x^0) - f_*\right)$. Using this and $\delta < p/(48cB)$ and $\gamma \leq 1/L$, we derive

$$
\begin{aligned}
\mathbb{E}[\Phi_0] \quad &\leq \quad \left(1 + \frac{8\gamma c\delta BL}{p}\right)\left(f(x^0) - f_*\right) + \frac{4\gamma c\delta\zeta^2}{p} \\
&\leq \quad \left(1 + \frac{\gamma L}{6}\right)\left(f(x^0) - f_*\right) + \frac{4\gamma c\delta\zeta^2}{p} \\
&\leq \quad 2\left(f(x^0) - f_*\right) + \frac{4\gamma c\delta\zeta^2}{p}. \quad (30)
\end{aligned}
$$

Plugging this upper bound in (29), we get

$$
\mathbb{E}\left[\|\nabla f(\widehat{x}^K)\|^2\right] \quad \leq \quad \frac{4\left(f(x^0) - f_*\right)}{\gamma\left(1 - \frac{48Bc\delta}{p}\right)(K+1)} + \frac{32c\delta\zeta^2}{p - 48Bc\delta}.
$$

Based on this inequality we derive following corollaries.

**Corollary E.1** (Homogeneous data, no compression ($\omega = 0$)). *Let the assumptions of Theorem E.1 hold, $\mathcal{Q}(x) \equiv x$ for all $x \in \mathbb{R}^d$ (no compression, $\omega = 0$), $p = b/m$, $B = 0$, $\zeta = 0$, and*

$$
\gamma = \frac{1}{L + \mathcal{L}_\pm\sqrt{6\left(\frac{4c\delta m^2}{b^3} + \frac{m}{b^2 G}\right)}}
$$

*Then for all $K \geq 0$ we have $\mathbb{E}\left[\|\nabla f(\widehat{x}^K)\|^2\right]$ of the order*

$$
\mathcal{O}\left(\frac{\left(L + \mathcal{L}_\pm\sqrt{\frac{c\delta m^2}{b^3} + \frac{m}{b^2 G}}\right)\Delta_0}{K}\right), \quad (31)
$$

*where $\widehat{x}^K$ is choosen uniformly at random from the iterates $x^0, x^1, \ldots, x^K$ produced by* Byz-VR-MARINA *and $\Delta_0 = f(x^0) - f_*$. That is, to guarantee $\mathbb{E}\left[\|\nabla f(\widehat{x}^K)\|^2\right] \leq \varepsilon^2$ for $\varepsilon^2 > 0$* Byz-VR-MARINA *requires*

$$
\mathcal{O}\left(\frac{\left(L + \mathcal{L}_\pm\sqrt{\frac{c\delta m^2}{b^3} + \frac{m}{b^2 G}}\right)\Delta_0}{\varepsilon^2}\right), \quad (32)
$$

*communication rounds and*

$$
\mathcal{O}\left(\frac{\left(bL + \mathcal{L}_\pm\sqrt{\frac{c\delta m^2}{b} + \frac{m}{G}}\right)\Delta_0}{\varepsilon^2}\right), \quad (33)
$$

*oracle calls per worker.*

**Corollary E.2** (No compression ($\omega = 0$)). *Let the assumptions of Theorem E.1 hold, $\mathcal{Q}(x) \equiv x$ for all $x \in \mathbb{R}^d$ (no compression, $\omega = 0$), $p = b/m$ and*

$$\gamma = \frac{1}{L + \sqrt{\frac{48L^2 Bc\delta m}{b} + \frac{24c\delta m^2}{b^2}L_\pm^2 + 6\left(\frac{4c\delta m^2}{b^2} + \frac{m}{bG}\right)\frac{\mathcal{L}_\pm^2}{b}}}$$

*Then for all $K \geq 0$ we have $\mathbb{E}\left[\|\nabla f(\widehat{x}^K)\|^2\right]$ of the order*

$$\mathcal{O}\left(\frac{\left(L + \sqrt{\frac{L^2 Bc\delta m}{b} + \frac{c\delta m^2}{b^2}L_\pm^2 + \left(\frac{c\delta m^2}{b^2} + \frac{m}{bG}\right)\frac{\mathcal{L}_\pm^2}{b}}\right)\Delta_0}{\left(1 - \frac{48Bc\delta m}{b}\right)K} + \frac{c\delta\zeta^2}{\frac{b}{m} - 48Bc\delta}\right), \quad (34)$$

*where $\widehat{x}^K$ is choosen uniformly at random from the iterates $x^0, x^1, \ldots, x^K$ produced by* Byz-VR-MARINA *and $\Delta_0 = f(x^0) - f_*$. That is, to guarantee $\mathbb{E}\left[\|\nabla f(\widehat{x}^K)\|^2\right] \leq \varepsilon^2$ for $\varepsilon^2 \geq \frac{12c\delta\zeta^2}{p - 48Bc\delta}$* Byz-VR-MARINA *requires*

$$\mathcal{O}\left(\frac{\left(L + \sqrt{\frac{L^2 Bc\delta m}{b} + \frac{c\delta m^2}{b^2}L_\pm^2 + \left(\frac{c\delta m^2}{b^2} + \frac{m}{bG}\right)\frac{\mathcal{L}_\pm^2}{b}}\right)\Delta_0}{\left(1 - \frac{48Bc\delta m}{b}\right)\varepsilon^2}\right), \quad (35)$$

*communication rounds and*

$$\mathcal{O}\left(\frac{\left(bL + \sqrt{L^2 Bc\delta mb + c\delta m^2 L_\pm^2 + \left(c\delta m^2 + \frac{mb}{G}\right)\frac{\mathcal{L}_\pm^2}{b}}\right)\Delta_0}{\left(1 - \frac{48Bc\delta m}{b}\right)\varepsilon^2}\right), \quad (36)$$

*oracle calls per worker.*

**Corollary E.3.** *Let the assumptions of Theorem E.1 hold, $p = \min\{b/m, 1/1+\omega\}$ and*

$$\gamma = \frac{1}{L + \sqrt{A}}, \quad \text{where}$$

$$A = 48L^2 Bc\delta \max\left\{\frac{m}{b}, 1 + \omega\right\}$$

$$+ 6\left(4c\delta \max\left\{\frac{m^2}{b^2}, (1 + \omega)^2\right\} + \frac{\max\left\{\frac{m}{b}, 1 + \omega\right\}}{2G}\right)\left(\omega L^2 + \frac{(1 + \omega)\mathcal{L}_\pm^2}{b}\right)$$

$$+ 6\left(4c\delta(1 + \omega)\max\left\{\frac{m^2}{b^2}, (1 + \omega)^2\right\} + \frac{\omega\max\left\{\frac{m}{b}, 1 + \omega\right\}}{2G}\right)L_\pm^2$$

*Then for all $K \geq 0$ we have $\mathbb{E}\left[\|\nabla f(\widehat{x}^K)\|^2\right]$ of the order*

$$\mathcal{O}\left(\frac{\left(L + \sqrt{A}\right)\Delta_0}{\left(1 - 48Bc\delta \max\left\{\frac{m}{b}, 1 + \omega\right\}\right)K} + \frac{c\delta\zeta^2}{\min\left\{\frac{b}{m}, \frac{1}{1+\omega}\right\} - 48Bc\delta}\right), \quad (37)$$

*where $\widehat{x}^K$ is choosen uniformly at random from the iterates $x^0, x^1, \ldots, x^K$ produced by* Byz-VR-MARINA *and $\Delta_0 = f(x^0) - f_*$. That is, to guarantee $\mathbb{E}\left[\|\nabla f(\widehat{x}^K)\|^2\right] \leq \varepsilon^2$ for $\varepsilon^2 \geq \frac{32c\delta\zeta^2}{p - 48Bc\delta}$* Byz-VR-MARINA *requires*

$$\mathcal{O}\left(\frac{\left(L + \sqrt{A}\right)\Delta_0}{\left(1 - 48Bc\delta \max\left\{\frac{m}{b}, 1 + \omega\right\}\right)\varepsilon^2}\right), \quad (38)$$

*communication rounds and*

$$\mathcal{O}\left(\frac{\left(bL + b\sqrt{A}\right)\Delta_0}{\left(1 - 48Bc\delta \max\left\{\frac{m}{b}, 1 + \omega\right\}\right)\varepsilon^2}\right), \quad (39)$$

*oracle calls per worker.*

**Corollary E.4** (Homogeneous data)**.** *Let the assumptions of Theorem E.1 hold, $p = \min\{b/m, 1/1+\omega\}$, $B = 0$, $\zeta = 0$, and*

$$\gamma = \frac{1}{L + \sqrt{A}}, \quad where$$

$$A = 6\left(3c\delta\max\left\{\frac{m^2}{b^2}, (1+\omega)^2\right\} + \frac{\max\left\{\frac{m}{b}, 1+\omega\right\}}{2G}\right)\left(\omega L^2 + \frac{(1+\omega)\mathcal{L}_\pm^2}{b}\right)$$

*Then for all $K \geq 0$ we have $\mathbb{E}\left[\|\nabla f(\widehat{x}^K)\|^2\right]$ of the order*

$$\mathcal{O}\left(\frac{\left(L + \sqrt{A}\right)\Delta_0}{K}\right), \tag{40}$$

*where $\widehat{x}^K$ is choosen uniformly at random from the iterates $x^0, x^1, \ldots, x^K$ produced by* Byz-VR-MARINA *and $\Delta_0 = f(x^0) - f_*$. That is, to guarantee $\mathbb{E}\left[\|\nabla f(\widehat{x}^K)\|^2\right] \leq \varepsilon^2$ for $\varepsilon^2 > 0$* Byz-VR-MARINA *requires*

$$\mathcal{O}\left(\frac{\left(L + \sqrt{A}\right)\Delta_0}{\varepsilon^2}\right), \tag{41}$$

*communication rounds and*

$$\mathcal{O}\left(\frac{\left(bL + b\sqrt{A}\right)\Delta_0}{\varepsilon^2}\right), \tag{42}$$

*oracle calls per worker.*

### E.4 FUNCTIONS SATISFYING POLYAK-ŁOJASIEWICZ CONDITION

**Theorem E.2** (Generalized version of Theorem 2.2)**.** *Let Assumptions 2.1, E.1, 2.3, 2.4, 2.5 hold. Assume that*

$$0 < \gamma \leq \min\left\{\frac{1}{L + \sqrt{2A}}, \frac{p}{4\mu\left(1 - \frac{96Bc\delta}{p}\right)}\right\}, \quad \delta < \frac{p}{96cB}, \tag{43}$$

*where $A = \frac{48BL^2c\delta}{p} + \frac{6(1-p)}{p}\left(\frac{4c\delta}{p} + \frac{1}{2G}\right)\left(\omega L^2 + \frac{(1+\omega)\mathcal{L}_\pm^2}{b}\right) + \frac{6(1-p)}{p}\left(\frac{4c\delta(1+\omega)}{p} + \frac{\omega}{2G}\right)L_\pm^2$. Then for all $K \geq 0$ the iterates produced by* Byz-VR-MARINA *satisfy*

$$\mathbb{E}\left[f(x^K) - f(x^*)\right] \leq \left(1 - \gamma\mu\left(1 - \frac{96Bc\delta}{p}\right)\right)^K \Phi_0 + \frac{24c\delta\zeta^2}{\mu(p - 96Bc\delta)}, \tag{44}$$

*where $\Phi_0 = f(x^0) - f(x^*) + \frac{2\gamma}{p}\|g^0 - \nabla f(x^0)\|^2$. The result of Theorem 2.1 is a special case of the statement above with $B = 0$, since for $B = 0$ we have $A = \frac{48BL^2c\delta}{p} + \frac{6(1-p)}{p}\left(\frac{4c\delta}{p} + \frac{1}{2G}\right)\left(\omega L^2 + \frac{(1+\omega)\mathcal{L}_\pm^2}{b}\right) + \frac{6(1-p)}{p}\left(\frac{4c\delta(1+\omega)}{p} + \frac{\omega}{2G}\right)L_\pm^2 = \frac{6(1-p)}{p}\left(\frac{4c\delta}{p} + \frac{1}{2G}\right)\left(\omega L^2 + \frac{(1+\omega)\mathcal{L}_\pm^2}{b}\right) + \frac{6(1-p)}{p}\left(\frac{4c\delta(1+\omega)}{p} + \frac{\omega}{2G}\right)L_\pm^2, \left(1 - \frac{96Bc\delta}{p}\right) = 1, p - 96Bc\delta = p$, and the second condition from (43) always holds.*

*Proof.* For all $k \geq 0$ we introduce $\Phi_k = f(x^k) - f_* + \frac{2\gamma}{p}\|g^k - \nabla f(x^k)\|^2$. Using the results of Lemmas E.1 and E.3, we derive

$$
\mathbb{E}[\Phi_{k+1}] \overset{(20),(23)}{\leq} \mathbb{E}\left[f(x^k) - f(x^*) - \left(\frac{1}{2\gamma} - \frac{L}{2}\right)\|x^{k+1} - x^k\|^2 + \frac{\gamma}{2}\|g^k - \nabla f(x^k)\|^2\right]
$$

$$
- \frac{\gamma}{2}\mathbb{E}\left[\|\nabla f(x^k)\|^2\right] + \frac{2\gamma}{p}\left(1 - \frac{p}{2}\right)\mathbb{E}\left[\|g^k - \nabla f(x^k)\|^2\right]
$$

$$
+ \frac{48Bc\delta\gamma}{p}\mathbb{E}\left[\|\nabla f(x^k)\|^2\right] + \frac{24c\delta\zeta^2\gamma}{p} + \gamma A\mathbb{E}\left[\|x^{k+1} - x^k\|^2\right]
$$

$$
= \mathbb{E}\left[f(x^k) - f(x^*)\right] + \frac{2\gamma}{p}\left(1 - \frac{p}{4}\right)\mathbb{E}\left[\|g^k - \nabla f(x^k)\|^2\right]
$$

$$
- \frac{\gamma}{2}\left(1 - \frac{96Bc\delta}{p}\right)\mathbb{E}\left[\|\nabla f(x^k)\|^2\right] + \frac{24c\delta\zeta^2\gamma}{p}
$$

$$
- \frac{1}{2\gamma}\left(1 - L\gamma - 2A\gamma^2\right)\mathbb{E}\left[\|x^{k+1} - x^k\|^2\right]
$$

$$
\overset{(8)}{\leq} \left(1 - \gamma\mu\left(1 - \frac{96Bc\delta}{p}\right)\right)\mathbb{E}\left[f(x^k) - f(x^*)\right]
$$

$$
+ \frac{2\gamma}{p}\left(1 - \frac{p}{4}\right)\mathbb{E}\left[\|g^k - \nabla f(x^k)\|^2\right] + \frac{24c\delta\zeta^2\gamma}{p}
$$

$$
\overset{(43)}{\leq} \left(1 - \gamma\mu\left(1 - \frac{96Bc\delta}{p}\right)\right)\mathbb{E}[\Phi_k] + \frac{24c\delta\zeta^2\gamma}{p}
$$

where in the last step we use Lemma C.1 and our choice of $\gamma$ from (43). Unrolling the recurrence, we obtain

$$
\mathbb{E}[\Phi_K] \leq \left(1 - \gamma\mu\left(1 - \frac{96Bc\delta}{p}\right)\right)^K\mathbb{E}[\Phi_0] + \frac{24c\delta\zeta^2\gamma}{p}\sum_{k=0}^{K-1}\left(1 - \gamma\mu\left(1 - \frac{96Bc\delta}{p}\right)\right)^k
$$

$$
\leq \left(1 - \gamma\mu\left(1 - \frac{96Bc\delta}{p}\right)\right)^K\mathbb{E}[\Phi_0] + \frac{24c\delta\zeta^2\gamma}{p}\sum_{k=0}^{\infty}\left(1 - \gamma\mu\left(1 - \frac{96Bc\delta}{p}\right)\right)^k
$$

$$
= \left(1 - \gamma\mu\left(1 - \frac{96Bc\delta}{p}\right)\right)^K\mathbb{E}[\Phi_0] + \frac{24c\delta\zeta^2}{\mu(p - 96Bc\delta)}.
$$

Taking into account $\Phi_k \geq f(x^k) - f(x^*)$, we get the result. $\qquad\square$

As in the case of general non-convex smooth functions, we need to estimate $\Phi_0$ to derive complexity results. Following exactly the same reasoning as in the derivation of (30), we get

$$
\mathbb{E}[\Phi_0] \leq 2\left(f(x^0) - f(x^*)\right) + \frac{8\gamma c\delta\zeta^2}{p}.
$$

Plugging this upper bound in (44), we get

$$
\mathbb{E}\left[f(x^K) - f(x^*)\right] \leq 2\left(1 - \gamma\mu\left(1 - \frac{96Bc\delta}{p}\right)\right)^K\left(f(x^0) - f(x^*)\right)
$$

$$
+ \left(1 - \gamma\mu\left(1 - \frac{96Bc\delta}{p}\right)\right)^K \cdot \frac{8\gamma c\delta\zeta^2}{p} + \frac{24c\delta\zeta^2}{\mu(p - 96Bc\delta)}
$$

$$
\leq 2\left(1 - \gamma\mu\left(1 - \frac{96Bc\delta}{p}\right)\right)^K\left(f(x^0) - f(x^*)\right)
$$

$$
+ \sum_{k=0}^{\infty}\left(1 - \gamma\mu\left(1 - \frac{96Bc\delta}{p}\right)\right)^k \cdot \frac{8\gamma c\delta\zeta^2}{p} + \frac{24c\delta\zeta^2}{\mu(p - 96Bc\delta)}
$$

$$
\leq 2\left(1 - \gamma\mu\left(1 - \frac{96Bc\delta}{p}\right)\right)^K\left(f(x^0) - f(x^*)\right) + \frac{32c\delta\zeta^2}{\mu(p - 96Bc\delta)}.
$$

Based on this inequality we derive following corollaries.

**Corollary E.5** (Homogeneous data, no compression ($\omega = 0$))**.** *Let the assumptions of Theorem E.2 hold, $\mathcal{Q}(x) \equiv x$ for all $x \in \mathbb{R}^d$ (no compression, $\omega = 0$), $p = b/m$, $B = 0$, $\zeta = 0$, and*

$$\gamma = \min \left\{ \frac{1}{L + 2\mathcal{L}_\pm \sqrt{\frac{12c\delta m^2}{b^3} + \frac{3m}{2b^2 G}}}, \frac{b}{4m\mu} \right\}.$$

*Then for all $K \geq 0$ we have $\mathbb{E}\left[ f(x^K) - f(x^*) \right]$ of the order*

$$\mathcal{O}\left( \exp\left( -\min\left\{ \frac{\mu}{L + 2\mathcal{L}_\pm \sqrt{\frac{12c\delta m^2}{b^3} + \frac{3m}{2b^2 G}}}, \frac{b}{m} \right\} K \right) \Delta_0 \right), \tag{45}$$

*where $\Delta_0 = f(x^0) - f(x^*)$. That is, to guarantee $\mathbb{E}\left[ f(x^K) - f(x^*) \right] \leq \varepsilon$ for $\varepsilon > 0$* Byz-VR-MARINA *requires*

$$\mathcal{O}\left( \max\left\{ \frac{L + \mathcal{L}_\pm \sqrt{\frac{c\delta m^2}{b^3} + \frac{m}{b^2 G}}}{\mu}, \frac{m}{b} \right\} \log \frac{\Delta_0}{\varepsilon} \right), \tag{46}$$

*communication rounds and*

$$\mathcal{O}\left( \max\left\{ \frac{bL + \mathcal{L}_\pm \sqrt{\frac{c\delta m^2}{b} + \frac{m}{G}}}{\mu}, m \right\} \log \frac{\Delta_0}{\varepsilon} \right), \tag{47}$$

*oracle calls per worker.*

**Corollary E.6** (No compression ($\omega = 0$))**.** *Let the assumptions of Theorem E.2 hold, $\mathcal{Q}(x) \equiv x$ for all $x \in \mathbb{R}^d$ (no compression, $\omega = 0$), $p = b/m$ and*

$$\gamma = \min\left\{ \frac{1}{L + \sqrt{\frac{96L^2 Bc\delta m}{b} + \frac{48c\delta m^2}{b^2} L_\pm^2 + 12\left( \frac{4c\delta m^2}{b^2} + \frac{m}{2bG} \right) \frac{\mathcal{L}_\pm^2}{b}}}, \frac{\frac{b}{m}}{4\mu\left(1 - 96Bc\delta\frac{m}{b}\right)} \right\}.$$

*Then for all $K \geq 0$ we have $\mathbb{E}\left[ f(x^K) - f(x^*) \right]$ of the order*

$$\mathcal{O}\Bigg( \exp\Bigg( -\min\Bigg\{ \frac{\mu\left(1 - 96Bc\delta\frac{m}{b}\right)}{L + \sqrt{\frac{96L^2 Bc\delta m}{b} + \frac{48c\delta m^2}{b^2} L_\pm^2 + 12\left(\frac{4c\delta m^2}{b^2} + \frac{m}{2bG}\right)\frac{\mathcal{L}_\pm^2}{b}}}, \frac{b}{m} \Bigg\} K \Bigg) \Delta_0$$

$$+ \frac{c\delta\zeta^2}{\mu\left(\frac{b}{m} - 96Bc\delta\right)} \Bigg), \tag{48}$$

*where $\Delta_0 = f(x^0) - f(x^*)$. That is, to guarantee $\mathbb{E}\left[ f(x^K) - f(x^*) \right] \leq \varepsilon$ for $\varepsilon \geq \frac{32c\delta\zeta^2}{\mu(p - 96Bc\delta)}$* Byz-VR-MARINA *requires*

$$\mathcal{O}\left( \max\left\{ \frac{L + \sqrt{\frac{L^2 Bc\delta m}{b} + \frac{c\delta m^2}{b^2} L_\pm^2 + \left(\frac{c\delta m^2}{b^2} + \frac{m}{bG}\right)\frac{\mathcal{L}_\pm^2}{b}}}{\mu\left(1 - 96Bc\delta\frac{m}{b}\right)}, \frac{m}{b} \right\} \log \frac{\Delta_0}{\varepsilon} \right), \tag{49}$$

*communication rounds and*

$$\mathcal{O}\left( \max\left\{ \frac{bL + \sqrt{L^2 Bc\delta mb + c\delta m^2 L_\pm^2 + \left(c\delta m^2 + \frac{mb}{G}\right)\frac{\mathcal{L}_\pm^2}{b}}}{\mu\left(1 - 96Bc\delta\frac{m}{b}\right)}, m \right\} \log \frac{\Delta_0}{\varepsilon} \right), \tag{50}$$

*oracle calls per worker.*

**Corollary E.7.** *Let the assumptions of Theorem E.2 hold, $p = \min\{b/m, 1/1+\omega\}$ and*

$$\gamma \;=\; \min\left\{\frac{1}{L+\sqrt{2A}}, \frac{\min\left\{\frac{b}{m}, \frac{1}{1+\omega}\right\}}{4\mu\left(1 - 96Bc\delta\max\left\{\frac{m}{b}, 1+\omega\right\}\right)}\right\}, \quad where$$

$$\begin{aligned}
A \;=\;& 48L^2 Bc\delta\max\left\{\frac{m}{b}, 1+\omega\right\} \\
& +6\left(4c\delta\max\left\{\frac{m^2}{b^2}, (1+\omega)^2\right\} + \frac{\max\left\{\frac{m}{b}, 1+\omega\right\}}{2G}\right)\left(\omega L^2 + \frac{(1+\omega)\mathcal{L}_\pm^2}{b}\right) \\
& +6\left(4c\delta(1+\omega)\max\left\{\frac{m^2}{b^2}, (1+\omega)^2\right\} + \frac{\omega\max\left\{\frac{m}{b}, 1+\omega\right\}}{2G}\right)L_\pm^2
\end{aligned}$$

*Then for all $K \geq 0$ we have $\mathbb{E}\left[f(x^K) - f(x^*)\right]$ of the order*

$$\mathcal{O}\Bigg(\exp\left(-\min\left\{\frac{\mu\left(1 - 96Bc\delta\max\left\{\frac{m}{b}, 1+\omega\right\}\right)}{L+\sqrt{A}}, \frac{b}{m}, \frac{1}{1+\omega}\right\}K\right)\Delta_0$$

$$+\frac{c\delta\zeta^2}{\mu\left(\min\left\{\frac{b}{m}, \frac{1}{1+\omega}\right\}-96Bc\delta\right)}\Bigg), \qquad (51)$$

*where $\Delta_0 = f(x^0) - f(x^*)$. That is, to guarantee $\mathbb{E}\left[f(x^K) - f(x^*)\right] \leq \varepsilon$ for $\varepsilon \geq \frac{32c\delta\zeta^2}{\mu(p-96Bc\delta)}$*
Byz-VR-MARINA *requires*

$$\mathcal{O}\left(\max\left\{\frac{L+\sqrt{A}}{\mu\left(1 - 96Bc\delta\max\left\{\frac{m}{b}, 1+\omega\right\}\right)}, \frac{m}{b}, 1+\omega\right\}\log\frac{\Delta_0}{\varepsilon}\right), \qquad (52)$$

*communication rounds and*

$$\mathcal{O}\left(\max\left\{\frac{bL+b\sqrt{A}}{\mu\left(1 - 96Bc\delta\max\left\{\frac{m}{b}, 1+\omega\right\}\right)}, m, b(1+\omega)\right\}\log\frac{\Delta_0}{\varepsilon}\right), \qquad (53)$$

*oracle calls per worker.*

**Corollary E.8** (Homogeneous data)**.** *Let the assumptions of Theorem E.2 hold, $p = \min\{b/m, 1/1+\omega\}$, $B = 0$, $\zeta = 0$, and*

$$\gamma \;=\; \min\left\{\frac{1}{L+\sqrt{2A}}, \frac{\min\left\{\frac{b}{m}, \frac{1}{1+\omega}\right\}}{4\mu}\right\}, \quad where$$

$$A \;=\; 6\left(4c\delta\max\left\{\frac{m^2}{b^2}, (1+\omega)^2\right\} + \frac{\max\left\{\frac{m}{b}, 1+\omega\right\}}{2G}\right)\left(\omega L^2 + \frac{(1+\omega)\mathcal{L}_\pm^2}{b}\right).$$

*Then for all $K \geq 0$ we have $\mathbb{E}\left[f(x^K) - f(x^*)\right]$ of the order*

$$\mathcal{O}\left(\exp\left(-\min\left\{\frac{\mu}{L+\sqrt{A}}, \frac{b}{m}, \frac{1}{1+\omega}\right\}K\right)\Delta_0\right), \qquad (54)$$

*where $\Delta_0 = f(x^0) - f(x^*)$. That is, to guarantee $\mathbb{E}\left[f(x^K) - f(x^*)\right] \leq \varepsilon$ for $\varepsilon > 0$* Byz-VR-MARINA *requires*

$$\mathcal{O}\left(\max\left\{\frac{L+\sqrt{A}}{\mu}, \frac{m}{b}, 1+\omega\right\}\log\frac{\Delta_0}{\varepsilon}\right), \qquad (55)$$

*communication rounds and*

$$\mathcal{O}\left(\max\left\{\frac{bL+b\sqrt{A}}{\mu}, m, b(1+\omega)\right\}\log\frac{\Delta_0}{\varepsilon}\right), \qquad (56)$$

*oracle calls per worker.*

### E.5 Further Details on the Obtained Results and the Comparison from Table 2

In this part, we discuss additional details about the obtained results and on the comparison of methods complexities given in Table 2. We mostly focus on the results for general smooth non-convex functions. Similar observations are valid for smooth PŁfunctions as well.

**Comparison of the assumptions on the stochastic gradient noise.** Many existing works rely on the uniformly bounded variance assumption (UBV): it is assumed that for all $x \in \mathbb{R}^d$ the good workers have an access to the unbiased estimators $g_i(x)$ of $\nabla f_i(x)$ such that $\mathbb{E}\|g_i(x) - \nabla f_i(x)\|^2 \leq \sigma^2$ for all $i \in \mathcal{G}$ and $\sigma \geq 0$. This assumption does not hold in many practical situations and even for simple convex finite-sum problems like sums of quadratic functions with non-identical Hessians. Moreover, in the situations when this assumption holds, the value of $\sigma^2$ can be huge. However, UBV assumption does not require individual stochastic realizations, i.e., summands $f_{i,j}$, to be smooth.

In contrast, we use Assumption 2.4 that holds for many situations when UBV assumption does not. For example, Assumption 2.4 holds whenever all functions $f_{i,j}$, $i \in \mathcal{G}$, $j \in m$ are $L_{i,j}$-smooth (see Appendix E.1). These facts allow us to cover a large class of problems that does not fit the setup considered in (Karimireddy et al., 2021; 2022; Gorbunov et al., 2021a). Moreover, since Assumption 2.4 is more general than smoothness of all $f_{i,j}$, our analysis covers the setup considered in (Wu et al., 2020; Zhu & Ling, 2021). However, it is worth mentioning that there exist problems such that UBV assumption holds and Assumption 2.4 does not, e.g., when the gradient noise is additive: $\widehat{\Delta}_i(x, y) = \nabla f_i(x) - \nabla f_i(y) + \xi_i$, where $\mathbb{E}\xi_i = 0$ and $\mathbb{E}\|\xi_i\|^2 = \sigma^2$.

**On the choice of $p$.** Our analysis is valid for any choice of $p \in (0, 1]$. As we explain in footnote 2, the choice of $p = \min\{b/m, 1/(1+\omega)\}$ leads to the fair comparison with other results, since this choice implies that the total expected (communication and oracle) cost of steps with full gradients computations/uncompressed communications coincides with the total cost of the rest of iterations. Indeed, to measure the communication efficiency one can use expected density $\zeta_{\mathcal{Q}}$ (see Definition 2.2). Then, the expected number of components that each worker sends to the server at each step is upper-bounded by $\zeta_{\mathcal{Q}}(1 - p) + pd$ meaning that $p = \zeta_{\mathcal{Q}}/d$ makes the expected number of components that each worker sends to the server at each step equal to $\mathcal{O}(\zeta_{\mathcal{Q}})$. In the case when $1 + \omega = \Theta(d/\zeta_{\mathcal{Q}})$ (which is the case for RandK sparsification and $\ell_2$-quantization, see (Beznosikov et al., 2020)), one can choose $p = 1/(1+\omega)$.

On the other hand, the expected number of oracle calls per iteration is $2b(1 - p) + mp$ meaning that $p = b/m$ makes the expected oracle cost of each iteration equal to $\mathcal{O}(b)$ like in the case of SGD. This means that the best $p$ for oracle complexity and the best $p$ for communication efficiency are different in general. When $b/m < 1/(1+\omega)$, the choice $p = \min\{b/m, 1/(1+\omega)\}$ implies that the algorithm could use uncompressed vectors more often without sacrificing the communication cost, and when $b/m > 1/(1+\omega)$, the choice $p = \min\{b/m, 1/(1 + \omega)\}$ implies that the algorithm could use full gradients more often without sacrificing the oracle cost. That is, the choice of $p \approx b/m$ implies better oracle complexity and the choice $p \approx 1/(1+\omega)$ leads to better communication efficiency. Depending on how much these two aspects are important for the particular application, one can choose $p$ in between these two values.

**The effect of compression.** For simplicity, consider a homogeneous case ($B = 0$, $\zeta = 0$) and let $b = 1$; similar arguments are valid for the general case. As Corollary E.4 states, the communication complexity[6] of Byz-VR-MARINA in this case equals

$$\mathcal{O}\left(\frac{\left(L + \sqrt{A}\right)\Delta_0}{\varepsilon^2}\right), \quad \text{where}$$

$$A = 6\left(3c\delta \max\left\{m^2, (1+\omega)^2\right\} + \frac{\max\left\{m, 1+\omega\right\}}{2G}\right)\left(\omega L^2 + (1+\omega)\mathcal{L}_{\pm}^2\right).$$

The above result shows that the communication complexity becomes worse with the growth of $\omega$. Larger $\omega$ means that the compression $\mathcal{Q}$ is more loose, i.e., less information is communicated. This is

---

[6]We remind here that by communication complexity we mean the total number of communication rounds needed for the algorithm to find point $x$ such that $\mathbb{E}\|\nabla f(x)\|^2 \leq \varepsilon^2$.

a common phenomenon for the methods with communication compression (Horváth et al., 2019b; Gorbunov et al., 2021b). However, when the compression is not severe, e.g., $1 + \omega \leq m$, then the complexity bound becomes

$$\mathcal{O}\left(\frac{\left(L + \sqrt{A}\right)\Delta_0}{\varepsilon^2}\right), \quad \text{where} \quad A = 6\left(3c\delta m^2 + \frac{m}{2G}\right)\left(\omega L^2 + (1+\omega)\mathcal{L}_\pm^2\right),$$

which is worse only $\mathcal{O}(\sqrt{\omega})$ times than the complexity of Byz-VR-MARINA without compression, while the number of communicated bits/components becomes $\mathcal{O}(d/\varsigma_{\mathcal{Q}})$ times smaller. For example, in the case of RandK sparsification we have $1 + \omega = d/K = d/\varsigma_{\mathcal{Q}}$, and $1 + \omega \leq m$ allows to have quite strong compression, e.g., for $m \geq 1000$, meaning that the dataset has at least 1000 samples, inequality $1 + \omega \leq m$ implies that workers can send just $0.1\%$ of information. In this case, the communication cost of each iteration becomes $\sim 1000$ times cheaper, while the number of communication rounds increases only $\sqrt{1 + \omega} \sim 30$ times. If the communication is the bottleneck, then the algorithm will converge much faster with compression than without it in this setup.

**On the batchsizes.** First, we note that our analysis is valid for any choice of $b \geq 1$. For simplicity of the further discussion of the batchsizes role in the complexities, consider the homogeneous case ($B = 0$, $\zeta = 0$) without compression ($\omega = 0$). As Corollary E.1 states, the communication complexity of Byz-VR-MARINA in this case equals

$$\mathcal{O}\left(\frac{\left(L + \mathcal{L}_\pm\sqrt{\frac{c\delta m^2}{b^3} + \frac{m}{b^2 G}}\right)\Delta_0}{\varepsilon^2}\right).$$

Note that the term depending on the ratio of Byzantine workers $\delta$ scales as $b^{-3/2}$ with the batchsize and the term depending on $1/G$ scales as $b^{-1}$. Table 2 illustrates that previous SOTA results in this case scale as $b^{-1}$ or $b^{-1/2}$, so, the complexity bound for Byz-VR-MARINA scales with $b$ no worse than the concurrent bounds.

Next, typically, there is no need to take $b$ larger than $\sqrt{m}$ for SARAH-based variance reduced methods (Horváth et al., 2022; Li et al., 2021): oracle complexity is always the same (neglecting the differences in the smoothness constants), while the iteration complexity stops improving once $b$ becomes larger than $\sqrt{m}$. However, the complexity bound for Byz-VR-MARINA contains the non-standard term $\mathcal{O}\left(\mathcal{L}_\pm m\sqrt{c\delta}/\sqrt{b^3}\varepsilon^2\right)$ appearing due to the presence of Byzantine workers. For simplicity, we assume that $L = \Theta(\mathcal{L}_\pm)$ (though $\mathcal{L}_\pm$ can be both smaller and larger than $L$). Then, when we increase batchsize $b$, the communication complexity stops improving once $b$ becomes larger than $\max\{\sqrt[3]{c\delta m^2}, m\}$. Interestingly, $\sqrt[3]{c\delta m^2}$ can be larger than the standard value $\sqrt{m}$: this is the case when $m > 1/c^2\delta^2$. In this case, the communication complexity of Byz-VR-MARINA benefits from the slightly larger batchsizes than in the classical case. This phenomenon has a natural explanation: when we increase the batchsize, the variance of the gradient noise decreases and it becomes even harder for Byzantine workers to shift the updates of the method significantly.

