# OpenReview forum: "Variance Reduction is an Antidote to Byzantines: Better Rates, Weaker Assumptions and Communication Compression as a Cherry on the Top"
_ICLR.cc/2023/Conference — ICLR 2023 poster_

### Official Review · Reviewer_9mHu · 2022-10-22

**Confidence:** 4
**Correctness:** 3
**Technical Novelty And Significance:** 2
**Empirical Novelty And Significance:** 2
**Recommendation:** 5

**Clarity, Quality, Novelty And Reproducibility:**

Most parts of this paper are clearly written. However, the proof details are ambiguous. The novelty is limited. Please see the concerns above for details.

**Strength And Weaknesses:**

The main idea of variance reduction is well presented. I appreciate that theoretical analysis is provided for both non-convex objective functions and functions satisfying PL inequality. Meanwhile, there are also some concerns.

1. My major concern is about the proof details. The expectation notation $\mathbb{E}$ is used ambiguously in the proof details in the Appendix. Specifically, the randomness comes from both the stochastic mapping $\mathcal{Q}$ and the mini-batched estimator $\hat{\Delta}_i$. However, it is ambiguous which stochastic operator each expectation is with respect to. It increases the difficulty of checking the correctness of the proof details and would make readers confused.

2. It is claimed that communication compression is a bonus of the proposed method. However, there seems no theoretical analysis or empirical results about the communication efficiency in the paper.

3. The proposed method is very similar to VR-MARINA [1], which limits the novelty of this paper.

4. The experiments are conducted with the logistic regression model on a9a and w8a datasets. The scale of the problem is small given the current computing power and the background of distributed learning. Adding experiments on larger-scale problems would make this work more solid and convincing.

5. (minor) It seems that only the proofs for the generalized version of Theorem 2.1 and 2.2 are provided in the Appendix. It is suggested to add some details in the Appendix about how to obtain the final results when $B=0$ in Assumption E.1.

6. (minor) I am wondering why the hyperparameter setting for bucketing is presented in the paragraph that introduces the attack method IPM (page 9).

[1] Gorbunov, Eduard, et al. "MARINA: Faster non-convex distributed learning with compression." International Conference on Machine Learning. PMLR, 2021.

**Summary Of The Paper:**

In this paper, the authors propose a new Byzantine-tolerant distributed learning method called Byz-VR-MARINA, which adopts variance reduction to obtain a better convergence rate than existing methods. Meanwhile, communication compression is a bonus of the proposed method. The authors provide theoretical analysis of Byz-VR-MARINA for non-convex objective functions and the functions satisfying Polyak-Lojasiewicz inequality. Besides, empirical results of Byz-VR-MARINA and the baselines on binary classification tasks are provided.

**Summary Of The Review:**

Given the concerns above, this paper seems currently below the acceptance threshold to me. Meanwhile, I am willing to raise my rating if the authors could properly address the concerns.

---

> ### Author Response · Authors · 2022-11-11
> **Authors' response to Reviewer 9mHU [Part 2/2]**
>
> > **The proposed method is very similar to VR-MARINA [1], which limits the novelty of this paper.**
>
> We believe that it is not always an easy task to combine known techniques to get some improvements in the convergence rates. Some techniques cannot be combined in general: for example, it is known that gradient descent is optimal for finding first-order stationary points in the non-convex smooth case among first-order methods [1]. Therefore, it is in general impossible to combine it with Nesterov’s acceleration to achieve theoretically better rate. Next, even when the techniques are combinable, it can be a non-trivial task to propose good combination and to rigorously analyze it. For example, Nesterov’s acceleration was discovered in 1980-th and SGD was proposed in 1950-th, but the first Accelerated SGD was proposed and analyzed only in 2012 [2].
>
> Moreover, in paragraph “Challenges in designing variance-reduced algorithm with tight rates and provable Byzantine-robustness” (page 7), we explain why it is not trivial to achieve the results that we derive. Although our work positions Byz-VR-MARINA as a natural combination of variance reduction and Byzantine-robustness, it was not straightforward beforehand whether VR-MARINA and robust aggregation are combinable, whether such a combination should be considered, and whether it would lead to the new SOTA theoretical results in Byzantine-robust distributed optimization.
>
> References:
>
> [1] Arjevani, Y., Carmon, Y., Duchi, J. C., Foster, D. J., Srebro, N., & Woodworth, B. (2022). Lower bounds for non-convex stochastic optimization. Mathematical Programming, 1-50.
>
> [2] Ghadimi, S., & Lan, G. (2012). Optimal stochastic approximation algorithms for strongly convex stochastic composite optimization i: A generic algorithmic framework. SIAM Journal on Optimization, 22(4), 1469-1492.
>
> ---
>
> > **The experiments are conducted with the logistic regression model on a9a and w8a datasets. The scale of the problem is small given the current computing power and the background of distributed learning. Adding experiments on larger-scale problems would make this work more solid and convincing.**
>
> We thank the reviewer for the suggestion. We are currently running additional experiments on larger datasets with a larger number of features.
>
> We also would like to note here that our paper focuses on the theoretical convergence guarantees and achieving new theoretical SOTA results in such a popular field as Byzantine-tolerant training is important. We note that all the provided experiments are well aligned with the theoretical results and do not undermine it.
>
> ---
>
> > **It seems that only the proofs for the generalized version of Theorem 2.1 and 2.2 are provided in the Appendix. It is suggested to add some details in the Appendix about how to obtain the final results when $B = 0$ in Assumption E.1.**
>
> To get the results of Theorems 2.1 and 2.2 one needs to set up $B = 0$ in Theorems E.1 and E.2. We have added the details on the derivation of Theorems 2.1 and 2.2 from their generalized versions.

---

> ### Author Response · Authors · 2022-11-11
> **Authors' response to Reviewer 9mHU [Part 1/2]**
>
> > **My major concern is about the proof details. The expectation notation $\mathbb{E}$ is used ambiguously in the proof details in the Appendix. Specifically, the randomness comes from both the stochastic mapping $\mathcal{Q}$ and the mini-batched estimator $\hat{\Delta}_i$. However, it is ambiguous which stochastic operator each expectation is with respect to. It increases the difficulty of checking the correctness of the proof details and would make readers confused.**
>
> If the opposite is not specified, by $\mathbb{E}[\cdot]$ in our results and proofs we always mean the full expectation, i.e., the expectation w.r.t. the all randomness. We have added this clarification to the text (see footnote 2). We have also clarified a few places in the proofs where we used the tower property of the expectation that might confuse the reader.
>
> ---
>
> > **It is claimed that communication compression is a bonus of the proposed method. However, there seems no theoretical analysis or empirical results about the communication efficiency in the paper.**
>
> We have added the discussion about the communication efficiency of the compressed version of our algorithm to Appendix E.5. Below we also provide this discussion for the reviewer’s convenience.
>
> For simplicity, consider a homogeneous case ($B = 0$, $\zeta = 0$) and let $b = 1$; similar arguments are valid for the general case. As Corollary E.4 states, the communication complexity (the total number of communication rounds needed for the algorithm to find point $x$ such that $\mathbb{E}\|\nabla f(x)\|^2 \leq \varepsilon^2$) of Byz-VR-MARINA in this case equals
>
> $$
> 	\mathcal{O}\left(\frac{\left(L + \sqrt{A}\right)\Delta_0}{\varepsilon^2}\right),\quad \text{where}
> $$
> $$
> 	A = 6\left(3c\delta\max\left[m^2, (1+\omega)^2\right] + \frac{\max\left[m, 1+\omega\right]}{2G}\right)\left(\omega L^2 + (1+\omega){\mathcal{L}}_{\pm}^2\right).
> $$
>
> The above result shows that the communication complexity becomes worse with the growth of $\omega$. Larger $\omega$ means that the compression $\mathcal{Q}$ is more loose, i.e., less information is communicated. This is a common phenomenon for the methods with communication compression [1,2]. However, when the compression is not severe, e.g., $1+\omega \leq m$, then the complexity bound becomes
> $$
> \mathcal{O}\left(\frac{\left(L + \sqrt{A}\right)\Delta_0}{\varepsilon^2}\right),\quad \text{where}\quad A = 6\left(3c\delta m^2 + \frac{m}{2G}\right)\left(\omega L^2 + (1+\omega){\mathcal{L}}_{\pm}^2\right),
> $$
>
> which is worse only $\mathcal{O}(\sqrt{\omega})$ times than the complexity of Byz-VR-MARINA without compression, while the number of communicated bits/components becomes $\mathcal{O}(d/\zeta_{\mathcal{Q}})$ times smaller. For example, in the case of RandK sparsification we have $1+\omega = \frac{d}{K} = \frac{d}{\zeta_{\mathcal{Q}}}$, and $1+\omega \leq m$ allows to have quite strong compression, e.g., for $m \geq 1000$, meaning that the dataset has at least $1000$ samples, inequality $1+\omega \leq m$ implies that workers can send just $0.1\%$ of information. In this case, the communication cost of each iteration becomes $\sim 1000$ times cheaper, while the number of communication rounds increases only $\sqrt{1+\omega} \sim 30$ times. If the communication is the bottleneck, then the algorithm will converge much faster with compression than without it in this setup.
>
>
> We are also working on the extra experiments right now. In particular, we will add the comparison in terms of the number of transmitted bits (and compare with the methods without the compression), which is a hardware/system and software independent metric of communication efficiency.
>
>
> References:
>
> [1] Horváth, S., Kovalev, D., Mishchenko, K., Richtárik, P., & Stich, S. (2022). Stochastic distributed learning with gradient quantization and double-variance reduction. Optimization Methods and Software, 1-16.
>
> [2] Mishchenko, K., Gorbunov, E., Takáč, M., & Richtárik, P. (2019). Distributed learning with compressed gradient differences. arXiv preprint arXiv:1901.09269.

---

> ### Author Response · Authors · 2022-11-17
> **Additional experiment: Compression vs No Compression**
>
> As promised, we include a comparison of methods with and without compression regarding the number of transmitted bits. The setup is the same as in Figure 2 in the manuscript.  As theory suggests, and also can be seen from the figures provided in the manuscript (Figures 1, 3 and 5), employing communication compression increases the number of iterations required to converge to the same quality solution. However, in the provided figure (https://ibb.co/xS0wwvR), we show that the compression speeds up the training (in terms of the number of transmitted bits).

---

### Official Review · Reviewer_xAhq · 2022-10-23

**Confidence:** 4
**Correctness:** 3
**Technical Novelty And Significance:** 3
**Empirical Novelty And Significance:** 3
**Recommendation:** 8

**Clarity, Quality, Novelty And Reproducibility:**

Starting with the title, the paper uses an offensive term that the Association for Computing Machinery recommends be avoided in professional writing and presentations: https://www.acm.org/diversity-inclusion/words-matter

The authors made an effort to clarify their use of language and make it less ambiguous and offensive, They shortened the pejorative term to three letters, while refusing to use any of the objective alternatives suggested. They also refused to provide clear technical definitions that would not rely on a medieval prejudice in order to convey a technical scientific meaning in the 21st century.

Thereby, the authors insisted on using the essence of the pejorative term. The three-letter term offered (at times interchangeably with terms "bad" and "malicious") does not make things clearer. The kind of workers envisaged in the work are only described by reference to previous work, but their properties are not made clear and are not defined technically. The only definition given is the following on Page 1:

"there is a high risk that some participants deviate from the prescribed protocol either on purpose or not."

Thereafter, despite this formulation and disclaimer about the motives of the workers in question, the workers are still characterized interchangeably as "bad" and "malicious". The short definition given is scientifically insufficient and does include all the intended technical connotations of the term, as the authors also acknowledged. Nevertheless, even this definition, short and technically insufficient as it may be, lends itself to a neutral technical term. That is the term "deviant". The authors declined using such a technical term and insisted that the paper somehow required using a derogatory stereotype that ACM recommends avoiding, shortened to three letters.

Yet such terminology is technically unclear and ambiguous. Alluding to a pejorative dictionary definition of an ethnoreligious exonym, distorting even the way Lamport used it in 1982, does not make the term technically clearer. The claim that it does so is scientifically dubious. These authors seem to believe that science should use derogatory stereotypes presented as technical terms to move forward. This reviewer disagrees and recommends following the advice of ACM on the matter.

**Details Of Ethics Concerns:**

Starting with the title, the paper uses an offensive term that the Association for Computing Machinery recommends be avoided in professional writing and presentations: https://www.acm.org/diversity-inclusion/words-matter

Despite the progress that has been made on this front over the last few years, it occasionally appears that, unfortunately, some people believe that it is necessary to explicitly and literally invoke the practice of slavery to describe a property of a distributed system architecture. Such people believe that the term "m_ster-sl_ve" is inherently better than several excellent, more respectful, more objective, more precise, and more meaningful alternatives proposed, and that those alternatives are, to quote the way some people usually put it, "ridiculous". In a similar fashion, as the response of the authors suggests, it seems there are people who find it necessary to explicitly and recognizably invoke an offensive and disrespectful enduring cultural prejudice of medieval European origin to describe a property of a distributed system or some of its constituent parts. According to those people, none of the reasonable, respectful, objective, and more meaningful alternatives proposed are good enough to describe those properties. Moreover, such authors do so in a convoluted manner that lacks scientific rigor, as they do not distinguish between the description of a system as a whole and the description of its faulty or deviant constituent parts. They refuse to provide a technically more precise definition, claiming that the prejudice-inspired term conveys the technical meaning in the best possible way. In the view of this reviewer, this practice is questionable from both a scientific and an ethical viewpoint, as it appears to denote a lack of scientific rigor as well as a lack of cultural sensitivity.

This paper uses the derogatory exonym "Byzantines", in its noun form, according to a pejorative meaning found in English dictionaries, to denote malicious or otherwise deviant workers, to whom an "antidote" should be given. The derogatory meaning of this English exonym reflects a centuries-old cultural and ethnoreligious prejudice (see, for example, https://hortulus-journal.com/journal/volume-13-2-2017/johnson/). This term was chosen in 1982 (see https://www.microsoft.com/en-us/research/publication/byzantine-generals-problem/, which describes the term as "obviously more appropriate", alluding to the dictionary definition that reflects the prejudice), before the ICLR Code of Ethics was written. However, even then, it was used to describe a set of all workers, rather than a subset of deviant, deceptive, demonic, malicious, devious, dishonest, or deceitful workers in the context of the properly named interactive consistency problem. The English exonym signifies the cultural heritage and identity of millions of people, including underrepresented and persecuted minorities. For example, it signifies members of the Eastern Catholic Church hailing from Eastern Europe (e.g., Albania, Romania, Slovakia, Ukraine, and elsewhere), who ecclesiastically identify themselves as "Byzantines" (see the usage at https://www.byzcath.org/index.php/about-us-mainmenu-60/about-byzantines-mainmenu-62; see also, for example, https://en.wikipedia.org/wiki/Albanian_Greek_Catholic_Church, https://www.wikiwand.com/en/Slovak_Greek_Catholic_Church, etc.), members of the persecuted Rumlar/Byzantine community of Istanbul (formerly Byzantium, see, for example, https://en.wikipedia.org/wiki/Greeks_in_Turkey#Current_situation, https://en.wikipedia.org/wiki/Istanbul_pogrom), and others, and it is used as a cultural signifier interchangeably with the English exonym "Greek" (see, for example, https://en.wikipedia.org/w/index.php?title=Byzantine_scholars_in_Renaissance&redirect=no). Using a signifier for all these people and their cultural heritage in its pejorative and prejudiced sense, as a synonym for "bad" and "malicious", to which an "antidote" needs to be given, as this paper does starting with such a call in the title, violates the standards of respectful, tolerant, inclusive, and equitable use of language that our community is trying to establish and the ICLR Code of Ethics advises to abide by. Therefore ACM recommends that this term be avoided in professional writing and presentations.

Unfortuntely, in their response, the authors expressed the view that using the pejorative and derogatory dictionary definition of an ethnoreligious term reflecting a medieval prejudice as scientific terminology was not self-evidently offensive. They refused to use scientifically neutral, objecgtive, and respectful alternative terms that carry the intended meaning with technical precision and may also be defined accordingly, such as, for example, deviant, deceptive, devious, deceitful, dishonest, demonic, treacherous, labyrinthine, etc. This stance deviates from the ICLR Code of Ethics, which explicitly advises respecting the cultural heritage of others, as in using tolerant, inclusive, and equitable scientific and technical language, hence avoiding the endorsement of prejudices by using derogatory ethnoreligious terms as scientific terminology.

The authors also argued that the attributes assigned to the ethnoreligious term used as a technical term include, apart from the antidote-calling qualities of being bad, malicious, dishonest, and deceitful, also the quality of being omniscient. This argument indicates the authors appreciate the offensive and disrespectful character of the term in question, yet believe that is amplified by including a positive characteristic in the mix. It seems to this reviewer that such arguments ridicule the spirit and the intent of the ICLR Code of Ethics.

**Strength And Weaknesses:**

Strength
- Novel theoretical results, though mostly based on previous work on VR-Marina, a variance-reducing method.
- Interesting combination of robust aggregation and VR-Marina in a deviance setting.

Weakness
- Ambiguous and undefined terminology using a lexicographically documented ethnoreligious prejudice as a technical term.
- Use of language violating ACM guidelines on inclusiveness and non-discrimination.
- Confusing use of terminology, mixing a subset with a set.
- Lack of experimental comparison to alternative deviance-tolerant variance-reducing methods.
- Refusal to use the objective term "Deviant" instead of perpetuating an unscientific derogatory stereotype.

**Summary Of The Paper:**

This paper proposes a method for training a deep neural network in a federated fashion in the presence of malicious or misled workers who misreport their gradient contributions, aiming to reduce variance of the stochastic gradients during learning and compress. It illustrates that the reduction of variance, is, unsurprisingly, key to withstanding the activities of such workers and achieve robust learning. The paper contributes tighter guarantees than previous work in the area with fewer assumptions, as well as a method that can work under nonuniform gradient sampling.

**Summary Of The Review:**

The authors made some effort to improve the scientific clarity of the terms employed in the paper to properly identify the kind of workers envisaged in this context, however they fell short of providing clear technically worded definitions that would not rely on a prejudice dating back to the 15th century to convey the required scientific meaning. In fact, they even insisted that alluding to such a prejudice is necessary to convey the intended technical neaning

Therefore, unfortunately, the core definitions and terms employed remain wanting clarity and scientific precision, and there have not been solid rectification steps; the authors also bluntly refused using any of the alternative terms suggested and to provide the definitions requested by this reviewer.

In particular, the authors disputed that the use of language in their original submission, which violates ACM guidelines on the matter, was ambiguous and potentially offensive. They argued that using an ethnoreligous term with a derogatory meaning as scientific terminology is fine as long as it is abbreviated to its first three letters. They used as their main piece of evidence the fact that this term was not offensive to themselves. They declined to engage in further discussion and to respond to either the scientific or the ethical concerns raised. They expressed open disagreement regarding the ambiguity and lack of scientific clarity in the definitions employed and decided to do nothing more about it, contrary to ACM guidelines.

Nevertheless, I appreciate the dedicated scientific effort made by the authors to provide additional results comparing with alternative variance-reducing methods as they could. I also appreciate their willingness to rectify the terminological ambiguity and omit allusions to an enduring cultural prejudice of medieval origin that have no place in a peer-reviewed academic article, albeit only superficially. Therefore, I recommend acceptance.

---

> ### Author Response · Authors · 2022-11-11
> **Authors' response to Reviewer xAhq [Argument 5]**
>
> **Argument 5: ICLR has a committee on ethical concerns.** To the best of our knowledge, ICLR has a committee on ethical concerns. This committee investigates arising ethical concerns. According to the ICLR reviewer guidelines (https://iclr.cc/Conferences/2023/ReviewerGuide), the reviewers should “summarize what the paper claims to contribute” and “be positive and constructive”, “list strong and weak points of the paper” and “be as comprehensive as possible”, “provide supporting arguments for your recommendation”, “ask questions you would like answered by the authors to help you clarify your understanding of the paper”, “provide additional feedback with the aim to improve the paper”. We emphasize that the review by Reviewer xAhq does not follow these guidelines. We also notice that ethical concerns follow after the review according to the guidelines. To the best of our understanding, the reviewers are required just to report potential ethical concerns and it should not be a base of the review. That is, it is not the reviewers job to write ethical reviews – this is a job of special ethical reviewers and the ethical committee.

---

> ### Author Response · Authors · 2022-11-11
> **Authors' response to Reviewer xAhq [Argument 4]**
>
> **Argument 4: We never intended to offend anybody and we are happy to change the term .** The thought that the term “Byzantine” can be offensive to some group of people never came to our minds during the work on the paper. This review is the first time we ever faced such an opinion. To indicate that we do not put any offensive meaning in the term, we have added to footnote 1 on page 1 the clarification. However, we do not know what is the best alternative to this term: the field actively uses this term. The usage of standard terms is crucial for the theoretical works, because the statements in Mathematics have to be as clear as possible. We are afraid that if we change the standard term “Byzantine” to another one, our paper will be harder to identify for the researchers in the field.
>
> However, we are open to good suggestions of replacements of the term. Moreover, we are against of any kind of the discrimination and we appreciate positive changes of the names and terms in the community, e.g., we believe the change of the name from “NIPS” to “NeurIPS” was a very important and absolutely right decision for the NeurIPS conference.

---

> > ### Comment · Reviewer_xAhq · 2022-11-11
> > **addressing all concerns in one place**
> >
> > Dear authors,
> >
> > Thank you for your responses. Please note that:
> >
> > 1. The review separated the comments on the scientific part, which it acknowledges, from the ethical concerns; there is a significant difference between pointing out that the use of some terminology is disrespectful to others, on the one hand, and an act of disrespect, on the other hand.
> > 2. Reviewers are asked to use their judgement in expressing ethical concerns, hence the corresponding field exists in the review form.
> > 3. Of course there are terms in science referring to groups of people, but they do not allude to a pejorative stereotype. The term "Hungarian algorithm", and other examples brought up, refer to the nationality of (some of) the inventors. There is a difference between that usage and the way the reviewed submission explicitly presents people of Byzantine affiliation as a poison or pest to which some antidote needs to be given.
> > 4. It is good to hear you would be happy to change the term. Indeed, the English language is richly endowed with several adjectives that exactly convey the notions of deviance, deviousness, deceit, deception, and dishonesty. Invoking a pejorative ethnoreligious stereotype found in English dictionaries to convey that notion is therefore superfluous and unnecessary. Therefore, I would suggest that:
> > - Using a proper noun for deceptive or deceitful workers is unnecessary to convey the message of the paper.
> > - The quality of being deviant, deceitful, devious, or dishonest may be expressed exactly by those terms.
> > - The condition of deviance, deviation, deceit, deception, deviousness, or dishonesty may be expressed exactly by those terms.
> > The message of the work will be much more clear to both scientists and laypersons expressed in such terms.

---

> > > ### Author Response · Authors · 2022-11-12
> > > **Response to Reviewer xAhq: we replaced the term**
> > >
> > > We thank the reviewer for the prompt reply. We replaced the term “Byzantines” with “Byz workers” (“Byzantine-robust” $\to$ “Byz-robust” and so on) to prevent a prejudiced use of language. This way our work remains recognizable in the community. In some sense, this change is very similar “NIPS” $\to$ “NeurIPS” change. Currently, the replacement is done only in PDF with the paper, since OpenReview does not allow changes to the title and the abstract on the page with the paper.
> > >
> > > The suggested terms by the reviewer do not fully capture the meaning of Byz workers, since such workers do not necessarily have bad intentions: Byz workers can deviate from the protocol due to the simple faults appearing on their side. Moreover, the suggested by the reviewer words do not reflect that such workers can be omniscient (please note that indeed, omniscience is a key component of the formal definition of Byzantine workers in the literature - that part of the meaning presumably can't be seen as offensive).
> > >
> > > Therefore, for now, this is the best alternative that we came up with hat satisfies these three desirable/important properties: i) being recognizable in the community, ii) having the same meaning as the initial term, and iii) not being offensive. We believe that any further decisions of changing the standard for the community term should be done not by the authors, reviewers, and ACs of this paper, but rather by the whole community. We suggest raising this issue on a global scale and somehow (via vote at the top-tier ML conferences) finding an alternative term that the majority of people in the community are satisfied with.

---

> > > > ### Comment · Reviewer_xAhq · 2022-11-12
> > > > **the term remains highly unsatisfactory on both ethical and scientific grounds**
> > > >
> > > > Thank you for being willing to make a change. However, please note that a shortening to an abbreviation does not constitute a change of the term.
> > > >
> > > > Regarding the scientific question of how satisfactory the term is technically, it should be said that the current term is highly unsatisfactory, opaque, and misleading in a technical sense. After all, in the original reference that introduced the term, the adjective "[B-word]" was used to characterize *all* the generals facing the so-called "[B-word] generals" problem, regardless whether they are loyal, disloyal, traitors, omniscient, or anything else. Therefore, the use of the term is technically misleading: an adjective meant to characterize the whole set is used for a subset. That use makes the paper scientifically vague and opaque. Differerent, more scientifically precise terms should be used in this paper to convey the subtleties of the problem in a semantically valuable and technically informative manner.
> > > >
> > > > The paper would be more scientifically neutral and valuable if it contained a proper scientific definition of what the term in question is meant to capture, since the current term is unsatisfactory for scientific and ethical grounds, as explained. Relying on a pejorative stereotype with regard to other people's cultural heritage and identity in order to convey a subtle technical meaning regarding set/subset attributes constitutes an opaque, scientifically insufficient, intellectually untenable, and ethically questionable use of terminology.
> > > >
> > > > Regarding the currently proposed solution, let me bring up an imaginary analogy: Imagine that some scientist had introduced a term using a derogatory cultural prejudice to express the well-known game theory concept that is currently expressed by the scientifically neutral, semantically valuable, and technically informative term "Prisoner's dilemma". Then imagine the term became widespread as a technical term in some scientific subcommunity fully unaware and uninformed about the nature and origin of the term in the particular cultural context it had arisen from. Imagine further that some reviewer pointed out that the term in question should not be used as a technical term on ethical grounds, as it had been chosen by reference to a cultural prejudice, had no semantic value in and of itself, and was therefore opaque and technically uninformative. One might use the semantically valuable, and technically informative term "Prisoner's dilemma" instead in its place, which is far less opaque and far more scientifically objective and precise. Imagine that was done in response to a paper calling for an "antidote" to those identified by the term in question in its title. Then, imagine that some author responded by proposing an abbreviation of the term to its first three letters, and raised a scientific disagreement with the reviewer as follows:
> > > >
> > > >    "*The suggested terms by the reviewer do not fully capture the meaning of the dilemma, since in that dilemma the prisoner in question has been arrested along with a collaborator. Moreover, the suggested by the reviewer words do not reflect that such prisoners are isolated.*"
> > > >
> > > > The above argument would be scientifically and epistemologically false. After all, people understand a technical term, such as "Prisoner's dilemma", with all its technical connotations only after having read about it, and do not understand any of those connotations if they have not read about it, but at least they do understand it has something to do with a prisoner. Therefore, while the term "Prisoner's dilemma" does not signify all those connotations in and of itself, it is much better than any alternative that would use an ethnic slur or a pejorative stereotype in the place of "Prisoner". Such alternatives would not make the term any better; they would make it far less informative and far more opaque. For the same reasons, using a pejorative stereotype as a technical term in the interactice consistency problem does not help signifying any of the required connotations and does not make the term scientifically informative. It is an idea at odds with the ICLR Code of Ethics that calls for respecting the cultural heritage of others. The problem addressed in the paper under review has nothing to do with people identifying with or drawing their cultural heritage from Byzantium, hence it has no justification to reinforce a cultural prejudice that sees such people as deviant. Any scientifically neutral and objective term, such as "Deviants", "Deviance-robust", "Traitor-robust", "Deceit-robust", and so on, would very well carry all those connotations to readers who have read about it, and would also be more semantically valuable and technically informative without disrespecting anybody's cultural heritage and thereby violating the ICLR Code of Ethics.

---

> > > > > ### Author Response · Authors · 2022-11-12
> > > > > **Response to "the term remains highly unsatisfactory on both ethical and scientific grounds"**
> > > > >
> > > > > Unfortunately, we believe your response is inappropriate, and for this reason, we unfortunately have to stop engaging with the kind of discussion you want us to have. We believe the discussion is non-scientific, and does not really belong to this forum. We are open to any scholarly debate pertaining to the real technical contributions of our paper.
> > > > >
> > > > > Based on previous discussion with you, we acknowledge that *you* feel that the term "Byzantine robustness", "Byzantine workers" and so on, may be seen by some (perhaps even you) as offensive. At the same time, we have never heard anyone complaining about it before, and do not know any person who holds this view. Indeed, your review is the first time we even became aware that this might be an issue.
> > > > >
> > > > > But again, we understand your point - and agree that it *may* be a valid point. However, we do not really know. A scientific study would need to be conducted for us to get some data on this point. We do not wish to *assume* this is the case just because one reviewer feels strongly about it. You may be correct, or not. We do not know. The term does not feel offensive to us as authors, despite the fact that some of us can indeed trace their roots (if we look far enough in history) to the Byzantine empire. So, we disagree it is self-evident that this terminology is offensive. Please note that the technical term "Byzantine worker" also includes omniscience, which is a strongly positive trait on its own. We have asked around some colleagues, and none of them find the term offensive. This is a small sample to draw any conclusions, of course. But it is larger than the sample of one provided by your view. Again, this underscores the need to conduct a study.
> > > > >
> > > > > Nevertheless, despite all of the above, we decided to lean over backwards, and proposed a solution which we believe is viable: to replace the term "Byzantine" with "Byz". Indeed, "Byz" is *not* the same word as "Byzantine", is potentially a recognizable replacement just like "NeurIPS" is not "NIPS" and retains recognizability. You suggest this is still not appropriate, but we disagree, and do not h ave better ideas. The alternatives you propose are problematic.
> > > > >
> > > > > You want us to not only change the terminology in our paper, but to also become passionate advocates for the change. Indeed you say "I think that the paper should contain a proper definition of what the term in question is meant to capture, and raise the problem that the term is highly unsatisfactory for both ethical and scientific reasons." However, we are not passionate about this change since we have doubts about the inappropriateness of the word both ethically, and scientifically. We are not aware of any evidence that supports these claims, and hence we can't defend such a proposal. If it is your view that the term is dubious on ethical and scientific basis - then perhaps you might be willing to write a study supporting these claims? Alternatively, you could point us to the evidence.
> > > > >
> > > > > We believe your analogy with the "N" word is inappropriate and misleading. We do not wish to go down the rabbit hole of addressing your point here, whatever it may be. If you believe that the word is so self-evidently offensive as the "N" word, you are offending us since you are indirectly implying that we must have been aware of its offensive power and still decided to do nothing. We sincerely hope this is not what you wanted to say.
> > > > >
> > > > > If you feel passionate about changing established scientific terminology you deem to be problematic, which seems like a worthy pursuit, please engage with the scientific community at large. We believe that it is unethical to hold the authors of a single paper, who are simply just using established terminology with about 40 years of history, hostage in the way you do so. It seems to us that you are exercising your power as a reviewer to fight a "policy" fight. You gave us the minimal score solely because we used established terminology which you personally find to be offensive.
> > > > >
> > > > > You do not engage with the scientific results we have spent many months developing - and we personally find that very offensive and unethical to us as authors, and to the community of AI researchers in general. Indeed, you decided to ignore our results altogether when making your recommendation, while this is your duty as a reviewer according to the official guidelines. It is known that AI conference reviews are problematic, and your review seems to add to the list of problems we face.
> > > > >
> > > > > As you can see, this kind of a debate does not seem to lead anywhere, and it certainly does not look appropriate in this particular forum. It is a philosophical and/or policy debate - and not a scientific debate. We hope the AC and PCs will be able to step in and resolve this situation.
> > > > >
> > > > > In conclusion, we will not be further discussing the issue of the terminology "Byzantine robustness" in this forum. We are, however, happy to have a technical debate about our scientific contributions.

---

> > > > > > ### Comment · Reviewer_xAhq · 2022-11-12
> > > > > > **Avoiding the technical points in my reply does not advance the discussion**
> > > > > >
> > > > > > Dear authors,
> > > > > >
> > > > > > It appears that the three technical points of my reply went unnoticed:
> > > > > >
> > > > > > 1. My reply has raised a question about the technical appropriateness of the term, in terms of confusion between set and subset.
> > > > > >
> > > > > > 2. The example I brought up was meant exactly to highlight the confusion of an ethical argument with a scientific one: a term does not have an inherent technical meaning in and of itself; it is simply defined to do so.
> > > > > >
> > > > > > 3. My reply did propose alternative technical terms.

---

> > > > > > > ### Author Response · Authors · 2022-11-12
> > > > > > > **Re:**
> > > > > > >
> > > > > > > "In conclusion, we will not be further discussing the issue of the terminology "Byzantine robustness" in this forum. We are, however, happy to have a technical debate about our scientific contributions."

---

> > > > > > ### Comment · Reviewer_xAhq · 2022-11-12
> > > > > > **Matters of principle cannot be decided by self-reference**
> > > > > >
> > > > > >   "The term does not feel offensive to us as authors, despite the fact that some of us can indeed trace their roots (if we look far enough in history) to the Byzantine empire. So, we disagree it is self-evident that this terminology is offensive."
> > > > > >
> > > > > > This reply misses the point: any ethnoreligous terms used to convey any negative trait as a technical term is inappropriate, since there are bound to be some people who find it offensive. The way to judge this scientifically is not to check whether you find that particular term offensive. Instead, the way to judge it is to check how you would find any ethnoreligious terms that you identify with used in that place. A capacity for such empathy is necessary to exercise such judgement.

---

> > > > > > > ### Author Response · Authors · 2022-11-12
> > > > > > > **Re:**
> > > > > > >
> > > > > > > "In conclusion, we will not be further discussing the issue of the terminology "Byzantine robustness" in this forum. We are, however, happy to have a technical debate about our scientific contributions."

---

> > > > > > ### Comment · Reviewer_xAhq · 2022-11-12
> > > > > > **The point of the analogy**
> > > > > >
> > > > > >     If you believe that the word is so self-evidently offensive as the "N" word, you are offending us since you are indirectly implying that we must have been aware of its offensive power and still decided to do nothing. We sincerely hope this is not what you wanted to say.
> > > > > >
> > > > > > Exactly: this is not what I said. In the analogy, as in the present case, the people using the word are not aware of its offensive power, and that was exactly the point. In the analogy, as in the present case, the scientist who introduced the term was aware that it was reinforcing a pejorative stereotype and considered it to be scientifically appropriate for that reason. In the analogy, as in the present case, the authors raised a scientific disagreement with the reviewer, arguing that using the pejorative stereotype was technically more appropriate. In the analogy, as in the present case, the reviewer disagreed on both scientific and ethical grounds, leading to a technical and an ethical disagreement between authors and reviewer.

---

> ### Author Response · Authors · 2022-11-11
> **Authors' response to Reviewer xAhq [Argument 3]**
>
> **Argument 3: There are a lot of examples of other geographically/nation inspired names and terms in science.** To name a few: Hungarian algorithm (https://en.wikipedia.org/wiki/Hungarian_algorithm), Polish space (https://en.wikipedia.org/wiki/Polish_space), Method of Four Russians (https://en.wikipedia.org/wiki/Method_of_Four_Russians), Markov chain Monte Carlo methods (https://en.wikipedia.org/wiki/Markov_chain_Monte_Carlo). These names/terms became standard as well as “Byzantine workers”. They do not offend or discriminate anybody as far as we know.

---

> ### Author Response · Authors · 2022-11-11
> **Authors' response to Reviewer xAhq [Argument 2]**
>
> **Argument 2: The review does not address the scientific part of the paper.** The only concern mentioned in the review is related to the usage of one particular term “Byzantines”. Based on this concern the reviewer gave our paper the score “1”. We believe that this is extremely unfair/inappropriate to our paper and disrespectful to us: we put a lot of effort into obtaining the results and writing the paper. From the reviewers of top-tier ML conferences like ICLR we expect to receive constructive feedback about the scientific part of our work (our work is entirely scientific, and nothing else). We believe our paper provides much more subjects for the discussion than the usage of one word: it becomes evident after going at least through Tables 1-2 and “Our contributions” part (pages 3-5), which clearly indicate that we achieved new theoretical SOTA results, and paragraph “Challenges in designing variance-reduced algorithm with tight rates and provable Byzantine-robustness” (page 7), which explains why the derived results are non-trivial to obtain.

---

> ### Author Response · Authors · 2022-11-11
> **Authors' response to Reviewer xAhq [References for Argument 1]**
>
> [1] Lamport, L., Shostak, R., & Pease, M. (1982). The Byzantine Generals Problem. ACM Transactions on Programming Languages and Systems, 4(3), 382-401.
>
> [2] Yang, Y. R., & Li, W. J. (2021, July). Basgd: Buffered asynchronous sgd for byzantine learning. In International Conference on Machine Learning (pp. 11751-11761). PMLR.
>
> [3] Karimireddy, S. P., He, L., & Jaggi, M. (2021, July). Learning from history for byzantine robust optimization. In International Conference on Machine Learning (pp. 5311-5319). PMLR.
>
> [4] Data, D., & Diggavi, S. (2021, July). Byzantine-resilient high-dimensional sgd with local iterations on heterogeneous data. In International Conference on Machine Learning (pp. 2478-2488). PMLR.
>
> [5] Gorbunov, E., Borzunov, A., Diskin, M., & Ryabinin, M. (2022, June). Secure distributed training at scale. In International Conference on Machine Learning (pp. 7679-7739). PMLR.
>
> [6] Farhadkhani, S., Guerraoui, R., & Villemaud, O. (2022, June). An Equivalence Between Data Poisoning and Byzantine Gradient Attacks. In International Conference on Machine Learning (pp. 6284-6323). PMLR.
>
> [7] Farhadkhani, S., Guerraoui, R., Gupta, N., Pinot, R., & Stephan, J. (2022). Byzantine Machine Learning Made Easy by Resilient Averaging of Momentums. (Accepted to NeurIPS 2022)
>
> [8] Zhang, X., Yuan, Z., Zhu, M. (2022). Byzantine-tolerant federated Gaussian process regression for streaming data. (Accepted to NeurIPS 2022)
>
> [9] Datar, A., Rajkumar, A., Augustine, J. (2022). Byzantine Spectral Ranking. (Accepted to NeurIPS 2022)
>
> [10] El-Mhamdi, E. M., Farhadkhani, S., Guerraoui, R., Guirguis, A., Hoang, L. N., & Rouault, S. (2021). Collaborative learning in the jungle (decentralized, byzantine, heterogeneous, asynchronous and nonconvex learning). Advances in Neural Information Processing Systems, 34, 25044-25057.
>
> [11] Sohn, J. Y., Han, D. J., Choi, B., & Moon, J. (2020). Election coding for distributed learning: Protecting signsgd against byzantine attacks. Advances in Neural Information Processing Systems, 33, 14615-14625.
>
> [12] Ghosh, A., Maity, R. K., & Mazumdar, A. (2020). Distributed newton can communicate less and resist byzantine workers. Advances in Neural Information Processing Systems, 33, 18028-18038.
>
> [13] Li, J., Abbas, W., & Koutsoukos, X. (2020). Byzantine resilient distributed multi-task learning. Advances in Neural Information Processing Systems, 33, 18215-18225.
>
> [14] Blanchard, P., El Mhamdi, E. M., Guerraoui, R., & Stainer, J. (2017). Machine learning with adversaries: Byzantine tolerant gradient descent. Advances in Neural Information Processing Systems, 30.
>
> [15] Karimireddy, S. P., He, L., & Jaggi, M. (2022). Byzantine-Robust Learning on Heterogeneous Datasets via Bucketing. In International Conference on Learning Representations.
>
> [16] Konan, S. G., Seraj, E., & Gombolay, M. (2022). Iterated Reasoning with Mutual Information in Cooperative and Byzantine Decentralized Teaming. In International Conference on Learning Representations.
>
> [17] Allen-Zhu, Z., Ebrahimianghazani, F., Li, J., & Alistarh, D. (2021). Byzantine-Resilient Non-Convex Stochastic Gradient Descent. In International Conference on Learning Representations.
>
> [18] El Mhamdi, E. M., Guerraoui, R., & Rouault, S. L. A. (2021). Distributed momentum for byzantine-resilient stochastic gradient descent. In International Conference on Learning Representations (ICLR).
>
> [19] Bouhata, D., & Moumen, H. (2022). Byzantine Fault Tolerance in Distributed Machine Learning: a Survey. arXiv preprint arXiv:2205.02572.
>
> [20] Lyu, L., Yu, H., Ma, X., Sun, L., Zhao, J., Yang, Q., & Yu, P. S. (2020). Privacy and robustness in federated learning: Attacks and defenses. arXiv preprint arXiv:2012.06337.

---

> ### Author Response · Authors · 2022-11-11
> **Authors' response to Reviewer xAhq [Argument 1]**
>
> We kindly disagree with the reviewer’s assessment of our paper. In particular, the reviewer gave our paper the score “1” solely based on the usage of one term, which is standard for the field. We emphasize that this is the only criticism that the reviewer provided in the review. Below we explain in detail why we believe that the review is unfair to our paper.
>
> **Argument 1: The term “Byzantines/Byzantine workers” is old and standard for the literature.** In our work, we follow the standard terminology for the field. To the best of our knowledge the term “Byzantine fault” was introduced in 1982 by Lamport et al. [1] (more than 40 years ago) and it goes back to the famous Byzantine general problem (e.g., see the article at Wikipedia https://en.wikipedia.org/wiki/Byzantine_fault). Since then this term and its multiple variations (including “Byzantines/Byzantine workers/Byzantine agents/Byzantine clients/Byzantine faulty agents”) were actively used in the research papers to denote the workers in the distributed system that can somehow deviate from the prescribed protocol. Below we provide an extensive list of the papers from the top-tier ML conferences (ICLR, NeurIPS, ICML) that use this terminology. We have also searched the papers on arXiv (with subject “Computer Science”) and **found 841 papers that use the word “Byzantine”. Quick skimming through the abstract of the first 100 papers shows that almost all of these works use the term in the same meaning that we use in our work.** There are also recent surveys [19, 20] that use the term “Byzantines/Byzantine attackers/Byzantine workers” frequently.

---

> ### Public Comment · ~Dmytro_Mishkin1 · 2022-11-13
> **Please, do not use Ukraine to justify your review.**
>
> I am Ukrainian and never in my life saw anyone identifying themselves as Byzantines. We are Ukrainians, some of us are Crimean Tatars, etc, but not Byzantines.
> Yes, Ortodox Church has roots there, but it was thousand years ago. Nor Constantinople uses this name now, instead being “ Ecumenical Patriarch of Constantinople”.
>
> So, please don’t use our name and don’t be offended on our behalf - we are capable to defend our identity. If we need some explicit help - we ask for it directly.
>
> Best, Dmytro.

---

> ### Public Comment · ~Georgios_Papadopoulos1 · 2022-11-14
> **Cultural Vigilantism**
>
> This review derails the focus from the authors and their work. In any other case I would be of the opinion to not give further credence to it, but the reviewer complains and argues about a matter that clearly have developed only a superficial knowledge. The reviewer seems to follow an elitist approach by skimming through some Wikipedia articles and then self-appointing themselves as the vigilantes of people who identify as “Byzantines”. The reviewer, since they chose to delve into such a non-trivial and academically (for historians and folklorists) sensitive cultural matter, should have invested more time to study the folklore, societies, history, and culture of the people who they refer too. What they have done instead is a cultural appropriation of the worst form.
>
> I hope these comments shed some light for the people (the authors and the other reviewers too) who might not be familiar with the topic of Eastern Roman Empire and the lasting cultural effect it had in Balkans and the East Mediterranean people.
>
> First, the reviewer confuses the national and historical identity of people with religious and theological matters. The examples that they provide to justify that some people “feel” as “Byzantines” are that in their Eastern Christian domination they use the so-called Byzantine rites. This is not a ethnical and national heritage, it’s a liturgical practice and nothing more. The original name of these rites was “Greek”. With the same logic these group of peoples feel Greek. They do not and they are not; the same way a French or a Canadian Catholic does not feel Italian because the rites come from the Holy See.
>
> Specifically, according to the official Code of Eastern Churches: “A rite is the liturgical, theological, spiritual and disciplinary patrimony, culture and circumstances of history of a distinct people, by which its own manner of living the faith is manifested in each Church sui iuris. The rites treated in this code, unless otherwise stated, are those which arise from the Alexandrian, Antiochene, Armenian, Chaldean and Constantinopolitan traditions.”
> (http://www.intratext.com/IXT/ENG1199/_PS.HTM)
>
> It is not a national identity, as people who follow the Eastern Greek Catholic Churches feel Albanian, Ukrainians, Romanians, etc. They do not identify neither as Greeks nor Byzantines (and in many cases would be offensive for someone to even suggest so). The only reason that the rites are called Byzantines and not Greeks is exactly because these people do not identify as Greeks and therefore to avoid any controversy (in a similar matter that we use the Byzantine generals and not Albanian generals). Therefore, the reviewer completely misses the point here of what they have been complaining about.
> (https://www.britannica.com/topic/Byzantine-rite)
>
>
> The reviewer also, in one of their examples, refer to the “Rumlar/Byzantine” peoples of Istanbul. First of all, they are called Phanariots, and they identify as Greeks, not Byzantines (which has no meaning whatsoever). It is a Greek community that has been prosecuted and targeted because of the 20th century nationalistic sentiment that rose in both Turkey and Greece that led to brutal ethno-catharsis. It has nothing to do with a Byzantine identity. As a Greek, I would advise the author to stay away from matters that clearly, they do not comprehend and are too sensitive to be used to support their personal self-righteous agenda. For more context, a very well written book to start would be the Greece: Biography of a Modern Nation by Roderick Beaton (2021).
>
> Contemporary, the only people who have developed a national and cultural identity that includes Byzantium are the Greeks, the Phanariots, and the Pontic Greeks (as well as Greek communities around the Black Sea). Even them, they do not identify as Byzantines but rather as Greeks. The Eastern Roman Empire (Byzantium) is considered part of our historical timeline, just a period amongst other periods. This identity comes from folklore stories about Digenes and the Marble Emperor; stories that existed for eons in the rural areas and been narrated in the Greek language, from myths and legends, from old believes (The Book of Greek and Roman Folktales, Legends, and Myths by William Hansen, 2017). It does not come just from a simple liturgical rite.
>
> Historically, to be called Byzantine also does not make any sense. Byzantium is a term that has been coined in 16th century Europe. The people in the Eastern Roman Empire were called and felt Romans. Their society, their economic system, their military tradition, their culture, diplomacy, their titles and laws, were all of Roman heritage (for more information about the Eastern Roman Empire society and its history, George Ostrogorsky’s books are the best and most comprehensive source). “Byzantine” in its current form is an abstract term that refers to a multi-ethnic Empire (dominated by Greeks) but not the people per se.
>
> Part (1/2)

---

> > ### Public Comment · ~Georgios_Papadopoulos1 · 2022-11-14
> > **Cultural Vigilantism**
> >
> > Part (2/2)
> >
> > Finally, Eastern Roman Empire lasted for more than a thousand years (almost 2000 years if we consider the reality that it was a continuous entity since Roman times). Of course, it would have left a cultural and religious mark around its sphere of influence. But so has done the U.S now. Should a German person who watches U.S made movies and listens to U.S music and speaks English be considered “American”? Does that person feel “American”? Religious conformity back then had to do with diplomacy and spheres of influence and not that a Slav or Ukrainian person would see themselves as Greeks or “Byzantines”.
> >
> > I profoundly apologise to the authors and the other reviewers for the out of topic comment. I do not wish to keep hostage this conversation further to ethnological matters but rather to provide support to their reasonable adjustments against a very unreasonable reviewer. The reviewer if truly cares how other cultures feel should invest more time speaking to them, visiting them, and learning their history rather than attacking people on unrelated topics based on what they read in Wikipedia and that do not comprehend.
> >
> > As a final example, this is the front page of a right-wing/nationalistic newspaper in Greece (13th November 2022). They use the term “Byzantine” in its derogatory form. It’s a common usage in Greek. If a pseudo-patriotic newspaper has no problem with that term, we understand how little it means to the average person.
> > (https://www.protoselidaefimeridon.gr/efimerides/221113/estia.JPG)

---

> > > ### Comment · Reviewer_xAhq · 2022-11-14
> > > **respect for cultural heritage**
> > >
> > > Thank you for the thoughtful contributions.
> > >
> > > First, please note that the ICLR Code of Ethics calls for respecting the cultural heritage of others, which has several dimensions:
> > > https://iclr.cc/public/CodeOfEthics
> > >
> > > Regarding current instances of using the term as an aspect of cultural identity in its substantive form, please see, for example:
> > > https://www.byzcath.org/index.php/about-us-mainmenu-60/about-byzantines-mainmenu-62
> > >
> > > Regarding the Rumlar community of Istanbul, please check Wiktionary on the English equivalents and translations of the term "Rum":
> > > https://en.wiktionary.org/wiki/Rum#Turkish
> > >
> > > Regarding the newspaper front page, please note that: (a) it uses the city name (not the adjective, neither the substantive) in quotes, and (b) for instance, African-Americans also use the N-word as a colloquial and even pejorative term among themselves, yet that usage does not justify peer-reviewed academic articles using the same term as a technical term alluding to a pejorative stereotype, instead of a semantically valuable, technically informative, and scientifically neutral term conforming to the ICLR Code of Ethics.

---

> > > > ### Public Comment · ~Georgios_Papadopoulos1 · 2022-11-14
> > > > **Cultural Vigilantism**
> > > >
> > > > I will stop responding because either the reviewer is clearly a troll or they have a different agenda from objectively reviewing the paper.
> > > >
> > > > My last comments:
> > > >
> > > > 1. Rumlar is the name who the Ottoman Turks (and contemporary Turks) gave to the Greek Orthodox citizens of the Byzantine empire. It means Roman. Rumelia == "Land of the Romans" == Greece. Eastern Roman Empire had many ethnicities, Armenians, Bulgarians, Slavs. There is no Byzantine person or identity.
> > > >
> > > > 2. I would advice the reviewer to read carefully the clearly random (quick google search) links that posts. From the byzcath.org "Byzantine Catholics in America are the spiritual descendants of Christians in Central and Eastern Europe and the Middle East who are the heirs of this Byzantine religious culture, and who therefore trace their spiritual heritage to the Great Church of Constantinople, known as Hagia Sophia (The Church of Holy Wisdom)." If the reviewer had read what posted, they would have seen the very distinct words of "spiritual descendants" and "religious culture". It is not a Byzantine identity. No person from the Eastern Catholic Church identifies or feels Byzantine. And is not a "cultural identity" (as mistakenly the reviewer insists) but a "spiritual and religious one".
> > > >
> > > > 3. Byzantium is a city. There was never a state called Byzantium, neither a nation, nor people. So, the newspaper is using it correctly. The term Byzantium was coined in 16th century exactly to deny the Greeks from their Roman Imperial identity. To show they were unworthy of the Western European standards and culture. By the same logic, you insist to use a derogatory term to forcefully identify people. To use your example, instead of calling African Americans (Eastern Romans) you want to call them the n-word. We thank you for that.

---

> > > > > ### Comment · Reviewer_xAhq · 2022-11-14
> > > > > **respect for cultural heritage**
> > > > >
> > > > > Thank you for all the thoughtful contributions.
> > > > >
> > > > > Please note that:
> > > > > 1. "Byzantines" is an English derogatory exonym for "Rumlar", just like, e.g., "Japs" is an English derogatory exonym for "日本人"; the fact that Japanese people do not call themsleves "Japs" does not imply that using this derogatory exonym as a technical term abides by the ICLR Code of Ethics.
> > > > > 2. Religious culture is part of culture.
> > > > > 3. I am asking to not use this derogatory exonym as a technical term, abiding by the ICLR Code of Ethics.

---

> ### Comment · Program_Chairs · 2022-11-14
> **Comments from Ethic Chair and PC committee**
>
> The PC committee and Ethics Chair have been following this thread closely. Upon preliminary investigation, the Ethics Chair finds that the use of B-word is a "possibly emerging" issue but not yet a "major ethics issue" that could justify rejecting research. There seems no widespread agreement that the B-word is offensive. This discussion between reviewer and authors is still valuable to our community, which raises awareness of this potentially emerging issue. We appreciate thoughts and comments from both the review and the authors.
>
> On the other hand, Ethics Chair points out the reviewer uses the N-word explicitly in drawing what seems to be a false equivalence between the N-word (where there’s widespread agreement that the word is offensive) and the B-word (which is still an emerging/unsettled issue but possibly with some merit that is beyond the scope of our ethics committee to fully investigate and resolve now). This misuse of the N-word poses a violation of the ICLR Code of Conduct. We strongly encourage the reviewer to edit his response and address this misuse voluntarily.
>
> If there is any more question regarding these issues, please feel free to contact the Ethic Chair and PCs. If needed, the Ethic Chair may decide to advance this discussion to a formal ethics review.

---

> > ### Comment · Reviewer_xAhq · 2022-11-15
> > **Voluntary edit**
> >
> > Thank you for your valuable feedback. Please note that I have edited the paragraphs in question voluntarily to forestall any misunderstandings of meaning and intent. Following your request, I further re-edited them to read as follows:
> >
> > "Regarding the scientific question of how satisfactory the term is technically, it should be said that the current term is highly unsatisfactory, opaque, and misleading in a technical sense. After all, in the original reference that introduced the term, the adjective "[B-word]" was used to characterize *all* the generals facing the so-called "[B-word] generals" problem, regardless whether they are loyal, disloyal, traitors, omniscient, or anything else. Therefore, the use of the term is technically misleading: an adjective meant to characterize the whole set is used for a subset. That use makes the paper scientifically vague and opaque. Differerent, more scientifically precise terms should be used in this paper to convey the subtleties of the problem in a semantically valuable and technically informative manner.
> >
> > The paper would be more scientifically neutral and valuable if it contained a proper scientific definition of what the term in question is meant to capture, since the current term is unsatisfactory for scientific and ethical grounds, as explained. Relying on a pejorative stereotype with regard to other people's cultural heritage and identity in order to convey a subtle technical meaning regarding set/subset attributes constitutes an opaque, scientifically insufficient, intellectually untenable, and ethically questionable use of terminology.
> >
> > Regarding the currently proposed solution, let me bring up an imaginary analogy: Imagine that some scientist had introduced a term using a derogatory cultural prejudice to express the well-known game theory concept that is currently expressed by the scientifically neutral, semantically valuable, and technically informative term "Prisoner's dilemma". Then imagine the term became widespread as a technical term in some scientific subcommunity fully unaware and uninformed about the nature and origin of the term in the particular cultural context it had arisen from. Imagine further that some reviewer pointed out that the term in question should not be used as a technical term on ethical grounds, as it had been chosen by reference to a cultural prejudice, had no semantic value in and of itself, and was therefore opaque and technically uninformative. One might use the semantically valuable, and technically informative term "Prisoner's dilemma" instead in its place, which is far less opaque and far more scientifically objective and precise. Imagine that was done in response to a paper calling for an "antidote" to those identified by the term in question in its title. Then, imagine that some author responded by proposing an abbreviation of the term to its first three letters, and raised a scientific disagreement with the reviewer as follows:
> >
> >    "*The suggested terms by the reviewer do not fully capture the meaning of the dilemma, since in that dilemma the prisoner in question has been arrested along with a collaborator. Moreover, the suggested by the reviewer words do not reflect that such prisoners are isolated.*"
> >
> > The above argument would be scientifically and epistemologically false. After all, people understand a technical term, such as "Prisoner's dilemma", with all its technical connotations only after having read about it, and do not understand any of those connotations if they have not read about it, but at least they do understand it has something to do with a prisoner. Therefore, while the term "Prisoner's dilemma" does not signify all those connotations in and of itself, it is much better than any alternative that would use an ethnic slur or a pejorative stereotype in the place of "Prisoner". Such alternatives would not make the term any better; they would make it far less informative and far more opaque. For the same reasons, using a pejorative stereotype as a technical term in the interactive consistency problem does not help signifying any of the required connotations and does not make the term scientifically informative. It is an idea at odds with the ICLR Code of Ethics that calls for respecting the cultural heritage of others. The problem addressed in the paper under review has nothing to do with people identifying with or drawing their cultural heritage from Byzantium, hence it has no justification to reinforce a cultural prejudice that sees such people as deviant. Any scientifically neutral and objective term, such as "Deviants", "Deviance-robust", "Traitor-robust", "Deceit-robust", and so on, would very well carry all those connotations to readers who have read about it, and would also be more semantically valuable and technically informative without disrespecting anybody's cultural heritage and thereby violating the ICLR Code of Ethics."
> >
> > I would be glad to hear suggestions on how to clarify this response further.

---

> ### Public Comment · ~Shaochen_Zhong1 · 2022-11-18
> **Good intentions with potentially flawed arguments: on the note of terminology vagueness and victimhood.**
>
> First, let me start off by saying that I believe replacing problematic terms from the science world is a noble act that we should all support (much like we changed "NIPS" to "NeurIPS"). To achieve that, it is vital to have a fair and open discussion regarding such issues.
>
> **Thus, though I believe the reviewer should have provided a much more constructive review on the scientific aspect, I also do think OpenReview serves as a decent platform to discuss such a word-use issue.** So I’d share my thoughts here at the risk of further derailing the review process. For that, I’d apologize in advance.
>
> ---
>
> To the best of my understanding, the reviewer’s arguments are mainly two-fold:
>
> ```
> 1. “B-word” is scientifically vague and opaque. Since the “B-word generals” can be “loyal, disloyal, traitors, omniscient, or anything else…” The reviewer argues it is technically misleading because “an adjective meant to characterize the whole set is used for a subset.”
> 2. “B-word” is used to carry negative connotations, as it is used to imply (and used interchangeably with terms like) “bad” and “malicious.”
> ```
> **The reviewer’s findings seem to be true. However, there is quite a gap between these findings to the conclusion of banning the B-word from literature use.**
>
> ---
> ## Scientific vagueness does not justify a ban.
>
> It is common knowledge that many popular science terms are defined in a vague or fuzzy matter. A good example is [***“kernel”***](https://en.wikipedia.org/wiki/Kernel) under the realm of STEM. However, **such ubiquitous polysemy or ambiguity should not disqualify a term’s usage**, since scholars can often infer the correct meaning from its context or by applying necessary modifiers.
>
> Not to mention that “B-word fault” is a well-known term in the distributed system community. So changing “B-word worker” to “deviant worker” or likes may lose this semantic in a title search scenario, **making the term even more scientifically vague.**
>
> ## Negative terms without a modern victim are potentially less offensive.
>
> Regarding the derogative use of the “B-word.” Although it is indeed used to carry negative meaning, **it lacks a modern victim — because no one identifies themselves as “Byzantines.”** While I can’t find any comparable term in English off the top of my head, there are quite many idioms in my culture/ethnicity (Chinese) that involve negative connotations to a certain historical group, to name a few:
>
> * 「齐梁世界」"The world of Qi and Liang": used to describe a weak empire due to corruption and bureaucracy — very similar to the B-word here.
> * 「郑人买履」"Man from the State of Zheng buying shoes": it is about a man — from a State with a quite “dogmatic culture” — who rather trusts his measurements than use his own feet when buying shoes, implying a stubborn adherence to doctrines without proper adaptation.
>
>
> All terms like these are regularly used, despite millions of people being culturally/geographically rooted to the State of Qi/Liang/Zheng (my parents included), presumably because no one identifies as the people of Qi/Liang/Zheng anymore. This further illustrates why a term like “N-word dilemma” is obviously not ok on top of the intensity difference, as the “N-word” targets the whole modern black community. **The reviewer’s analogy is definitely off there.**
>
>
> ---
> **It is not a universal rule that a term with unfavorable connotations must have a modern victim to be banned.** Hell, some terms with modern victims are being regularly thrown around, e.g., “Spanish flu,” “Finlandization,” or even “barbarian.” **But it is without a doubt that the lack of modern victims will provide a healthy reduction in its severity.**
>
>
> In this case, given the argument of scientific vagueness does not hold much water. It is IMO hard to conclude banning the B-word from
> literature use solely because it is a group/cultured-oriented term carrying negative connotations (while lacking a modern victim).
>
>
> I would love to see the distributed system community walks away from this term, **as “Byzantine” is a word that’s losing its currency among historians because of the historical inaccuracy and potential derogative incentives it carries** (here’s a good video on why, as it seems many people on Twitter are unfamiliar: [short](https://www.youtube.com/shorts/up0gj3GKly8) and [long](https://www.youtube.com/watch?v=rN9sg2XKuuo) version.). But that will open a whole new can of worms on the nomenclature of locations in different languages (e.g., “Macao”). **The reviewer might come with good intentions, but IMO really shouldn’t hold the authors hostage on such a widespread matter that’s beyond their ability to fix.**

---

> > ### Comment · Reviewer_xAhq · 2022-11-18
> > **Should pejorative English exonyms be used as technical terms?**
> >
> > Thank you for the contribution.
> > Please note that:
> > 1. The word in question was chosen deliberately for its negative connotations that reflect a medieval cultural prejudice.
> > 2. There are indeed living people and living cultural heritage identified by that English exonym.
> > 3. Noone identifies as a "Chinaman", but as "中国人". Should that word be used as a technical term alluding to its pejorative sense?

---

> > > ### Public Comment · ~Shaochen_Zhong1 · 2022-11-18
> > > **If it has a modern victim, then I am in favor of answering "No."**
> > >
> > >
> > >
> > > 1. I have never objected to that (maybe with a question mark on the "prejudice" part, as it is debatable whether the Eastern Roman generals back then exhibited the described inefficient/bureaucratic behavior, but I believe we'd both agree that's not the core topic).
> > > Here I quote: *"[the 'B-word'] it is indeed used to carry negative meaning"* and many more sentences.
> > >
> > > I just believe — to the best of my knowledge — it lacks a modern victim, and therefore discounted its severity.
> > >
> > > 2. True, but "inheriting some culture/religious practices from the Eastern Roman Empire" is very different from "self-identified as a Byzantine." Again, in the examples I have given, millions of populations carry cultural practices from the State of Qi/Liang/Zheng, yet they have no problem using idioms that has negative connotations towards such states, presumably because they don't identify themselves as people of such states.
> > >
> > > 3. "Chinaman" is a term that aims to describe the Chinese population or people of Asian descent (which later took on negative connotations, much like "orientals"). **These people identify themselves as "中国人/华人/Chinese," and they exist today**, so using a technical term made upon an offensive slur towards them is undoubtedly problematic.
> > >
> > > It is, however, different from the presented B-word problem here, again because no one self-identified as a "Byzantine."
> > >
> > > ---
> > >
> > > Back to your subject question, I believe if the term/exonym involves a modern victim, then I favor not using such a term. In the examples I gave (*"Spanish flu," "Finlandization," or even "barbarian."*), the medical community has already moved away from naming viruses according to locations, and that is IMO a welcomed change.
> > >
> > > If it is without a modern victim, I am still in favor of using a neural, historically accurate term. But as discussed in the last paragraph of my first comment, that would raise a much bigger problem beyond the authors' ability to address. In this particular case, there is no modern victim, and it is hard to find a perfect replacement for the "B-word" given its unique tie to the distributed system community.
> > >
> > > I applaud you for raising a concern that most CS folks are unfamiliar with (as many people in various Twitter threads I came across believe the term "Byzantine" is historically accurate and neutral, which is wrong), but I don't think it is fair to hold the authors hostage to press for a community-level term change. It creates unnecessary hostility that dilutes the core issue. **Most term changes I witnessed gradually gathered momentum starting from attention and discussions — you did that already; now, I'd encourage you to forgo this aspect and focus on the scientific aspect of the work.** Judging from the [publications](https://openreview.net/forum?id=pfuqQQCB34&noteId=Mt1fJ3PJ8TS) surveyed by the authors, the community is not ready for this change, so pressing on the authors is like putting them in a dilemma between being (more likely to be) rejected or sacrificing academic exposure — a lose-lose situation they can't escape.

---

> > > > ### Comment · Reviewer_xAhq · 2022-11-18
> > > > **By the same argument, the term "Chinaman" would have no modern victim either**
> > > >
> > > > Thank you for your thoughtful response. Please note that:
> > > > 1. It is inaccurate to use a name for a people as a generalizing derogatory term for some behavior and to use the temporal qualifier "back then" for a time duration spanning millennia.
> > > > 2. There exist people (members of the Eastern Catholic Church) who call themselves “Byzantines”; further, it is peculiar to expect people of other cultures to use English exonyms for themselves: e.g., Germans call their culture "Deutsch", not "German"; similarly, there exist people identified in Turkish as "Rumlar", a term translatable to English as "Byzantines".
> > > > 3. By the same argument, one might wrongly claim that "Chinaman" would have no modern victim, as it refers to people of the 19th-century Qing empire.
> > > > 4. Reading my review would also be worthwhile.

---

> > > > > ### Public Comment · ~Shaochen_Zhong1 · 2022-11-19
> > > > > **Simply not true.**
> > > > >
> > > > > 1. **A term may be inaccurate, but that is far from worth being banned.** The two Chinese idiom examples I have given describe nations/people with negative connotations. Are they accurate? Maybe not, but no one advocates banning such terms because no one identifies themselves as victims of such terms. I believe the existence of modern victims, or not, plays a vital role in the determination of banning.
> > > > >
> > > > > Also, “back then” refers to a *“particular time in the past that you are talking about”* ([ref.](https://dictionary.cambridge.org/us/dictionary/english/back-then)), where the Eastern Roman Empire fits this definition perfectly. The lasting duration of an empire has no say on the use of “back then.”
> > > > > This is not even tangentially related to the core topic. I’d encourage the reviewer to refrain from word playing/picking like this, as it won’t nourish any helpful conversation; especially when you adhered to a wrong definition.
> > > > >
> > > > > 2. Again — as mentioned by Georgios Papadopoulos — **adopting a religious practice rooted in the Eastern Christian Church of Constantinople is very different from “self-identified as a Byzantine.”** Just like an American practitioner of Indian Buddhism is not an Indian.
> > > > >
> > > > > 3. Last, **claiming "Chinaman" only "refers to people of the 19th-century Qing empire" is just blatantly wrong**. A quick [Wiki read](https://en.wikipedia.org/wiki/Chinaman_(term)) and some Google news searches will teach you there are many incidents where people used “Chinaman” to address/attack people way past the Qing empire (there was literally [one incident](https://news.yahoo.com/woman-videoed-going-racist-rant-204329881.html) happened around three days ago). The people of Chinese (or even Asian) descent are clear modern victims of the term "Chinaman," and I'd welcome an apology on this matter.

---

> > > > > > ### Comment · Reviewer_xAhq · 2022-11-19
> > > > > > **Should people use English exonyms for themselves in order to be respected?**
> > > > > >
> > > > > > Thank you for your thoughtful reply. Please consider:
> > > > > > 1. Are these two Chinese idioms used as technical terms?
> > > > > >     (please note that a duration would need to be shorter than millennia to be called "a particular time")
> > > > > > 2. Is up to you to decide whether people calling themselves "Byzantines" do so for the correct reasons?
> > > > > >     Further, it is peculiar to expect people of other cultures to use English exonyms for themselves: e.g., Germans call their culture "Deutsch", not "German"; similarly, there exist people identified in Turkish as "Rumlar", a term translatable to English as "Byzantines". Should people use English exonyms for themselves in order to be respected?
> > > > > > 3. Yes, that was my point: that English exonym was invented in the 19th century in reference to people of the Qing Empire, and people do not use it to identify themselves. Does that render the term appropriate to use as a technical term?

---

### Official Review · Reviewer_GiVQ · 2022-10-24

**Confidence:** 4
**Correctness:** 3
**Technical Novelty And Significance:** 3
**Empirical Novelty And Significance:** 2
**Recommendation:** 5

**Clarity, Quality, Novelty And Reproducibility:**

Clarity: The presentation is clear.

Novelty: See W2

Reproducibility: Experimental details and codes are provided.


**Strength And Weaknesses:**

 **Strength:**

- S1. The theoretical perspective of proposed algorithms is studied thoroughly.
- S2. The presentation is clear.


**Weakness:**
- W1. (Theory.) The proof of Byz-VR-MARINA seems to rely on different assumptions (see assumptions 2.3, 2.4) with other works in Table 2. Can the authors compare these assumptions, and is it possible to conclude which one is stronger? Another question is that for those convergence rates marked with x in Table 2, is it impossible to establish? Or the original paper did not report? For example, in the nonconvex setting, if combining the analysis to nonconvex SAGA and Byrd-SAGA, is it possible to get a rate?

- W2. (Methodology.) This paper combines several techniques from existing works, such as VR-MARINA and robust aggregator. This makes the novelty boils down to applying existing works to the byzantine setting, which seems not sufficient for conferences like ICLR.

- W3. (Experiments.) The numerical results can be improved in multiple aspects. **i)** No variance reduced method is compared. It is helpful to include the performance of e.g., VR-MARINA to verify how much the proposed algorithm is affected by byzantine workers. In addition, no alternatives from byzantine-tolerant VR methods are compared. Although Bryd-SAGA uses additional memory, but the datasets e.g., a9a, is small and such memory consumption is affordable. Even if Bryd-SAGA is not a good choice to compare with, the authors can compare to heuristics such as Bryd-SARAH / Bryd – SVRG.  **ii)** The numerical experiments do not reflect the influences of gradient compression. No runtime is compared. **iii)** Although claimed for nonconvex problems, most of tests are on logistic regression, which is convex.


**Summary Of The Paper:**

This paper studies Byzantine-robust variance reduced methods, and further combines them with gradient compression for communication efficiency. The paper takes a theoretical point of view, and numerical results on simple tasks such as logistic regression are provided.

**Summary Of The Review:**

This paper tackles Byzantine-robust distributed optimization mostly from a theoretical perspective. However, the comparison with related works seems difficult to carry out due to the different assumptions in analysis seems, hence its difficult to evaluate the theoretical contribution. In addition, the numerical results are relatively thin and cannot fully support the claimed merits.

---

> ### Author Response · Authors · 2022-11-11
> **Authors' response to Reviewer GiVQ [Part 2/2]**
>
> > **This paper combines several techniques from existing works, such as VR-MARINA and robust aggregator. This makes the novelty boils down to applying existing works to the byzantine setting, which seems not sufficient for conferences like ICLR.**
>
> We believe that it is not always an easy task to combine known techniques to get some improvements in the convergence rates. Some techniques cannot be combined in general: for example, it is known that gradient descent is optimal for finding first-order stationary points in the non-convex smooth case among first-order methods [1]. Therefore, it is in general impossible to combine it with Nesterov’s acceleration to achieve theoretically better rate. Next, even when the techniques are combinable, it can be a non-trivial task to propose good combination and to rigorously analyze it. For example, Nesterov’s acceleration was discovered in 1980-th and SGD was proposed in 1950-th, but the first Accelerated SGD was proposed and analyzed only in 2012 [2].
>
> Moreover, in paragraph “Challenges in designing variance-reduced algorithm with tight rates and provable Byzantine-robustness” (page 7), we explain why it is not trivial to achieve the results that we derive. Although our work positions Byz-VR-MARINA as a natural combination of variance reduction and Byzantine-robustness, it was not straightforward beforehand whether VR-MARINA and robust aggregation are combinable, whether such a combination should be considered, and whether it would lead to the new SOTA theoretical results in Byzantine-robust distributed optimization.
>
> References:
>
> [1] Arjevani, Y., Carmon, Y., Duchi, J. C., Foster, D. J., Srebro, N., & Woodworth, B. (2022). Lower bounds for non-convex stochastic optimization. Mathematical Programming, 1-50.
>
> [2] Ghadimi, S., & Lan, G. (2012). Optimal stochastic approximation algorithms for strongly convex stochastic composite optimization i: A generic algorithmic framework. SIAM Journal on Optimization, 22(4), 1469-1492.
>
> ---
>
> > **No variance reduced method is compared. It is helpful to include the performance of e.g., VR-MARINA to verify how much the proposed algorithm is affected by byzantine workers. In addition, no alternatives from byzantine-tolerant VR methods are compared. Although Bryd-SAGA uses additional memory, but the datasets e.g., a9a, is small and such memory consumption is affordable. Even if Bryd-SAGA is not a good choice to compare with, the authors can compare to heuristics such as Bryd-SARAH / Bryd – SVRG.**
>
> Thank you for the suggestion. Although Byrd-SAGA does not fit well the current implementation, Byrd-SVRG fits and can be considered as a good proxy of Byrd-SAGA (given the similarities in theory and practical behavior of SAGA and SVRG). We are currently running additional experiments with Byrd-SVRG. We will add the results to the paper soon.
>
> We also would like to note here that our paper focuses on the theoretical convergence guarantees and achieving new theoretical SOTA results in such a popular field as Byzantine-tolerant training is important. We note that all the provided experiments are well aligned with the theoretical results and do not undermine it.
>
> ---
>
> > **The numerical experiments do not reflect the influences of gradient compression. No runtime is compared.**
>
> Since we run the methods in a simulated distributed environment, the runtime comparison will not reflect the influence of the compression fairly. Instead, we can report the number of transmitted bits, which is a software and hardware independent analog of the runtime. We will add the comparison in terms of the number of transmitted bits soon (and compare with the methods without the compression).
>
> ---
>
> > **Although claimed for nonconvex problems, most of tests are on logistic regression, which is convex.**
>
> We are working on the extra experiments right now. In particular, we plan to add some results for non-convex problems (logistic regression with non-convex regularization).

---

> ### Author Response · Authors · 2022-11-11
> **Authors' response to Reviewer GiVQ [Part 1/2]**
>
> > **The proof of Byz-VR-MARINA seems to rely on different assumptions (see assumptions 2.3, 2.4) with other works in Table 2. Can the authors compare these assumptions, and is it possible to conclude which one is stronger?**
>
> Thank you for your comment. We have added the detailed discussion of the used assumptions to Appendix E.5. Below we also provide this comparison for the reviewer’s convenience.
>
> Many existing works rely on the uniformly bounded variance assumption (UBV): it is assumed that for all $x \in \mathbb{R}^d$ the good workers have an access to the unbiased estimators $g_i(x)$ of $\nabla f_i(x)$ such that $\mathbb{E}\|g_i(x) - \nabla f_i(x)\|^2 \leq \sigma^2$ for all $i \in \mathcal{G}$ and $\sigma \geq 0$. This assumption does not hold in many practical situations and even for simple convex finite-sum problems like sums of quadratic functions with non-identical Hessians. Moreover, in the situations when this assumption holds, the value of $\sigma^2$ can be huge. However, UBV assumption does not require individual stochastic realizations, i.e., summands $f_{i,j}$, to be smooth.
>
> In contrast, we use Assumption 2.4 that holds for many situations when UBV assumption does not. For example, Assumption 2.4 holds whenever all functions $f_{i,j}$, $i \in \mathcal{G}$, $j \in m$ are $L_{i,j}$-smooth (see Appendix E.1). These facts allow us to cover a large class of problems that does not fit the setup considered in [1,2,3]. Moreover, since Assumption 2.4 is more general than smoothness of all $f_{i,j}$, our analysis covers the setup considered in [4, 5]. However, it is worth mentioning that there exist problems such that UBV assumption holds and Assumption 2.4 does not, e.g., when the gradient noise is additive: $\widehat{\Delta}_i(x,y) = \nabla f_i(x) - \nabla f_i(y) + \xi_i$, where $\mathbb{E}\xi_i = 0$ and $\mathbb{E}\|\xi_i\|^2 = \sigma^2$.
>
> References:
>
> [1] Sai Praneeth Karimireddy, Lie He, and Martin Jaggi. Learning from history for byzantine robust optimization. In International Conference on Machine Learning, pp. 5311–5319. PMLR, 2021
>
> [2] Sai Praneeth Karimireddy, Lie He, and Martin Jaggi. Byzantine-robust learning on heterogeneous datasets via bucketing. International Conference on Learning Representations, 2022.
>
> [3] Eduard Gorbunov, Alexander Borzunov, Michael Diskin, and Max Ryabinin. Secure distributed training at scale. In International Conference on Machine Learning, pp. 7679-7739. PMLR, 2022.
>
> [4] Zhaoxian Wu, Qing Ling, Tianyi Chen, and Georgios B Giannakis. Federated variance-reduced stochastic gradient descent with robustness to byzantine attacks. IEEE Transactions on Signal Processing, 68:4583–4596, 2020
>
> [5] Heng Zhu and Qing Ling. Broadcast: Reducing both stochastic and compression noise to robustify communication-efficient federated learning. arXiv preprint arXiv:2104.06685, 2021
>
> ---
>
> > **Another question is that for those convergence rates marked with x in Table 2, is it impossible to establish? Or the original paper did not report? For example, in the nonconvex setting, if combining the analysis to nonconvex SAGA and Byrd-SAGA, is it possible to get a rate?**
>
> The “x” mark in the table means that the original paper does not contain the result in the corresponding case. We have not tried to adjust the analysis of SAGA in the non-convex case to Byrd-SAGA. However, taking into account the intuition that we described in the paragraph “Challenges in designing variance-reduced algorithm with tight rates and provable Byzantine-robustness” (page 7) we expect similar drawbacks of the convergence rate of Byrd-SAGA as the ones that we discussed in that paragraph. Taking into account that Byrd-SAGA has the drawbacks even in the strongly convex case, it reduces the potential interest of the extension of its analysis to the non-convex case.

---

> ### Author Response · Authors · 2022-11-17
> **Additional experiments: comparison with Byrd-SVRG**
>
> In the following figure (https://ibb.co/Vmk0Jsn), we compare Byrd-SVRG and Byz-VR-Marina, as suggested by the reviewer. The setup that we consider is the same as we consider in the main paper with compression, with the difference that we use a non-convex penalty
> $\lambda \sum_{i=1}^d \frac{x_i^2}{1 + x_i^2}$
> as we propose in the response, instead of standard L2 regularization.
>
> One can note that Byz-VR-MARINA converges to a reasonable solution while Byrd-SVRG cannot provide the same quality solution as the theory suggests due to problems with handling biasedness and extra noise coming from the compression operator.

---

> ### Author Response · Authors · 2022-11-21
> **Additional experiment: non-convex loss function**
>
> In the following figure (https://ibb.co/2WywY6X), we compare all the methods together, including Byrd-SVRG, as suggested by the reviewer. The setup that we consider is the same as we consider in the main paper **without compression** (in the results reported in our previous message https://openreview.net/forum?id=pfuqQQCB34&noteId=Bh_k0Gzvh_Y, compression was used), with the difference that we use a non-convex penalty
> $\lambda \sum_{i=1}^d \frac{x_i^2}{1 + x_i^2}$
> as we propose in the response, instead of standard L2 regularization.
>
> One can note that Byz-VR-MARINA converges to an optimal solution. In contrast, all the baseline methods converge only to the neighborhood as predicted by the theory due to problems discussed before. For instance, an issue with handling biasedness (Byrd-SVRG), non-robust aggregation (SGD, Byrd-SVRG) and non-reducing variance due to stochastic gradients (SGD, BR-SGD with momentum).

---

### Official Review · Reviewer_QBBT · 2022-10-28

**Confidence:** 3
**Correctness:** 4
**Technical Novelty And Significance:** 3
**Empirical Novelty And Significance:** 2
**Recommendation:** 6

**Clarity, Quality, Novelty And Reproducibility:**

- The paper is well written and is definitely of high quality.
- The novelty of the paper is medium, because of it being a combination of existing ideas.
- The authors have provided a link to their code. But the code is simply a collection of files and the README file is barebones. It is therefore hard to judge the reproducibility power of this.

**Strength And Weaknesses:**

**Strengths**

- This appears to be first work in the literature that focuses on communications efficiency in Byzantine-robust distributed optimization.
- The paper includes a rigorous set of theoretical guarantees for both the general nonconvex functions as well as functions that satisfy the PL condition. In addition, the analysis in the paper includes the case of heterogeneous data.

**Weaknesses**

- In most works, the theory is presented for functions that are either strongly strongly convex or that satisfy the PL condition, but the experiments are then presented for nonconvex functions. It is strange that the experiments in this paper are only for strongly convex functions. Experiments with a total of five workers and just one of them being Byzantine are also not convincing.
- At some level, the paper appears to be a mash-up of ideas from two existing works. The authors could have done a better job of clarifying the challenges they have to face in terms of this.

**Summary Of The Paper:**

The focus of this paper is on communication-efficient and Byzantine-robust distributed optimization. To this end, the authors propose an algorithm Byz-VR-MARINA that combines ideas from the Byzantine-robust distributed optimization work of Karimireddy et al., 2021 and communication-efficient distributed optimization work of Gorbunov et al., 2021b. The main contributions of the work include both theoretical guarantees for the proposed algorithm as well as numerical experiments that highlight the effectiveness of the proposed algorithm.

**Summary Of The Review:**

Overall, it is a solid piece of work that primarily suffers from somewhat limited novelty and also lack of appropriate experimental results.

---

> ### Author Response · Authors · 2022-11-11
> **Authors' response to Reviewer QBBT**
>
> > **In most works, the theory is presented for functions that are either strongly strongly convex or that satisfy the PL condition, but the experiments are then presented for nonconvex functions. It is strange that the experiments in this paper are only for strongly convex functions. Experiments with a total of five workers and just one of them being Byzantine are also not convincing.**
>
> We are working on the extra experiments right now. In particular, we plan to add some results for non-convex problems (logistic regression with non-convex regularization) and also experiments with a bigger number of workers.
>
> We also would like to note here that our paper focuses on the theoretical convergence guarantees and achieving new theoretical SOTA results in such a popular field as Byzantine-tolerant training is important. We note that all the provided experiments are well aligned with the theoretical results and do not undermine it.
>
> ---
>
> > **At some level, the paper appears to be a mash-up of ideas from two existing works. The authors could have done a better job of clarifying the challenges they have to face in terms of this.**
>
> We explain the challenges on page 7, see the paragraph “Challenges in designing variance-reduced algorithm with tight rates and provable Byzantine-robustness”. In particular we list several observations that we made before obtaining our results. We believe these observations are not trivial beforehand. We are happy to improve this part of the paper if the reviewer has some particular questions/comments related to this.
>
> ---
>
> > **The authors have provided a link to their code. But the code is simply a collection of files and the README file is barebones. It is therefore hard to judge the reproducibility power of this.**
>
> Thank you for raising this issue, but we respectfully disagree with this claim. We believe that the provided code is sufficient to replicate the experiments presented in our paper fully. As mentioned in the README file, all the run scripts are provided, which can be used to reproduce the results displayed in the paper. Furthermore, we also provide the jupyter notebook to produce the plots shown in the manuscript. Finally, the provided code is supposed to be self-explanatory. For instance, in provided run scripts, all considered parameters are part of the script. Also, the implementation itself is organized in a modular hierarchical way with self-explanatory names, e.g., aggregation, attacks, compressors, etc., where the reader can verify the correctness of each component.
>
> We would be very grateful if the reviewer could suggest ways to improve the readability of our code if something needs to be added or clarified.

---

### Official Review · Reviewer_kABX · 2022-11-01

**Confidence:** 4
**Correctness:** 3
**Technical Novelty And Significance:** 4
**Empirical Novelty And Significance:** 3
**Recommendation:** 8

**Clarity, Quality, Novelty And Reproducibility:**

The paper is quite clear with high quality and novelty. The steps to reproduce the experiment results are presented in the paper.

**Strength And Weaknesses:**

Strength:
- The paper is well-written and covers a sufficient amount of related works.
- The idea of using variance reduction for Byzantine robustness is natural but it is straightforward to make the rate tight. This paper did a very good job of proving the rates under standard or weak assumptions.
- The gradient compression improves the communication efficiency but has little influence on the convergence rate.
- This paper additionally allows non-uniform sampling of stochastic gradient which is not seen in previous works.

Weakness / Possible improvement:
- What's the relation between b and m? For variance reduction techniques like SPIDER, the optimal choice of b is $b=\sqrt{m}$. If $m$ is significantly large than b, then in Table 1 (homogeneous + no compression + NC), Byz-VR-MARINA has a scaling of $c\delta\frac{m}{b} + 1/n$ which seems to be worse than BR-SGDm $c\delta + 1/n$.
- The probability p looks very interesting and can be further discussed. For example，
    - The p serves two goals in variance reduction and compression. If $b=\sqrt{m}$ for SPIDER, does it mean we may not simultaneously choose the best p for variance reduction and compression?
    - The lower bound term in (7) has a coefficient 1/p. Is it optimal?
    - Eq. (7) suggests that while variance reduction improves the convergence rate but at the same time makes the lower bound worse. Is there any way we can circumvent this? The overparameterization trick in (Karimireddy et al., 2021) seems not working as it increases the gradient size.



**Summary Of The Paper:**

This paper considers the Byzantine resilient optimization problem and proposes to use variance reduction and compression for improved convergence rates. More concretely, they propose a novel method, Byz-VR-MARINA, which integrates VR-MARINA and an unbiased gradient compressor together for Byzantine robustness. They show that Byz-VR-MARINA enjoys a better convergence rate under weaker assumptions when compared to previous works. The empirical evaluations also show that Byz-VR-MARINA has outstanding performances.

**Summary Of The Review:**

I recommend accepting this paper as it gives good theoretical analysis and good empirical performance.

---

> ### Author Response · Authors · 2022-11-11
> **Authors' response to Reviewer kABX [Part 2/2]**
>
> > **The $p$ serves two goals in variance reduction and compression. If $b = \sqrt{m}$ for SPIDER, does it mean we may not simultaneously choose the best p for variance reduction and compression?**
>
> This is an excellent comment.
>
> - The expected number of oracle calls per iteration is $2b(1-p) + mp$ meaning that $p = b/m$ makes the expected oracle cost of each iteration equal to $\mathcal{O}(b)$ like in the case of SGD.
>
> - To measure the communication efficiency one can use expected density $\zeta_{\mathcal{Q}}$ (see Definition 2.2). Then, the expected number of components that each worker sends to the server at each step is upper-bounded by $\zeta_{\mathcal{Q}}(1-p) + pd$ meaning that $p = \zeta_{\mathcal{Q}}/d$ makes the expected number of components that each worker sends to the server at each step equal to $\mathcal{O}(\zeta_{\mathcal{Q}})$. In the case when $1 + \omega = \Theta(d / \zeta_{\mathcal{Q}})$, one can choose $p = 1 / (1+ \omega)$.
>
> As we explain in footnote 3, to achieve both $\mathcal{O}(b)$ expected oracle cost and $\mathcal{Q}(\zeta_{\mathcal{Q}})$ expected number of components that each worker sends to the server at each step, one can take $p = \min\{b/m, 1/(1+\omega)\}$. However, as the reviewer fairly noticed, the best $p$ for variance reduction and the best $p$ for compression are different in general. When $b/m < 1/(1+\omega)$, the choice $p = \min\{b/m, 1/(1+\omega)\}$ implies that the algorithm could use uncompressed vectors more often without sacrificing the communication cost, and when $b/m > 1/(1+\omega)$, the choice $p = \min\{b/m, 1/(1+\omega)\}$ implies that the algorithm could use full gradients more often without sacrificing the oracle cost. That is, the choice of $p \approx b/m$ implies better oracle complexity and the choice $p \approx 1/(1+\omega)$ leads to better communication efficiency. Depending on how much these two aspects are important for the particular application, one can choose $p$ in between these two values.
>
> We have added the discussion to Appendix E.5.
>
> ---
>
> > **The lower bound term in (7) has a coefficient 1/p. Is it optimal?**
> > ** Eq. (7) suggests that while variance reduction improves the convergence rate but at the same time makes the lower bound worse. Is there any way we can circumvent this? The overparameterization trick in (Karimireddy et al., 2021) seems not working as it increases the gradient size.**
>
> The second term in (7) is not optimal: as Karimireddy et al. (2022) show, one can achieve $\mathbb{E}||\nabla f(x)||^2 = \mathcal{O}(\delta\zeta^2)$, which is $1/p$ times smaller than the second term from (7). Formally speaking, it is unclear for us whether it is possible to improve the second term in (7) using the current analysis. However, the smaller choice of $p$ makes the updates more noisy. Therefore, it is natural to expect that this will lead to the worse accuracy of the solution that the algorithm is able to achieve, though we do not have a rigorous proof this is necessarily the case.
>
> Similarly to Karimireddy et al., (2022), one can use a slightly more general assumption about heterogeneity (see Assumption E.1). In the appendix, we provide the results under this assumption (see Theorems E.1 and E.2). We emphasize that when $\zeta = 0$ and $B > 0$, the problem is allowed to be heterogeneous, while the method can converge to any accuracy of the solution. However, in this case, the maximal ratio of Byzantine workers allowed in theory becomes smaller than in the case of $B = 0$ (see the second inequality from (28)). Similar phenomenon is observed in Karimireddy et al., (2022), though we have $1/p$ times smaller $\delta$ in this case.
>
> We also would like to note that during the work on the paper we were mostly motivated by the homogeneous case. As we explain after Assumption 2.2 the case of $\zeta = 0$ (and $B =0$) is realistic in collaborative learning. In this case, Byz-VR-MARINA can converge to any accuracy of the solution and we do not make any additional assumptions on $\delta$ beyond ones from Karimireddy et al., (2022).

---

> ### Author Response · Authors · 2022-11-11
> **Authors' response to Reviewer kABX [Part 1/2]**
>
> We thank the reviewer for a positive evaluation of our paper and excellent comments.
>
> > **What's the relation between $b$ and $m$? For variance reduction techniques like SPIDER, the optimal choice of $b$ is $b = \sqrt{m}$. If $m$ is significantly large than $b$, then in Table 1 (homogeneous + no compression + NC), Byz-VR-MARINA has a scaling of $c\delta \frac{m}{b} + \frac{1}{n}$ which seems to be worse than BR-SGDm .**
>
> We thank the reviewer for the good question. We have added the detailed discussion of the choice of $b$ and how the complexity scales with $b$ to the paper (see Appendix E.5). Below we also provide this discussion for the reviewer’s convenience.
>
> First, we note that our analysis is valid for any choice of $b \geq 1$. For simplicity of the further discussion of the batchsizes role in the complexities, consider the homogeneous case ($B = 0$, $\zeta = 0$) without compression ($\omega = 0$). As Corollary E.1 states, the communication complexity of Byz-VR-MARINA in this case equals
>
> $$
> 	\mathcal{O}\left(\frac{\left(L + {\mathcal{L}}_{\pm}\sqrt{\frac{c\delta m^2}{b^3} + \frac{m}{b^2G}}\right)\Delta_0}{\varepsilon^2}\right).
> $$
>
> Note that the term depending on the ratio of Byzantines $\delta$ scales as $b^{-3/2}$ with the batchsize and the term depending on $1/G$ scales as $b^{-1}$. Table 2 illustrates that previous SOTA results in this case scale as $b^{-1}$ or $b^{-1/2}$, so, the complexity bound for Byz-VR-MARINA scales with $b$ no worse than the concurrent bounds.
>
> Next, as the reviewer fairly noticed, typically, there is no need to take $b$ larger than $\sqrt{m}$ for SARAH-based variance reduced methods [1, 2]: oracle complexity is always the same (neglecting the differences in the smoothness constants), while the iteration complexity stops improving once $b$ becomes larger than $\sqrt{m}$. However, the complexity bound for Byz-VR-MARINA contains the non-standard term
>
> $$\mathcal{O}\left(\frac{{\mathcal{L}}_{\pm}m\sqrt{c\delta}}{\sqrt{b^3}\varepsilon^2}\right)$$
>
>
> appearing due to the presence of Byzantine workers. For simplicity, we assume that
>
> $$L = \Theta({\mathcal{L}}_{\pm})$$
>
> (though $\mathcal{L}_{\pm}$ can be both smaller and larger than $L$). Then, when we increase batchisze $b$, the communication complexity stops improving once $b$ becomes larger than $\max\{\sqrt[3]{c\delta m^2}, m\}$. Interestingly, $\sqrt[3]{c\delta m^2}$ can be larger than the standard value $\sqrt{m}$: this is the case when $m > 1/(c^2\delta^2)$. In this case, the communication complexity of Byz-VR-MARINA benefits from the slightly larger batchsizes than in the classical case. This phenomenon has a natural explanation: when we increase the batchsize, the variance of the gradient noise decreases and it becomes even harder for Byzantines to shift the updates of the method significantly.
>
> References:
>
> [1] Horváth, S., Lei, L., Richtárik, P., & Jordan, M. I. (2022). Adaptivity of stochastic gradient methods for nonconvex optimization. SIAM Journal on Mathematics of Data Science, 4(2), 634-648.
>
> [2] Li, Z., Bao, H., Zhang, X., & Richtárik, P. (2021, July). PAGE: A simple and optimal probabilistic gradient estimator for nonconvex optimization. In International Conference on Machine Learning (pp. 6286-6295). PMLR.

---

### Author Response · Authors · 2022-11-11
**Authors' response to all reviewers**

We thank the reviewers for their time and constructive feedback. We would like to notice first that the reviewers acknowledged the different strengths of our work.

**Reviewer kABX:**

- Byz-VR-MARINA enjoys a better convergence rate under weaker assumptions when compared to previous works.

- The empirical evaluations also show that Byz-VR-MARINA has outstanding performances.

- The paper is well-written and covers a sufficient amount of related works.

- This paper did a very good job of proving the rates under standard or weak assumptions.

- The gradient compression improves the communication efficiency but has little influence on the convergence rate.

- This paper additionally allows non-uniform sampling of stochastic gradient which is not seen in previous works.

- The paper is quite clear with high quality and novelty. The steps to reproduce the experiment results are presented in the paper.

**Reviewer QBBT:**

- The paper includes a rigorous set of theoretical guarantees for both the general nonconvex functions as well as functions that satisfy the PL condition.

- In addition, the analysis in the paper includes the case of heterogeneous data.

- The paper is well written and is definitely of high quality.

**Reviewer GiVQ:**

- The theoretical perspective of proposed algorithms is studied thoroughly.

- The presentation is clear.

**Reviewer 9mHu:**

- The main idea of variance reduction is well presented.

-  I appreciate that theoretical analysis is provided for both non-convex objective functions and functions satisfying PL inequality.

- Most parts of this paper are clearly written.

The reviewers also raised several concerns and very good questions/suggestions. In our detailed responses below, we believe we properly addressed all of them. We have also applied several corrections to our paper (the revised version has been uploaded, the changes are highlighted in olive color). Due to the space limitations we put some newly added discussions to the appendix, but in case of acceptance we will be able to move them (at least partially) to the main part of the paper using the additional 10-th page. We would highly appreciate it if the reviewers checked the corrected places we highlighted in olive

Following the reviewers’ requests, we are currently running additional experiments. We will add new numerical results to the paper as soon as possible and notify the reviewers in a separate message.

We are also happy to engage in further discussions and provide additional clarifications/comments if needed.

---

### Decision · Program_Chairs · 2023-01-20

**Decision:**

Accept: poster

**Justification For Why Not Higher Score:**

The numerical demonstration is somewhat stylistic with 5 workers with only one of them being 1 Byz-worker.

**Justification For Why Not Lower Score:**

The presentation is simply great with the proofs, contributions, and the ideas. The result is also quite solid.

**Metareview: Summary, Strengths And Weaknesses:**

The work builds a distributed optimization framework that uses variance reduction as a main tool to cope with a training setting that has adversarial clients. The approach handles smooth non-convex losses, incorporates communication compression in its theoretical characterization, allows non-uniform sampling of stochastic gradients, and can deal with heterogeneous data. The approach also features a tight theoretical characterization when there is no adversarial worker (aka Byz-worker, as modified by the authors after the ethics discussions with one of the reviewers).

The authors do a great job at explaining the background, highlighting the contributions, and summarizing the theoretical results. Numerical evidence shows that the resulting algorithm can attain better objective rates with their Byz modifications than the sota variance reduction algorithm or the distributed SGD.

**Note From Pc:**

if the above contains the word "oral" or "spotlight" please see: "oral" presentation means -> notable-top-5% and "spotlight" means -> notable-top-25%. As stated in our emails, we are disassociating presentation type from AC recommendations